# Context-Driven Nearest Neighbor Imputation using Language Representation

## Abstract

Missing data poses significant challenges for machine learning and deep learning algorithms. In this paper, we aim to enhance post-imputation performance, measured by machine learning utility (MLu). We introduce a nearest-neighbor-based imputation method, DrIM, designed for heterogeneous tabular datasets. However, calculating similarity in the data space becomes challenging due to the varying presence of missing entries across different columns. To address this issue, we leverage the representation learning capabilities of language models. By transforming the tabular dataset into a text-format dataset and replacing the missing entries with mask (or unk) tokens, we extract representations that capture contextual information. This mapping to a continuous representation space enables the use of well-defined similarity measurements. Additionally, we incorporate a contrastive learning framework to refine the representations, ensuring that the representations of observations with similar information in the observed columns, regardless of the missingness patterns, are closely aligned. To validate our proposed model, we evaluate its performance in missing data imputation across 10 real-world tabular datasets, demonstrating its ability to produce a Complete dataset having high MLu.

## 1 Introduction

Handling missing data is challenging, as machine learning (ML) and deep learning (DL) methods typically require complete datasets (Tan et al., 2013; Kingma & Welling, 2014; Goodfellow et al., 2014; Chen & Guestrin, 2016; Ke et al., 2017; Vaswani et al., 2017; Ho et al., 2020; An & Jeon, 2023). Rubin says that the goal of imputation tasks is to provide complete statistical inference (Rubin & Schenker, 1986). However, many recent imputation methods have focused on providing complete datasets that can yield high post-imputation performance, including machine learning utility (MLu). Ivanov et al. (2019); Yoon et al. (2018); Mattei & Frellsen (2019); Ipsen et al. (2021); Zhao et al. (2023); Du et al. (2024a); Wen et al. (2024) emphasize the importance of imputing missing data in terms of MLu. In this paper, we also focus on imputing incomplete datasets to provide a complete dataset that enables effective training of many ML and DL algorithms while achieving high MLu.

We empirically observed that nearest-neighbor-based imputation methods, such as $k$-nearest neighbors imputation (kNNI) (Troyanskaya et al., 2001) and its variants (Jiang & Yang, 2015; Thomas & Rajabi, 2021; Du et al., 2024b), can achieve high MLu. Moreover, Anil Jadhav & Ramanathan (2019); Keerin & Boongoen (2022); Ding et al. (2024) have demonstrated that these methods often outperform other imputation techniques. These nearest-neighbor-based methods impute missing values by leveraging the similarities between observations and utilizing the values of their nearest neighbors, thereby preserving local data characteristics (Troyanskaya et al., 2001; Tarsitano & Falcone, 2011).

However, nearest-neighbor-based imputation methods face two key challenges: 1) They struggle to effectively represent or address missing values. While various distance or similarity metrics have been proposed for missing data, their performance depends on how they handle these missing values (Juhola & Laurikkala, 2007; Muñoz & Hernández-González, 2012; AbedAllah & Shimshoni, 2016; Santos et al., 2020). For instance, assigning a distance of $0$ when both values are missing can be inflexible and adversely affect the imputation process (Santos et al., 2020). 2) Applying these methods to heterogeneous (i.e., mixed-type) tabular data presents challenges. Jerez et al. (2010);

García-Laencina et al. (2010) have demonstrated that continuous and categorical variables exhibit an imbalance when measuring distance, with continuous variables having a more significant influence on the distance metric, affecting the results of imputation.

Therefore, we introduce a nearest-neighbor-based imputation method termed **DrIM** (Contextual-Driven Missing IMputer) to address these challenges. Our proposed method includes taking advantage of the language model's ability to generate contextualized representations, which are learned from a large collection of text datasets. We utilize textual encoding to convert tabular data into text-format data and apply language models to obtain representations (Yin et al., 2020; Mei et al., 2021; Borisov et al., 2023; Radford & Narasimhan, 2018; Devlin et al., 2019; Nazir et al., 2023).

Our representation-based nearest-neighbor imputation method provides solutions to address challenges 1) and 2), respectively: 1) Missing values are represented as `[MASK]` (or `[UNK]`) tokens, allowing language models to utilize their representation capabilities and leverage contextual hints from the other columns. 2) By mapping both continuous and categorical columns into a continuous space using their representation vectors, we enable the use of well-defined similarity measures, such as cosine similarity. Furthermore, we incorporate fine-tuning using a contrastive learning framework to further enhance the language model's representational capacity. This allows it to build more meaningful neighbors, which in turn improves the post-imputation performance.

Our primary contributions can be summarized as follows:

1. By adopting language models and transforming each record into a representation vector, we propose a nearest-neighbor-based imputation method that is applicable to heterogeneous tabular datasets.

2. We achieve state-of-the-art imputation performance in terms of MLu without any model training.

3. We provide an intuitive explanation while empirically confirming that our fine-tuning using contrastive learning enhances the MLu of the imputed dataset.

We validate the effectiveness of our proposed method by evaluating its imputation performance across 10 real-world tabular datasets, accounting for four missing data mechanisms and five missingness rates. We employ four MLu performance metrics, including classification tasks, model selection, and feature selection performance. Additionally, we demonstrate that applying a contrastive learning framework enhances imputation performance in terms of MLu.

## 2 RELATED WORKS

**Imputation methods.** The effectiveness of missing imputation ultimately lies in reducing bias and enhancing estimation efficiency in inference after imputation (Little & Rubin, 2019). This always depends on the missing data mechanism, and in general, a single imputation has the limitation of not accounting for the uncertainty induced by the imputation (van Buuren, 2012). Nevertheless, single imputation is widely used because of its convenience and its ability to improve estimation efficiency in downstream tasks (Álvarez Verdejo et al., 2021). In particular, a single imputation based on estimating conditional expectation performs well when the bias introduced by the imputation does not significantly affect the performance of the specific downstream task (Troyanskaya et al., 2001). Deep learning has garnered significant attention for its ability to flexibly model conditional expectations of observed data and effectively learn metrics in the observed data space (Yoon et al., 2018; Ivanov et al., 2019; Kyono et al., 2021; Wen et al., 2024). These two tasks are closely related to the nonparametric kNNI. kNNI is a method for calculating local conditional expectations and involves dimension reduction and metric learning to compute conditional expectations effectively.

**Distance measures for missing data.** kNNI is a widely used imputation method because it effectively utilizes the similarity between patterns to generate accurate estimates (Wilson & Martinez, 1997) and maintains the overall data distribution (Santos et al., 2017). However, its performance heavily relies on the chosen distance function. Several distance functions have been developed to handle heterogeneous tabular data and account for missing values. The HEOM (Aha et al., 1991; Wilson & Martinez, 1997) calculates distances based on feature types, using normalized Euclidean distance for continuous features and an overlap metric for categorical ones. Missing values are assigned a distance of 1, and if both are missing, 0. Juhola & Laurikkala (2007) refined HEOM to

maintain this approach. The HVDM (Stanfill & Waltz, 1986; Wilson & Martinez, 1997) functions similarly to HEOM but requires class target information for categorical features. SIMDIST (Muñoz & Hernández-González, 2012) introduces a heterogeneous similarity function incorporating prior knowledge, with similarity based on the presence or absence of values. Additionally, redefined versions of HVDM treat missing values as special cases (Santos et al., 2020), and the Mean Euclidean Distance (AbedAllah & Shimshoni, 2016) has been used for managing incomplete data in clustering. In contrast, our approach utilizes metric learning for imputing missing values. Instead of relying on existing distance functions, our method learns a distance (similarity) metric on a manifold using language models, specifically by computing the Euclidean distance between the representation vectors obtained from language models.

**Notations.** Suppose that an observation $\mathbf{x} \in \mathcal{X}_1 \times \cdots \times \mathcal{X}_p$ consists of both continuous and categorical variables (columns), where $\mathcal{X}_j$ denotes the support of $j$th variable. $I_C$ and $I_D$ represent the index sets for continuous and categorical variables, respectively, where $I_C \cup I_D = \{1, \cdots, p\}$. $\mathbf{x}_j$ is a value of the $j$th column having its column name with $C_j$. Here, subscript $j$ refers to the $j$th element. $g(\cdot; \theta)$ is a function that extracts the representation vector from the text input, where $\theta$ is its trainable parameter. The missingness pattern of an observation is defined by a corresponding missingness indicator vector $\mathbf{m} \in \{0, 1\}^p$, where $\mathbf{m}_j = 0$ if $\mathbf{x}_j$ is missing. An incomplete dataset is given $\{(\mathbf{x}^{(i)}, \mathbf{m}^{(i)})\}_{i=1}^n$ comprising $n$ i.i.d. realizations of $\mathbf{x}$ and $\mathbf{m}$.

## 3 METHODOLOGY

DrIM[1] is a single imputation method based on the kNNI employing representations for each tubular record using language models. The main difference between the DrIM and the conventional kNNI approach lies in the way the distance metric between data points is learned using an underlying model for imputing missing values. This distance metric is defined as the Euclidean distance between the representation vectors of `[CLS]` tokens output by the BERT model (Devlin et al., 2019)[2]. We will elucidate its theoretical properties in this section.

### 3.1 REPRESENTATION OF BERT

In this section, we will present some theoretical results demonstrating how the $L_2$-distance between BERT representation vectors can reflect the similarity between observations. In the pre-training process of BERT, suppose that the pre-training dataset consists of paired sequences, $(\mathbf{t}, \mathbf{u}) \in \mathcal{T} \times \mathcal{U}$, where all paired sequences in $\mathcal{T} \times \mathcal{U}$ have the same fixed total length. Let $p(\mathbf{t}, \mathbf{u}), p(\mathbf{t})$, and $p(\mathbf{u})$ are joint and marginal distributions of $\mathbf{t}$ and $\mathbf{u}$, respectively. Let $\mathbf{e} \in \mathcal{U}$ be a sequence of the `[PAD]` tokens. We formally define the representation of BERT in the following definition.

**Definition 1** (Representation of BERT). *Let $g(\cdot, \cdot; \theta) : \mathcal{T} \times \mathcal{U} \mapsto \mathbb{R}^D$ is a map of output vector located at the position of the `[CLS]` token in BERT. Then,*

1. *in the pre-training of BERT, the representation of BERT for $\mathbf{t} \in \mathcal{T}$ with respect to $\mathbf{u} \in \mathcal{U}$ is defined as $g(\mathbf{t}, \mathbf{u}; \theta)$;*

2. *in the process of DrIM for $\mathbf{t} \in \mathcal{T}$, the representation of BERT is defined as $g(\mathbf{t}, \mathbf{e}; \theta)$.*

In Definition 1, $\mathbf{u} \in \mathcal{U}$ represents auxiliary data used in the next sentence prediction task of BERT, which will be described later. This definition, while restrictive in that it fixes the dimensional sizes of $\mathbf{t}$ and $\mathbf{u}$ in BERT's training, provides a convenient framework for analysis. In Definition 1, note that the sequence $\mathbf{u}$ is replaced with the sequence of `[PAD]` tokens in the process of DrIM.

**Definition 2** (Pre-training of Next Sentence Prediction (NSP) via logistic regression). *In the pre-training of NSP, suppose that the positive and negative samples are sampled from the following distributions:*

$$
\begin{aligned}
(\mathbf{t}, \mathbf{u}) &\sim p(\mathbf{t}, \mathbf{u}) \quad \textit{(positive sample)} \\
(\mathbf{t}, \mathbf{u}') &\sim p(\mathbf{t})p(\mathbf{u}') \quad \textit{(negative sample)}.
\end{aligned}
$$

---

[1]The overall structure of DrIM is outlined in Figure 1.

[2]In this paper, we chose BERT for a comprehensive comparison and performance analysis; however, we have also included the experimental results using GPT-based models in Appendix A.8.

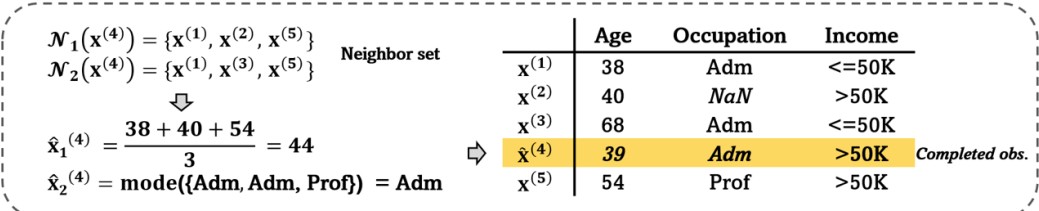

Figure 1: Overall structure of DrIM. In this case, we impute the missing values in $\mathbf{x}^{(4)}$ using 3 nearest neighbors. The neighbor set varies since the presence of missing data differs by column.

*Let a risk function*

$$J_{LR}(\theta, \eta) = -\mathbb{E}_{p(\mathbf{t}, \mathbf{u})} \left[ \log \frac{\exp(r(\mathbf{t}, \mathbf{u}; \theta, \eta))}{1 + \exp(r(\mathbf{t}, \mathbf{u}; \theta, \eta))} \right] - \mathbb{E}_{p(\mathbf{t})p(\mathbf{u}')} \left[ \log \frac{1}{1 + \exp(r(\mathbf{t}, \mathbf{u}'; \theta, \eta))} \right],$$

*where $r(\mathbf{t}, \mathbf{u}; \theta, \eta) = \Phi(g(\mathbf{t}, \mathbf{u}; \theta); \eta)$ and $\Phi(\cdot; \eta) : \mathbb{R}^D \mapsto \mathbb{R}$ is a scoring function parameterized with $\eta$. The training BERT for NSP is defined by estimating $(\theta, \eta)$ that minimizes an empirical version of $J_{LR}$.*

In practice, BERT training is achieved by minimizing two loss functions: one for the masked language model (MLM) and the other for NSP. Here, we focus only on the loss function for NSP training, which can be analyzed within the theoretical framework of contrastive learning in unsupervised learning.

**Theorem 1.** *Suppose that $\Phi(\cdot; \eta) : \mathbb{R}^D \mapsto \mathbb{R}$, the scoring function in Definition 2, is modeled as a linear function, i.e., $\forall \mathbf{z} \in \mathbb{R}^D, \Phi(\mathbf{z}; \eta) = \eta^\top \mathbf{z}$ and BERT is pre-trained by the NSP of Definition 2. Then, for two `[CLS]` tokens of $\mathbf{t}, \mathbf{t}' \in \mathcal{T}$ we have*

$$\|g(\mathbf{t}, \mathbf{e}; \theta) - g(\mathbf{t}', \mathbf{e}; \theta)\| \geq \frac{1}{\|\eta\|} \cdot \left| \log \frac{p(\mathbf{e} \mid \mathbf{t})}{p(\mathbf{e} \mid \mathbf{t}')} \right|,$$

*where $p$ is the distribution defined on sequences from $\mathcal{T} \times \mathcal{U}$.*

Theorem 1 demonstrates that the representation of BERT has the property that the $L_2$-distance between representation vectors can reflect the similarity between observations. Using Bayes' rule, the log-likelihood ratio term in Theorem 1 can be re-written as $\log p(\mathbf{e} \mid \mathbf{t})/p(\mathbf{e} \mid \mathbf{t}') = \log \left( \frac{p(\mathbf{t}, \mathbf{e})/p(\mathbf{t}', \mathbf{e})}{p(\mathbf{t})/p(\mathbf{t}')} \right)$. As an input of BERT, the observed data is considered as pairs of $(\mathbf{t}, \mathbf{e}) \in \mathcal{T} \times \mathcal{U}$. Thus, the likelihood is written in terms of $p(\mathbf{t}, \mathbf{e})$, not $p(\mathbf{t})$. Therefore, the Euclidean distance between the representation vectors is an upper bound on the log-likelihood ratio with respect to the observed dataset, $\log p(\mathbf{t}, \mathbf{e})/p(\mathbf{t}', \mathbf{e})$, when compared to the marginalized log-likelihood ratio, $\log p(\mathbf{t})/p(\mathbf{t}')$. Theorem 1 provides a partially theoretical justification for defining neighbors based on the Euclidean distance between `[CLS]` tokens in a pre-trained BERT model.

### 3.2 PHASE1: CONTEXTUAL REPRESENTATION

To bridge an inherent gap between the structured nature of tabular data and the unstructured form of text data, we employ a textual encoding approach similar to Yin et al. (2020); Mei et al. (2021);

Borisov et al. (2023). By leveraging the encoding process of Definition 3, we obtain the textual dataset denoted as $\{\mathbf{t}^{(i)}\}_{i=1}^{n}$.

**Definition 3** (Textual Encoding). *Each observation $\mathbf{x}^{(i)}$ is transformed into a sentence-like format $\mathbf{t}^{(i)}$. This transformation incorporates both the column names and their corresponding values, applying the following subject-predicate-object transformation:*

$$
\mathbf{t}_j^{(i)} = \begin{cases} \text{``$C_j$ is $\mathbf{x}_j^{(i)}$,''}, & \text{if } \mathbf{m}_j^{(i)} = 1 \\ \text{``$C_j$ is \texttt{[MASK]},''}, & \text{if } \mathbf{m}_j^{(i)} = 0 \end{cases},
$$

$$
\mathbf{t}^{(i)} = \text{``\texttt{[CLS]}''} \& \mathbf{t}_1^{(i)} \& \mathbf{t}_2^{(i)} \& \cdots \& \mathbf{t}_p^{(i)} \& \text{``\texttt{[SEP]}''},
$$

*where &, called the ampersand operator, is used to concatenate one or more text strings to form a single text, and $\texttt{[CLS]}$ and $\texttt{[SEP]}$ token denote the start and end of the sequence, respectively.*

Then, as in Definition 1, we obtain the representation vector corresponding to the observation $\mathbf{x}^{(i)}$, denoted as $\mathbf{z}^{(i)}$, as follows:

$$
\mathbf{z}^{(i)} = g(\mathbf{t}^{(i)}, \mathbf{e}; \theta) \in \mathbb{R}^D,
$$

where $D$ denotes the dimension of the representation vector. The transformation process of contextual representation from $\mathbf{x}^{(i)}$ to $\mathbf{z}^{(i)}$ offers several advantages as follows:

**1. Handling missing values with contextual information.** Since we employ BERT, which is a MLM, we can represent missing values with $\texttt{[MASK]}$ tokens. This approach enables BERT to represent missing entries by leveraging contextual hints from other columns. It allows for the generation of a complete contextual representation of each observation without the need to disregard missing entries.

**2. Missing-invariant mapping to continuous space.** The language model $g(\cdot, \cdot; \theta)$ consistently outputs a $D$-dimensional real-valued vector for each observation, regardless of the various missingness patterns. By mapping to continuous representation space, we can utilize well-defined distances, such as the $L_2$ norm, between different observations.

**3. Addressing heterogeneous tabular datasets.** In the tabular domain, a central challenge lies in heterogeneity, as tabular data includes continuous, categorical, and even date (time) columns. Huang et al. (2020); Somepalli et al. (2021) address this challenge by processing continuous and categorical variables differently to obtain their representations. In contrast, we employ textual encoding to convert each observation into a sentence-like format, deriving representations from this sentence. This provides a type-agnostic preprocessing of the heterogeneous tabular dataset.

### 3.3 PHASE2: CONTEXT-DRIVEN IMPUTATION

Our proposed imputation procedure is based on the nearest neighbor set within the continuous representation space. The top-$k$ nearest neighborhood on the representation space is defined as follows.

**Definition 4.** *The top-k nearest neighbors of $\mathbf{x}^{(i)}, i \in \{1, \cdots, n\}$ with respect to the $j$th column are defined by the index set $\mathcal{N}_j(\mathbf{x}^{(i)}, k) \subset \{1, 2, \cdots, n\}$, satisfying the following conditions:*

1. $|\mathcal{N}_j(\mathbf{x}^{(i)}, k)| = k$,

2. $\mathbf{m}_j^{(l)} = 1$ *for all $l \in \mathcal{N}_j(\mathbf{x}^{(i)}, k)$, and*

3. $\forall s \in \{1, \cdots, n\} \backslash \mathcal{N}_j(\mathbf{x}^{(i)}, k)$ *such that $\mathbf{m}_j^{(s)} = 1$*

$$
cos(\mathbf{z}^{(i)}, \mathbf{z}^{(s)}) < \min_{l \in \mathcal{N}_j(\mathbf{x}^{(i)}, k)} cos(\mathbf{z}^{(i)}, \mathbf{z}^{(l)}),
$$

*where $cos(\cdot, \cdot)$ denotes the cosine similarity.*

Note that the neighbor set includes only the indices of observations that are not missing in the dataset with respect to the $j$th column.

Our imputation procedure utilizes the neighbor set $\mathcal{N}_j(\mathbf{x}^{(i)}, k)$. For continuous columns, we impute using the average of the neighbor set. For categorical columns, we impute using the most frequent category among the neighbor set (see Algorithm 1 in Appendix for detailed procedure).

$$
\hat{\mathbf{x}}_j^{(i)} \;=\; \begin{cases} \dfrac{1}{k}\sum_{l \in \mathcal{N}_j(\mathbf{x}^{(i)},k)} \mathbf{x}_j^{(l)}, & \text{if } j \in I_C \\[2mm] \text{mode}\Big(\mathbf{x}_j^{(l)} : l \in \mathcal{N}_j(\mathbf{x}^{(i)},k)\Big), & \text{if } j \in I_D \end{cases}.
$$

Then, we obtain completed observation $\tilde{\mathbf{x}}$ as $\tilde{\mathbf{x}} = \mathbf{x} \odot \mathbf{m} + \hat{\mathbf{x}} \odot (\mathbf{1} - \mathbf{m})$, where $\odot$ denotes element-wise multiplication.

### 3.4 FINE-TUNING: CONTRASTIVE LEARNING

The population version of the objective function for fine-tuning is defined as follows (Ruderman et al., 2012; Belghazi et al., 2018):

$$
\max_{\theta} J_{CL}(\theta)
$$
$$
= \;\; \mathbb{E}_{q(\mathbf{t},\mathbf{u},\mathbf{e})}\Big[\cos\big(g(\mathbf{t},\mathbf{e};\theta), g(\mathbf{u},\mathbf{e};\theta)\big)\Big] - \log \mathbb{E}_{p(\mathbf{t},\mathbf{e})p(\mathbf{u},\mathbf{e})}\Big[\exp\big(\cos\big(g(\mathbf{t},\mathbf{e};\theta), g(\mathbf{u},\mathbf{e};\theta)\big)\big)\Big],
$$

where $q(\mathbf{t}, \mathbf{u}, \mathbf{e})$ is the distribution of the positive sequence pair, and $p(\mathbf{t}, \mathbf{e})p(\mathbf{u}, \mathbf{e})$ is the distribution of the negative sequence pair. $q(\mathbf{t}, \mathbf{e}, \mathbf{u}) = q(\mathbf{u} \mid \mathbf{t}) \cdot p(\mathbf{t}, \mathbf{e})$ and the generating process of $\mathbf{u}$, i.e., $q(\mathbf{u} \mid \mathbf{t})$ is defined in Definition 5.

**Definition 5** (Re-mask). *Let $\phi$ be the re-masking function, which takes an anchor sample and a parameter $r$, specifying the number of columns to be masked. It randomly selects and masks $r$ columns from the anchor sample, $\mathbf{t}$, to produce a positive sample, $\mathbf{u}$. Formally, the generating process of $\mathbf{u}$ is written as:*

$$
\begin{aligned} (\mathbf{t}, \mathbf{e}) &\sim p(\mathbf{t}, \mathbf{e}) \\ \mathbf{u} &= \phi(\mathbf{t}, r). \end{aligned}
$$

The dropout augmentation strategy of Definition 5 is widely adopted to generate positive samples due to its simplicity and effectiveness (Chen et al., 2020; Miao et al., 2021; Wu et al., 2022; Jiang & Wang, 2022). For instance, if $r = 1$, we can obtain a positive sample of $\mathbf{t}^{(i)}$, which is denoted as $\mathbf{t}^{(i+)}$, by randomly selecting and masking one column from $\mathbf{t}^{(i)}$ as follows:

$$
\begin{aligned} \mathbf{t}^{(i)} &= \text{``[CLS]}, C_1 \text{ is } \mathbf{x}_1, C_2 \text{ is [MASK]}, C_3 \text{ is } \mathbf{x}_3, C_4 \text{ is } \mathbf{x}_4, \text{[SEP]''} \\ \mathbf{t}^{(i+)} &= \text{``[CLS]}, C_1 \text{ is } \mathbf{x}_1, C_2 \text{ is [MASK]}, C_3 \text{ is } \mathbf{x}_3, C_4 \text{ is [MASK]}, \text{[SEP]''}, \end{aligned}
$$

where $\mathbf{t}^{(i+)}$ is sampled from $q(\mathbf{u} \mid \mathbf{t}^{(i)})$. In this case, the value of the $C_4$ column is re-masked.

**Theorem 2.** *For $\theta$ such that the maximizer of $J_{CL}(\theta)$, $g(\mathbf{t}, \mathbf{e}; \theta)$ and $g(\mathbf{u}, \mathbf{e}; \theta)$ provide the representation vectors that maximize the lower bound of the mutual information between $\mathbf{t}$ and $\mathbf{u}$, where $(\mathbf{t}, \mathbf{u}) \sim q(\mathbf{u} \mid \mathbf{t}) \cdot p(\mathbf{t}, \mathbf{e})$.*

Based on the InfoNCE loss calculation in the batch, in practice, we approximate $J_{CL}(\theta)$ as follows (Oord et al., 2018):

$$
\max_{\theta} \mathcal{L}(\theta)
$$
$$
= \; \log \frac{\exp\Big(\cos\big(g(\mathbf{t}^{(i)},\mathbf{e};\theta), g(\mathbf{t}^{(i+)},\mathbf{e};\theta)\big)\Big)}{\exp\Big(\cos\big(g(\mathbf{t}^{(i)},\mathbf{e};\theta), g(\mathbf{t}^{(i+)},\mathbf{e};\theta)\big)\Big) + \sum_{l=1, l \neq i}^{B} \exp\Big(\cos\big(g(\mathbf{t}^{(i)},\mathbf{e};\theta), g(\mathbf{t}^{(l)},\mathbf{e};\theta)\big)\Big)},
$$

where $B$ is the mini-batch size, $\mathbf{t}^{(i+)}$ is the positive sample of the $i$th sample ($i \in \{1, \cdots, B\}$), and we define negative samples within a mini-batch (Tang et al., 2015; Mei et al., 2021). Note that, in the approximate of $J_{CL}(\theta)$ with $\mathcal{L}(\theta)$, $\mathbb{E}_{q(\mathbf{t},\mathbf{u},\mathbf{e})}$ is approximated with a single positive pair of $(\mathbf{t}^{(i)}, \mathbf{t}^{(i+)})$, and $\mathbb{E}_{p(\mathbf{t},\mathbf{e})p(\mathbf{u},\mathbf{e})}$ is approximated with the sequence pairs $(\mathbf{t}^{(i)}, \mathbf{t}^{(i+)}), (\mathbf{t}^{(i)}, \mathbf{t}^{(1)}), \cdots, (\mathbf{t}^{(i)}, \mathbf{t}^{(B)})$.

**Theorem 3.** *For $\theta$ such that the maximizer of $\mathcal{L}(\theta)$, $g(\mathbf{t}^{(i)}, \mathbf{e}; \theta)$ and $g(\mathbf{t}^{(i+)}, \mathbf{e}; \theta)$ provide the representation vectors that their cosine similarity is proportional to mutual information between $\mathbf{t}^{(i)}$ and $\mathbf{t}^{(i+)}$, where $\mathbf{t}^{(i+)} \sim q(\mathbf{u} \mid \mathbf{t}^{(i)})$.*

We want to emphasize that Theorem 2 and 3 demonstrate that fine-tuning using contrastive learning allows the representations of BERT to be trained in such a way that they represent and maximize the mutual information between input sequences. In other words, our fine-tuning further enhances the representational capacity of the language model, capturing statistical dependencies between observed and missing entries in the *unseen tabular data scenario*. **Therefore, through fine-tuning with contrastive learning, we can perform missing data imputation using the fully observed data points that have high mutual information with the observations containing missing entries.**

## 4 EXPERIMENTS

### 4.1 OVERVIEW

We propose two versions of our model, DrIM: 1) DrIM$_{\text{BASE}}$, which utilizes the pre-trained BERT directly, and 2) DrIM$_{\text{FINE}}$, which incorporates contrastive learning framework[3]. While any BERT model can be used, we employ the BERT-base model [4] to extract representation vectors.

**Datasets.** Similar to several recent studies (Mattei & Frellsen, 2019; Nazábal et al., 2020; Muzellec et al., 2020; Ipsen et al., 2021; Jarrett et al., 2022; Zhao et al., 2023), we utilize 10 real tabular datasets from the UCI repository[5] and Kaggle[6], which vary in sample sizes. The datasets are split into training and testing sets, with an 80% training and 20% testing ratio. Detailed statistics of these datasets are provided in Appendix A.4.

Additionally, following the approach in Muzellec et al. (2020); Jarrett et al. (2022); Zhao et al. (2023), we generate missing value masks for each dataset using three mechanisms (MCAR, MAR, MNAR) across four settings (MCAR, MAR, MNARL, MNARQ). Refer to Appendix A.7.1 for detailed descriptions and implementations of these mechanisms. We generate missing values at rates of 0.2, 0.3, 0.4, 0.6, and 0.8 for each mechanism.

**Baseline models.** In this experiment, we investigate our proposed method, DrIM, alongside 13 imputation methods, categorized into statistical imputation, ML-based imputation, and DL-based imputation as follows:

- Statistical imputation: Mean (Farhangfar et al., 2007), and kNNI (Troyanskaya et al., 2001)
- ML-based imputation: EM (Nelwamondo et al., 2007), SoftImpute (Mazumder et al., 2010), MICE (van Buuren & Groothuis-Oudshoorn, 2011), missForest (Stekhoven & Bühlmann, 2012), and Sinkhorn (Muzellec et al., 2020)
- DL-based imputation: GAIN (Yoon et al., 2018), VAEAC (Ivanov et al., 2019), MIWAE (Mattei & Frellsen, 2019), not-MIWAE (Ipsen et al., 2021), MIRACLE (Kyono et al., 2021), and ReMasker (Du et al., 2024a).

Detailed experimental settings for the reproducibility of these baseline models and our proposed model are provided in Appendix A.5.

**Evaluation Metrics.** To assess the imputation performance, we use two types of evaluation criteria: 1) imputation fidelity and 2) machine learning utility. The performance of imputers is reported by averaging the metrics across 10 real tabular datasets. Imputation fidelity is measured by the sum of the SMAPE (symmetric mean absolute percentage error) for continuous columns and the AR (accuracy error) for categorical columns, which we refer to as ARSMAPE, similar to (Miao et al., 2022). For machine learning utility, we adopt the metrics proposed by Hansen et al. (2023) to

---

[3]We conduct experiments using an NVIDIA A10 GPU, with our experimental codes implemented in `PyTorch` and `scikit-learn`.

[4]https://huggingface.co/google-bert/bert-base-uncased

[5]https://archive.ics.uci.edu/

[6]https://www.kaggle.com/datasets/

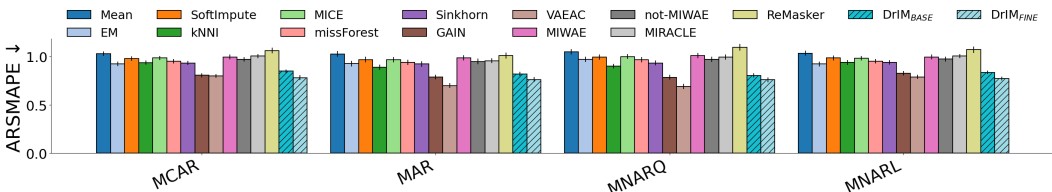

Figure 2: **Imputation fidelity** at 0.3 missingness rate. The missing mechanism corresponding to the figure is indicated below the figure. The means and the standard errors of the mean across 10 datasets and 10 repeated experiments are reported.

Table 1: **Machine learning utility** at 0.3 missingness under MAR and MNARL. The means and the standard errors of the mean across 10 datasets and 10 repeated experiments are reported. ↑ denotes higher is better. The best value is bolded, and the second best is underlined.

| model | MAR | | | MNARL | | |
|---|---|---|---|---|---|---|
| | $F_1$ ↑ | Model ↑ | Feature ↑ | $F_1$ ↑ | Model ↑ | Feature ↑ |
| Mean | $.591_{\pm.027}$ | $.548_{\pm.041}$ | $.744_{\pm.021}$ | $.560_{\pm.027}$ | $.428_{\pm.041}$ | $.797_{\pm.017}$ |
| kNNI | $.601_{\pm.027}$ | $.560_{\pm.043}$ | $.868_{\pm.014}$ | $.575_{\pm.027}$ | $.419_{\pm.049}$ | $.843_{\pm.016}$ |
| EM | $.598_{\pm.027}$ | $.531_{\pm.043}$ | $.833_{\pm.016}$ | $.580_{\pm.027}$ | $.407_{\pm.048}$ | $.804_{\pm.019}$ |
| SoftImpute | $.590_{\pm.027}$ | $.554_{\pm.039}$ | $.845_{\pm.014}$ | $.562_{\pm.027}$ | $.342_{\pm.045}$ | $.818_{\pm.015}$ |
| MICE | $.598_{\pm.027}$ | $.543_{\pm.043}$ | $.859_{\pm.014}$ | $.571_{\pm.027}$ | $.339_{\pm.050}$ | $.825_{\pm.017}$ |
| missForest | $.596_{\pm.027}$ | $.548_{\pm.044}$ | $.833_{\pm.015}$ | $.569_{\pm.027}$ | $.366_{\pm.048}$ | $.809_{\pm.016}$ |
| Sinkhorn | $.599_{\pm.027}$ | $.545_{\pm.044}$ | $.845_{\pm.015}$ | $.569_{\pm.027}$ | $.377_{\pm.049}$ | $.821_{\pm.017}$ |
| GAIN | $\underline{.640}_{\pm.025}$ | $.602_{\pm.029}$ | $.877_{\pm.011}$ | $.621_{\pm.025}$ | $.446_{\pm.040}$ | $.843_{\pm.016}$ |
| VAEAC | $.606_{\pm.025}$ | $.612_{\pm.018}$ | $.827_{\pm.007}$ | $.598_{\pm.025}$ | $\mathbf{.553}_{\pm.024}$ | $.816_{\pm.012}$ |
| MIWAE | $.588_{\pm.027}$ | $.519_{\pm.048}$ | $.834_{\pm.014}$ | $.556_{\pm.028}$ | $.369_{\pm.053}$ | $.821_{\pm.016}$ |
| not-MIWAE | $.587_{\pm.027}$ | $.540_{\pm.043}$ | $.839_{\pm.013}$ | $.558_{\pm.027}$ | $.394_{\pm.049}$ | $.818_{\pm.017}$ |
| MIRACLE | $.597_{\pm.027}$ | $.473_{\pm.046}$ | $.851_{\pm.014}$ | $.564_{\pm.027}$ | $.264_{\pm.051}$ | $.825_{\pm.016}$ |
| ReMasker | $.592_{\pm.027}$ | $.480_{\pm.046}$ | $.817_{\pm.019}$ | $.566_{\pm.028}$ | $.279_{\pm.050}$ | $.782_{\pm.021}$ |
| DrIM$_{BASE}$ | $\underline{.640}_{\pm.024}$ | $\underline{.617}_{\pm.041}$ | $\underline{.894}_{\pm.010}$ | $\underline{.625}_{\pm.024}$ | $\underline{.496}_{\pm.043}$ | $\underline{.886}_{\pm.009}$ |
| DrIM$_{FINE}$ | $\mathbf{.659}_{\pm.025}$ | $\mathbf{.658}_{\pm.032}$ | $\mathbf{.905}_{\pm.007}$ | $\mathbf{.653}_{\pm.025}$ | $\mathbf{.553}_{\pm.040}$ | $\mathbf{.915}_{\pm.006}$ |

evaluate the performance of post-imputation prediction. Specifically, we use three metrics detailed in Hansen et al. (2023): classification performance ($F_1$ score), model selection performance (Model), and feature selection performance (Feature). For a comprehensive evaluation procedure, please refer to Appendix A.6.

### 4.2 RESULTS

**Imputation fidelity.** Figure 2 shows the imputation fidelity of DrIM and baseline models at a 0.3 missingness rate. DrIM consistently demonstrates competitive performance across the four missingness mechanisms (MCAR, MAR, MNARQ, and MNARL), with consistently low ARSMAPE scores. This indicates that our model is robust to different missing data mechanisms, which are often encountered in real-world datasets. Notably, DrIM$_{FINE}$ benefits from the contrastive learning strategy, as shown by its superior performance compared to DrIM$_{BASE}$. This improvement suggests that the contrastive learning approach enhances DrIM's ability to identify relevant nearest neighbors, resulting in more accurate imputation. These enhanced imputations are crucial for post-imputation tasks, such as MLu, that depend on the quality of the imputed data.

**Machine learning utility.** We evaluate the post-imputation performance of DrIM and baseline models in terms of MLu. As shown in Table 1, DrIM consistently achieves the highest metric scores in $F_1$ score, model selection, and feature selection. Notably, DrIM$_{BASE}$ outperforms other baselines, such as GAIN and VAEAC, which require training, even though these models achieve similar performance to DrIM. This highlights the superior performance of DrIM$_{BASE}$ without the need for training. This suggests that our approach is cost-effective, providing strong performance without the need for extensive training. Additionally, we confirmed that kNNI demonstrates competitive performance compared to other baseline models. Given that our method outperforms kNNI, this empirically vali-

Table 2: **Effect of constrastive learning** under MAR. The means and the standard errors of the mean across 10 datasets and 10 repeated experiments are reported. ↑ denotes higher is better. Values in parentheses indicate the performance difference compared to $\text{DrIM}_{\text{BASE}}$, and the red highlights the positive improvement.

| Rate | model | $F_1$ ↑ | Model ↑ | Feature ↑ |
|------|-------|---------|---------|-----------|
| 0.2 | $\text{DrIM}_{\text{BASE}}$ | $.673_{\pm.024}$ | $.732_{\pm.029}$ | $.934_{\pm.006}$ |
|      | $\text{DrIM}_{\text{FINE}}$ | $.684_{\pm.025}$ (+1.64%) | $.773_{\pm.023}$ (+5.59%) | $.948_{\pm.004}$ (+1.50%) |
| 0.4 | $\text{DrIM}_{\text{BASE}}$ | $.607_{\pm.024}$ | $.452_{\pm.044}$ | $.837_{\pm.015}$ |
|      | $\text{DrIM}_{\text{FINE}}$ | $.639_{\pm.025}$ (+5.27%) | $.556_{\pm.035}$ (+23.03%) | $.853_{\pm.012}$ (+1.91%) |
| 0.6 | $\text{DrIM}_{\text{BASE}}$ | $.545_{\pm.024}$ | $.231_{\pm.052}$ | $.684_{\pm.024}$ |
|      | $\text{DrIM}_{\text{FINE}}$ | $.589_{\pm.024}$ (+8.08%) | $.373_{\pm.040}$ (+61.47%) | $.734_{\pm.026}$ (+7.32%) |
| 0.8 | $\text{DrIM}_{\text{BASE}}$ | $.477_{\pm.023}$ | $.072_{\pm.054}$ | $.446_{\pm.041}$ |
|      | $\text{DrIM}_{\text{FINE}}$ | $.534_{\pm.024}$ (+11.96%) | $.257_{\pm.047}$ (+257.64%) | $.604_{\pm.035}$ (+35.62%) |

dates that the distance between representations of BERT is effective. In Appendix A.9.2, we provide more comprehensive results under different missingness mechanisms and missingness rates. Additionally, you can find the MLu performance for each dataset.

**Effect of contrastive learning.** The results presented in Table 2 indicate that incorporating the contrastive learning framework into our proposed method improves performance. As shown in Table 1 and Table 2, $\text{DrIM}_{\text{FINE}}$ outperforms $\text{DrIM}_{\text{BASE}}$, supporting our claim in Section 3.4 that fine-tuning with contrastive learning allows the representations of BERT to be trained in a way that performs metric learning to maximize the mutual information between input sequences. Furthermore, we observe performance improvements not only in MAR but also in MNARL, further validating the effectiveness of our approach. Additional results for different missing data mechanisms (MCAR, MNARL, MNARQ) can be found in Appendix A.9.4.

**Sensitivity analysis.** To more comprehensively evaluate the post-imputation performance of DrIM, we also conducted a sensitivity analysis by varying the missingness rate of the datasets. Figure 3 shows that although the performance of each model decreases as the missingness rate increases, both $\text{DrIM}_{\text{BASE}}$ and $\text{DrIM}_{\text{FINE}}$ exhibit stable performance compared to the baseline models. This reflects the effectiveness of leveraging the language model's representation learning capabilities, even without fine-tuning. Notably, $\text{DrIM}_{\text{FINE}}$ consistently outperforms the baseline models in terms of $F_1$ score and feature selection performance. Although the model selection performance of DrIM crosses with VAEAC as the missingness rate changes from 0.4 to 0.6, $\text{DrIM}_{\text{FINE}}$ still demonstrates superior performance compared to other baseline models. This indicates that fine-tuning with contrastive learning continues to be effective even at higher missingness rates. Additional results for different missing data mechanisms can be found in Appendix A.9.5.

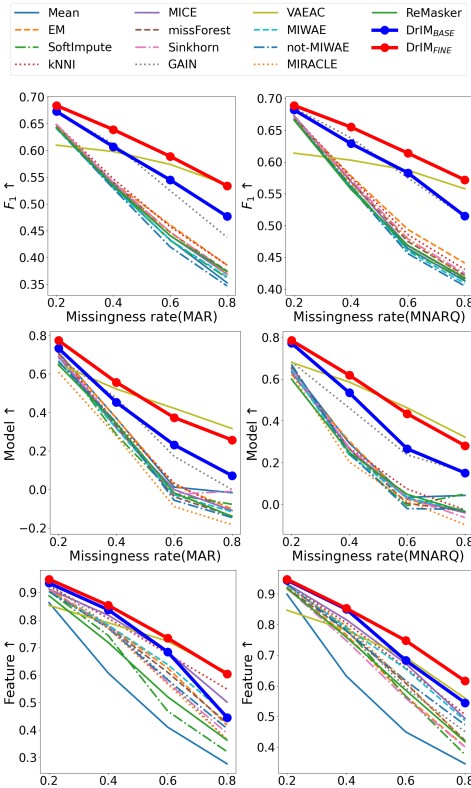

Figure 3: **Sensitivity analysis for missingness rates.** The results MAR and MNARQ are shown, with the first-row representing classification performance ($F_1$ score) and the last two rows displaying model selection and feature selection performance. The means of the average across 10 datasets and 10 repeated experiments are reported.

## 5 CONCLUSIONS

This paper introduces DrIM, a context-driven nearest-neighbor-based imputation method specifically designed to handle incomplete heterogeneous tabular datasets. Our proposed method leverages the representation learning capabilities of language models and integrates a contrastive learning framework for fine-tuning. In other words, in our proposed method, BERT performs dimensionality reduction on (numerical) tabular datasets in text format through the bidirectional attention mechanism, and we perform metric learning to maximize mutual information between observed and missing data entries. Our extensive experiments across 10 real-world datasets demonstrate that DrIM consistently outperforms existing imputation methods, achieving state-of-the-art post-imputation performance in terms of MLu across various missing data mechanisms, including MCAR, MAR, MNARL, and MNARQ.

In this paper we have not explicitly demonstrated how the neighbors using a BERT model for kNNI specifically improves downstream task performance. However, we believe our proposed method provides insights into the relationship between learned metrics obtained through contrastive learning and conditional independence among variables by emphasizing the enhancement of machine learning utility rather than statistical fidelity. This connection between contextual embeddings derived from BERT and downstream task performance may serve as a critical clue for further exploration, which we leave for future research.

### REPRODUCIBILITY STATEMENT

Detailed information about the datasets we used can be found in Appendix A.4. For the reproducibility of the baseline models and our proposed model, detailed experimental settings are provided in Appendix A.5. The reproduced codes for the baseline models and our proposed model are available in the supplementary material. For a comprehensive evaluation of MLu, please refer to Appendix A.6 for the detailed evaluation procedure and ML model configuration.

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

# A   APPENDIX

## A.1   PROOF OF THEOREM 1

*Proof.* Sugiyama et al. (2012); Sasaki & Takenouchi (2022) showed that the minimizer of $J_{LR}(\theta, \eta)$ is given by

$$r(\mathbf{t}, \mathbf{u}; \theta, \eta) = \log \frac{p(\mathbf{t}, \mathbf{u})}{p(\mathbf{t})p(\mathbf{u})}.$$

Then, for the minimizer of $J_{LR}(\theta, \eta)$, we have

$$
\left| r(\mathbf{t}, \mathbf{e}; \theta, \eta) - r(\mathbf{t}', \mathbf{e}; \theta, \eta) \right| = \left| \log \frac{p(\mathbf{t}, \mathbf{e})}{p(\mathbf{t})p(\mathbf{e})} - \log \frac{p(\mathbf{t}', \mathbf{e})}{p(\mathbf{t}')p(\mathbf{e})} \right|
$$
$$
= \left| \log \frac{p(\mathbf{e} \mid \mathbf{t})}{p(\mathbf{e} \mid \mathbf{t}')} \right|,
$$

and

$$
\left| r(\mathbf{t}, \mathbf{e}; \theta, \eta) - r(\mathbf{t}', \mathbf{e}; \theta, \eta) \right| = \left| \Phi(g(\mathbf{t}, \mathbf{e}; \theta); \eta) - \Phi(g(\mathbf{t}', \mathbf{e}; \theta); \eta) \right|
$$
$$
= \left| \eta^\top \left( g(\mathbf{t}, \mathbf{e}; \theta) - g(\mathbf{t}', \mathbf{e}; \theta) \right) \right| \quad \text{(the scoring function is a linear map)}
$$
$$
\leq \|\eta\| \cdot \|g(\mathbf{t}, \mathbf{e}; \theta) - g(\mathbf{t}', \mathbf{e}; \theta)\|,
$$

where the last inequality follows from Cauchy-Schwarz inequality.

The proof is complete. □

## A.2   PROOF OF THEOREM 2

*Proof.* Ruderman et al. (2012); Belghazi et al. (2018) showed that

$$
I\big((\mathbf{t}, \mathbf{e}), (\mathbf{u}, \mathbf{e})\big) \geq J_{CL}(\theta)
$$
$$
= \mathbb{E}_{q(\mathbf{t}, \mathbf{u}, \mathbf{e})} \Big[ \cos\big(g(\mathbf{t}, \mathbf{e}; \theta), g(\mathbf{u}, \mathbf{e}; \theta)\big) \Big] - \log \mathbb{E}_{p(\mathbf{t}, \mathbf{e})p(\mathbf{u}, \mathbf{e})} \Big[ \exp\big(\cos\big(g(\mathbf{t}, \mathbf{e}; \theta), g(\mathbf{u}, \mathbf{e}; \theta)\big)\big) \Big],
$$

where $I\big((\mathbf{t}, \mathbf{e}), (\mathbf{u}, \mathbf{e})\big)$ is the mutual information between $\mathbf{t}$ and $\mathbf{u}$.

Therefore, if we obtain $\theta$ such that the maximizer of $J_{CL}(\theta)$, $g(\mathbf{t}, \mathbf{e}; \theta)$ and $g(\mathbf{u}, \mathbf{e}; \theta)$ provide the representation vectors that maximizes the lower bound of the mutual information between $\mathbf{t}$ and $\mathbf{u}$.

The proof is complete. □

## A.3   PROOF OF THEOREM 3

The proof is a restatement of the results from the paper by Oord et al. (2018).

*Proof.* $\mathcal{L}(\theta)$ is the categorical cross-entropy for classifying the positive sample correctly, with the following expression for the prediction from BERT:

$$
\frac{\exp\Big(\cos\big(g(\mathbf{t}^{(i)}, \mathbf{e}; \theta), g(\mathbf{t}^{(i+)}, \mathbf{e}; \theta)\big)\Big)}{\exp\Big(\cos\big(g(\mathbf{t}^{(i)}, \mathbf{e}; \theta), g(\mathbf{t}^{(i+)}, \mathbf{e}; \theta)\big)\Big) + \sum_{l=1, l \neq i}^{B} \exp\Big(\cos\big(g(\mathbf{t}^{(i)}, \mathbf{e}; \theta), g(\mathbf{t}^{(l)}, \mathbf{e}; \theta)\big)\Big)}.
$$

With respect to the $i$th sample, the optimal probability for $\mathcal{L}(\theta)$ is denoted as

$$
\Pr\big(i+ \text{ is the positive sample} \mid \mathbf{t}^{(l)}, l \in \{i+, 1, 2, \cdots, B\}\big).
$$

Note that the positive sample $\mathbf{t}^{(i+)}$ is drawn from the conditional distribution of $q(\mathbf{u} \mid \mathbf{t}^{(i)})$ and $\Pr(l$ is the positive sample$) = 1/B$ for all $l \in \{i+, 1, 2, \cdots, B\}$ and $l \neq i$. Then, we have

$$\Pr\left(i+ \text{ is the positive sample} \mid \mathbf{t}^{(l)}, l \in \{i+, 1, 2, \cdots, B\}\right)$$

$$= \frac{p\left(\mathbf{t}^{(l)}, l \in \{i+, 1, 2, \cdots, B\} \mid i+ \text{ is the positive sample}\right) \Pr(i+ \text{ is the positive sample})}{\sum_{j \in \{i+, 1, 2, \cdots, B\}, j \neq i} p\left(\mathbf{t}^{(l)}, l \in \{i+, 1, 2, \cdots, B\} \mid j \text{ is the positive sample}\right) \Pr(j \text{ is the positive sample})}$$

$$= \frac{q(\mathbf{t}^{(i+)} \mid \mathbf{t}^{(i)}) \prod_{l=1, l \neq i}^{B} p(\mathbf{t}^{(l)}, \mathbf{e})}{q(\mathbf{t}^{(i+)} \mid \mathbf{t}^{(i)}) \prod_{l=1, l \neq i}^{B} p(\mathbf{t}^{(l)}, \mathbf{e}) + \sum_{j=1, j \neq i}^{B} q(\mathbf{t}^{(j)} \mid \mathbf{t}^{(i)}) \prod_{l \in \{i+, 1, 2, \cdots, B\}, l \neq j, l \neq i} p(\mathbf{t}^{(l)}, \mathbf{e})}$$

$$= \frac{\dfrac{q(\mathbf{t}^{(i+)} \mid \mathbf{t}^{(i)})}{p(\mathbf{t}^{(i+)}, \mathbf{e})}}{\dfrac{q(\mathbf{t}^{(i+)} \mid \mathbf{t}^{(i)})}{p(\mathbf{t}^{(i+)}, \mathbf{e})} + \sum_{j=1, j \neq i}^{B} \dfrac{q(\mathbf{t}^{(j)} \mid \mathbf{t}^{(i)})}{p(\mathbf{t}^{(j)}, \mathbf{e})}}$$

$$= \frac{\dfrac{p(\mathbf{t}^{(i)}, \mathbf{t}^{(i+)}, \mathbf{e})}{p(\mathbf{t}^{(i)}, \mathbf{e}) p(\mathbf{t}^{(i+)}, \mathbf{e})}}{\dfrac{p(\mathbf{t}^{(i)}, \mathbf{t}^{(i+)}, \mathbf{e})}{p(\mathbf{t}^{(i)}, \mathbf{e}) p(\mathbf{t}^{(i+)}, \mathbf{e})} + \sum_{j=1, j \neq i}^{B} \dfrac{p(\mathbf{t}^{(i)}, \mathbf{t}^{(j)}, \mathbf{e})}{p(\mathbf{t}^{(i)}, \mathbf{e}) q(\mathbf{t}^{(j)}, \mathbf{e})}}.$$

Therefore, for the optimal value of the prediction from BERT with $\theta$ such that the maximizer of $\mathcal{L}(\theta)$,

$$\cos\left(g(\mathbf{t}^{(i)}, \mathbf{e}; \theta), g(\mathbf{t}^{(i+)}, \mathbf{e}; \theta)\right) \propto \log \frac{p(\mathbf{t}^{(i)}, \mathbf{t}^{(i+)}, \mathbf{e})}{p(\mathbf{t}^{(i)}, \mathbf{e}) p(\mathbf{t}^{(i+)}, \mathbf{e})} = \log \frac{p(\mathbf{t}^{(i)} \mid \mathbf{t}^{(i+)})}{p(\mathbf{t}^{(i)}, \mathbf{e})},$$

where the (unnormalized) density ratio is modeled as log-linear (Bachman et al., 2019), and $q(\mathbf{t}^{(i)}, \mathbf{t}^{(i+)})$, $q(\mathbf{t}^{(i)})$, and $q(\mathbf{t}^{(i+)})$ denote the joint and marginal distributions of $\mathbf{t}^{(i)}$ and $\mathbf{t}^{(i+)}$, respectively. $\qquad \square$

## A.4 DATASET DESCRIPTIONS

**Download links.**

- `abalone` (Nash et al., 1994):
  `https://archive.ics.uci.edu/dataset/1/abalone`
- `anuran` (Colonna et al., 2015):
  `https://archive.ics.uci.edu/dataset/406/anuran+calls+mfccs`
- `banknote` (Lohweg, 2012):
  `https://archive.ics.uci.edu/dataset/267/banknote+`
  `authentication`
- `breast` (Wolberg et al., 1993):
  `https://archive.ics.uci.edu/dataset/17/breast+cancer+`
  `wisconsin+diagnostic`
- `concrete` (Yeh, 1998):
  `https://archive.ics.uci.edu/dataset/165/concrete+`
  `compressive+strength`
- `kings` (CC0: Public Domain):
  `https://www.kaggle.com/datasets/harlfoxem/`
  `housesalesprediction`
- `letter` (Slate, 1991):
  `https://archive.ics.uci.edu/dataset/59/letter+recognition`
- `loan` (CC0: Public Domain):
  `https://www.kaggle.com/datasets/teertha/`
  `personal-loan-modeling`
- `redwine` (Cortez et al., 2009):
  `https://archive.ics.uci.edu/dataset/186/wine+quality`
- `whitewine` (Cortez et al., 2009):
  `https://archive.ics.uci.edu/dataset/186/wine+quality`

Table 3: **Description of datasets.** #C represents the number of continuous variables. #D denotes the number of categorical (discrete) variables. The 'Target' refers to the variable used as the response variable in a classification task to evaluate machine learning utility.

| Dataset | Split | #C | #D | Target |
|---------|-------|----|----|--------|
| abalone | 3.3K/0.8K | 7 | 2 | Rings |
| anuran | 5.7K/1.5K | 22 | 3 | Species |
| banknote | 1.1K/0.3K | 4 | 1 | class |
| breast | 0.5K/0.1K | 30 | 1 | Diagnosis |
| concrete | 0.8K/0.2K | 8 | 1 | Age |
| kings | 17.3K/4.3K | 11 | 7 | grade |
| letter | 16K/4K | 16 | 1 | lettr |
| loan | 4K/1K | 5 | 6 | Personal Loan |
| redwine | 1.3K/0.3K | 11 | 1 | quality |
| whitewine | 3.9K/1K | 11 | 1 | quality |

---

**Algorithm 1** Imputation procedure of DrIM

---

**Input**: Incomplete observation: $(\mathbf{x}, \mathbf{m})$, Columns name: $C = \{C_1, C_2, \cdots, C_p\}$,
      Pre-trained BERT: $g(\cdot, \cdot; \theta)$, A sequence of the [PAD] tokens: e
**Output**: Complete (imputed) observation: $\tilde{\mathbf{x}}$

---

    *Phase 1 – Contextual Representation*

---

 1: **for** $j = 1, 2, \cdots, p$ **do**
 2:    **if** $\mathbf{m}_j = 1$ **then**
 3:        $\mathbf{t}_j \leftarrow$ "$C_j$ is $\mathbf{x}_j$"
 4:    **end if**
 5:    **if** $\mathbf{m}_j = 0$ **then**
 6:        $\mathbf{t}_j \leftarrow$ "$C_j$ is [MASK]"              ▷ replacing missing value with [MASK] token
 7:    **end if**
 8: **end for**
 9: $\mathbf{t} \leftarrow$ "[CLS]" & $\mathbf{t}_1$ & $\mathbf{t}_2$ & $\cdots$ & $\mathbf{t}_p$ & "[SEP]"           ▷ Textual encoding
10: $\mathbf{z} \leftarrow g(\mathbf{t}, \mathbf{e}; \theta) \in \mathbb{R}^D$

---

    *Phase 2 – Context-Driven Imputation*

---

11: $\hat{\mathbf{x}} \leftarrow [0, 0, \cdots, 0]$
12: **for** $j = \{j : \mathbf{m}_j = 0, j = 1, 2, \cdots, p\}$ **do**
13:    $\mathcal{N}_j(\mathbf{x}, k) \subset \{1, 2, \cdots, n\}$          ▷ construct the neighbor index set by Definition 4
14:    **if** $j \in I_C$ **then**
15:        $\hat{\mathbf{x}}_j \leftarrow \frac{1}{k} \sum_{l \in \mathcal{N}_j(\mathbf{x}, k)} \mathbf{x}_j^{(l)}$
16:    **end if**
17:    **if** $j \in I_D$ **then**
18:        $\hat{\mathbf{x}}_j \leftarrow \text{mode}\left(\mathbf{x}_j^{(l)} : l \in \mathcal{N}_j(\mathbf{x}, k)\right)$
19:    **end if**
20: **end for**
21: $\tilde{\mathbf{x}} \leftarrow \mathbf{x} \odot \mathbf{m} + \hat{\mathbf{x}} \odot (1 - \mathbf{m})$

---

## A.5 EXPERIMENTAL SETTINGS FOR REPRODUCTION

- We run experiments using NVIDIA A10 GPU, and our experimental codes are available with `PyTorch` and `scikit-learn`.

**Hyper-parameter of DrIM**: This demonstrates the generalizability of our proposed model to various tabular datasets. Our implementation codes for the proposed model, DrIM, are provided in the supplementary material. For all tabular datasets, we applied the following hyperparameters uniformly without any additional tuning:

- batch size: 16
- $k$ (the number of neighbors): 5

For fine-tuning the BERT $g(\cdot, \cdot; \theta)$,

- epochs: 5 (with AdamW optimizer (Loshchilov & Hutter, 2017))
- learning rate: 5e-5
- $r$ (the number of re-masking): 3
- $\tau$: 1

### A.5.1 DETAILS OF IMPLEMENTING BASELINE MODELS

To compare our proposed method with baseline models, we performed experiments across four missing data mechanisms and five missingness rates. Our reproduced codes for the baseline models

are provided in the supplementary material. Below are the detailed implementations of the baseline methods:

- Mean (Farhangfar et al., 2007): We employ the `SimpleImputer` package [7], with the `strategy` parameter set to 'mean'.

- kNNI (Troyanskaya et al., 2001): We employ the `KNNImputer` package [8] from `scikit-learn` for missing data imputation.

- EM (Nelwamondo et al., 2007):
We employ the EM module from `hyperimpute.Plugins` [9].

- SoftImpute (Mazumder et al., 2010):
We employ the SoftImpute module in `hyperimpute.Plugins` [10].

- MICE (van Buuren & Groothuis-Oudshoorn, 2011): We employed the `IterativeImputer` package [11] from `scikit-learn`.

- missForest (Stekhoven & Bühlmann, 2012):
We employ the missForest module in `hyperimpute.Plugins` [12].

- Sinkhorn (Muzellec et al., 2020):
We employ the Sinkhorn module in `hyperimpute.Plugins` [13].

- GAIN (Yoon et al., 2018): We follow the implementations provided in the official repository[14]. As the paper does not explicitly discuss the separate handling of categorical and continuous variables, the code treats them simultaneously. Consequently, a rounding process is employed to handle categorical variables afterward.

- VAEAC (Ivanov et al., 2019): We follow the implementations provided in the official repository[15]. The authors provided hyperparameters that adequately address both continuous and categorical variables, so we used these without further modification during model fitting.

- MIWAE (Mattei & Frellsen, 2019): The implemented MIWAE code in the official repository[16] was designed for continuous variables only. To accommodate heterogeneous tabular datasets, we treated the conditional distribution of categorical columns as categorical distributions and employed cross-entropy loss for reconstruction.

- not-MIWAE (Ipsen et al., 2021): The implemented not-MIWAE code in the official repository[17] also focused on continuous variables exclusively. To handle categorical variables, we made the same modifications as in MIWAE.

- MIRACLE (Kyono et al., 2021):
We utilize the MIRACLE module in `hyperimpute.Plugins` [18].

- ReMasker (Du et al., 2024a): We follow the implementations provided in official repository [19].

---

[7]https://scikit-learn.org/1.5/modules/generated/sklearn.impute.SimpleImputer.html

[8]https://scikit-learn.org/stable/modules/generated/sklearn.impute.KNNImputer.html

[9]https://github.com/vanderschaarlab/hyperimpute/blob/main/src/hyperimpute/plugins/imputers/plugin_EM.py

[10]https://github.com/vanderschaarlab/hyperimpute/blob/main/src/hyperimpute/plugins/imputers/plugin_softimpute.py

[11]https://scikit-learn.org/stable/modules/generated/sklearn.impute.IterativeImputer.html

[12]https://github.com/vanderschaarlab/hyperimpute/blob/main/src/hyperimpute/plugins/imputers/plugin_missforest.py

[13]https://github.com/vanderschaarlab/hyperimpute/blob/main/src/hyperimpute/plugins/imputers/plugin_sinkhorn.py

[14]https://github.com/jsyoon0823/GAIN/tree/master

[15]https://github.com/tigvarts/vaeac

[16]https://github.com/pamattei/miwae

[17]https://github.com/nbip/notMIWAE

[18]https://github.com/vanderschaarlab/hyperimpute/blob/main/src/hyperimpute/plugins/imputers/plugin_miracle.py

[19]https://github.com/tydusky/remasker

## A.6 EVALUATION SETTINGS

Regarding machine learning utility, we conduct classification, model selection, and feature selection tasks for post-imputation evaluation (Please refer to Table 3 for the classification target variable, and Table 4 for detailed machine learning model configuration).

**Classification performance ($F_1$).**

1. Train an imputer using a given incomplete training dataset.

2. Impute the incomplete training dataset using trained imputer.

3. Train machine learning models (Logistic Regression (Cox, 1958), Gaussian Naive Bayes, k-nearest neighbors classifier (Altman, 1992), Decision Tree classifier (Wu et al., 2008), and Random Forest classifier (Ho, 1998; Breiman, 2001)) using the imputed dataset.

4. Assess classification prediction performance by averaging the $F_1$ scores from the complete test dataset from five different classifiers.

**Model selection performance (Model).**

1. Train an imputer using a given incomplete training dataset.

2. Impute the incomplete training dataset using trained imputer.

3. Train machine learning models (Logistic Regression (Cox, 1958), Gaussian Naive Bayes, k-nearest neighbors classifier (Altman, 1992), Decision Tree classifier (Wu et al., 2008), and Random Forest classifier (Ho, 1998; Breiman, 2001) using both the real complete training dataset and the imputed dataset.

4. Evaluate the classification performance (AUROC) of all trained classifiers on the complete test dataset.

5. Assess model selection performance by comparing the AUROC rank orderings of classifiers trained on the real (original) complete training dataset and those trained on the imputed dataset using Spearman's Rank Correlation.

**Feature selection performance (Feature).**

1. Train an imputer using a given incomplete training dataset.

2. Impute the incomplete training dataset using trained imputer.

3. Train a Random Forest classifier (Ho, 1998; Breiman, 2001) using both the real (original) complete training dataset and the imputed dataset.

4. Determine the rank-ordering of important features for both classifiers.

5. Assess feature selection performance by comparing the feature importance rank orderings of classifiers trained on the real complete training dataset and those trained on the imputed dataset using Spearman's Rank Correlation.

Table 4: **Classifier used to evaluate imputed data quality in machine learning utility.** The names of all parameters used in the description are consistent with those defined in corresponding packages.

| Tasks | Model | Description |
|---|---|---|
| | Logistic Regression | Package: `sklearn.linear_model.LogisticRegression`, setting: random_state=0, max_iter=1000, defaulted values |
| | Gaussian Naive Bayes | Package: `sklearn.naive_bayes.GaussianNB`, setting: defaulted values |
| Classification | k-Nearest Neighbors | Package: `sklearn.neighbors.KNeighborsClassifier`, setting: defaulted values |
| | Decision Tree | Package: `sklearn.tree.DecisionTreeClassifier`, setting: random_state=0, defaulted values |
| | Random Forest | Package: `sklearn.ensemble.RandomForestClassifier`, setting: random_state=0, defaulted values |

## A.7 RELATED WORKS

### A.7.1 MISSING MECHANISM

**Descriptions.** Data missingness is a common challenge in research and practical analysis, categorized into three primary missing mechanisms: 1) Missing Completely at Random (MCAR), 2) Missing at Random (MAR), and 3) Missing Not at Random (MNAR).

Under the **MCAR** mechanism, the reason for missingness has no relationship with any data, neither observed nor unobserved. In other words, the likelihood of data being missing is equal across all observations. The primary advantage of MCAR is that it does not introduce bias into the data analysis. However, despite this advantage, data missingness can still reduce the statistical power of the study because of the reduced sample size.

**MAR** occurs when the probability of missingness is related to the observed data but not the unobserved missing data. Essentially, even though the data is missing, the mechanism assumes that the missingness is explainable by other variables in the dataset. That is, the missingness can be modeled and imputed using the information available in the data, allowing for more accurate analyses despite the missingness.

Lastly, if the missingness is not specified by either MCAR or MAR, it becomes **MNAR**. The MNAR is the most challenging mechanism, as it implies that the missingness is related to the unobserved data itself. In this case, the missing data is systematically different from the observed data, which introduces bias if not properly accounted for. For example, patients with severe symptoms may be less likely to report their health status, making their data missing. MNAR requires sophisticated statistical methods to address, as ignoring or improperly handling it can lead to biased and unreliable results.

**Implementations.** Following Muzellec et al. (2020); Jarrett et al. (2022); Zhao et al. (2023), we generate the missing value mask for each dataset with three mechanisms in four settings. (MCAR) In the MCAR setting, each value is masked according to the realization of a Bernoulli random variable with a fixed parameter. (MAR) In the MAR setting, for each experiment, a fixed subset of variables that cannot have missing values is sampled. Then, the remaining variables have missing values according to a logistic model with random weights, which takes the non-missing variables as inputs. A bias term is fitted using line search to attain the desired proportion of missing values. (MNAR) Finally, two different mechanisms are implemented in the MNAR setting. The first, MNARL, is identical to the previously described MAR mechanism, but the inputs of the logistic model are then masked by an MCAR mechanism. Hence, the logistic model's outcome depends on missing values. The second mechanism, MNARQ, samples a subset of variables whose values in the lower and upper $p$th percentiles are masked according to a Bernoulli random variable, and the values in-between are left not missing.

## A.8 ADDITIONAL EXPERIMENTS

We conduct extensive additional experiments using a variety of pre-trained language models, including both `Autoencoding` (BERT, RoBERTa) and `Autoregressive` (GPT-2, GPT-NEO, LLaMA) models, across different parameter scales.

### A.8.1 CONFIGURATION OF LANGUAGE MODELS

Table 5: The language model used to generate contextualized representations. '#Params' represents the number of parameters as defined in the respective language model specifications.

| Type | model | #Params |
|---|---|---|
| Autoencoding | BERT$_{BASE}$ (Devlin et al., 2019) | 0.11B |
| | RoBERTa (Liu et al., 2019) | 0.12B |
| | BERT$_{LARGE}$ (Devlin et al., 2019) | 0.34B |
| Autoregressive | GPT-2 (Radford et al., 2019) | 1.56B |
| | GPT-NEO (Black et al., 2021) | 1.32B |
| | LLaMA (Touvron et al., 2023) | 6.61B |

### A.8.2 EXPERIMENT RESULTS

Table 6: **Machine learning utility** at 0.3 missingness rate. Results using a pretrained language model without fine-tuning. ↑ denotes that higher values are better. The means and standard errors of the mean across 10 datasets and 10 repeated experiments are reported.

| Type | model | MCAR | | | MAR | | |
|---|---|---|---|---|---|---|---|
| | | $F_1$ ↑ | Model ↑ | Feature ↑ | $F_1$ ↑ | Model ↑ | Feature ↑ |
| Autoencoding | BERT$_{BASE}$ | $.638_{\pm.024}$ | $.517_{\pm.037}$ | $.920_{\pm.006}$ | $.640_{\pm.024}$ | $.617_{\pm.041}$ | $.894_{\pm.010}$ |
| | RoBERTa | $.652_{\pm.024}$ | $.538_{\pm.038}$ | $.922_{\pm.006}$ | $.652_{\pm.024}$ | $.628_{\pm.037}$ | $.907_{\pm.007}$ |
| | BERT$_{LARGE}$ | $.647_{\pm.025}$ | $.625_{\pm.031}$ | $.898_{\pm.008}$ | $.652_{\pm.025}$ | $.641_{\pm.030}$ | $.909_{\pm.007}$ |
| Autoregressive | GPT-2 | $.658_{\pm.024}$ | $.620_{\pm.032}$ | $.908_{\pm.008}$ | $.665_{\pm.025}$ | $.685_{\pm.032}$ | $.907_{\pm.007}$ |
| | GPT-NEO | $.659_{\pm.025}$ | $.606_{\pm.032}$ | $.922_{\pm.006}$ | $.665_{\pm.025}$ | $.688_{\pm.033}$ | $.916_{\pm.006}$ |
| | LLaMA | $.657_{\pm.024}$ | $.659_{\pm.030}$ | $.911_{\pm.009}$ | $.664_{\pm.024}$ | $.737_{\pm.026}$ | $.890_{\pm.010}$ |

| Type | model | MNARL | | | MNARQ | | |
|---|---|---|---|---|---|---|---|
| | | $F_1$ ↑ | Model ↑ | Feature ↑ | $F_1$ ↑ | Model ↑ | Feature ↑ |
| Autoencoding | BERT$_{BASE}$ | $.625_{\pm.024}$ | $.496_{\pm.043}$ | $.886_{\pm.009}$ | $.658_{\pm.025}$ | $.694_{\pm.029}$ | $.913_{\pm.007}$ |
| | RoBERTa | $.641_{\pm.023}$ | $.561_{\pm.042}$ | $.901_{\pm.008}$ | $.672_{\pm.024}$ | $.673_{\pm.028}$ | $.918_{\pm.006}$ |
| | BERT$_{LARGE}$ | $.635_{\pm.025}$ | $.596_{\pm.033}$ | $.886_{\pm.009}$ | $.670_{\pm.025}$ | $.713_{\pm.025}$ | $.911_{\pm.007}$ |
| Autoregressive | GPT-2 | $.655_{\pm.024}$ | $.607_{\pm.036}$ | $.906_{\pm.007}$ | $.677_{\pm.025}$ | $.741_{\pm.024}$ | $.921_{\pm.006}$ |
| | GPT-NEO | $.653_{\pm.024}$ | $.640_{\pm.032}$ | $.908_{\pm.006}$ | $.675_{\pm.025}$ | $.710_{\pm.026}$ | $.920_{\pm.005}$ |
| | LLaMA | $.653_{\pm.024}$ | $.666_{\pm.032}$ | $.900_{\pm.008}$ | $.676_{\pm.024}$ | $.726_{\pm.025}$ | $.886_{\pm.012}$ |

### A.8.3 TIME EFFICIENCY

Table 7: **Time Efficiency**. The reported unit is second(s). Results using a pretrained language model without fine-tuning. The means and standard errors of the mean across 10 datasets and 10 repeated experiments are reported.

| Time (s) | model | MCAR | MAR | MNARL | MNARQ |
|---|---|---|---|---|---|
| Autoencoding | BERT$_{BASE}$ | $93.240_{\pm9.887}$ | $86.160_{\pm9.089}$ | $94.030_{\pm9.767}$ | $97.860_{\pm10.399}$ |
| | RoBERTa | $92.780_{\pm9.814}$ | $84.190_{\pm8.809}$ | $96.020_{\pm10.274}$ | $94.770_{\pm10.136}$ |
| | BERT$_{LARGE}$ | $122.380_{\pm13.025}$ | $113.220_{\pm12.186}$ | $127.010_{\pm13.699}$ | $128.810_{\pm14.126}$ |
| Autoregressive | GPT-2 | $292.780_{\pm32.322}$ | $272.340_{\pm30.304}$ | $298.500_{\pm33.329}$ | $284.930_{\pm31.403}$ |
| | GPT-NEO | $237.670_{\pm25.515}$ | $216.765_{\pm24.932}$ | $233.190_{\pm24.546}$ | $235.620_{\pm25.319}$ |
| | LLaMA | $1196.240_{\pm132.644}$ | $1276.640_{\pm144.409}$ | $1196.470_{\pm132.269}$ | $1277.320_{\pm143.210}$ |

## A.9 DETAILED EXPERIMENTAL RESULT

### A.9.1 IMPUTATION FIDELITY

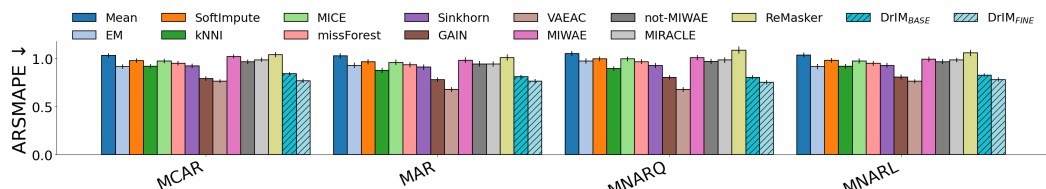

Figure 4: Imputation fidelity at **0.2** missingness rate. The missing mechanism corresponding to the figure is indicated below the figure. The means and the standard errors of the mean across 10 datasets and 10 repeated experiments are reported.

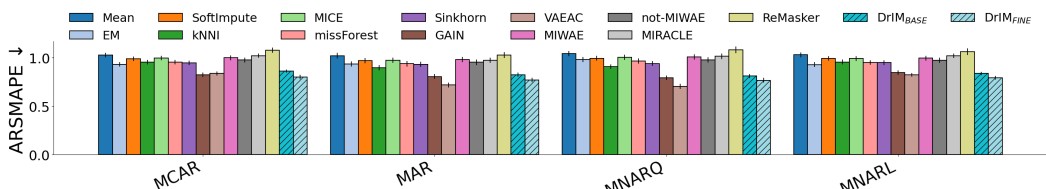

Figure 5: Imputation fidelity at **0.4** missingness rate. The missing mechanism corresponding to the figure is indicated below the figure. The means and the standard errors of the mean across 10 datasets and 10 repeated experiments are reported.

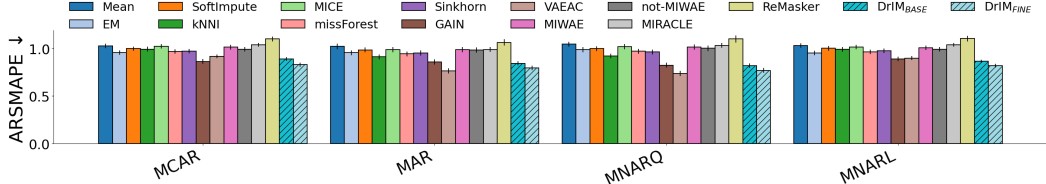

Figure 6: Imputation fidelity at **0.6** missingness rate. The missing mechanism corresponding to the figure is indicated below the figure. The means and the standard errors of the mean across 10 datasets and 10 repeated experiments are reported.

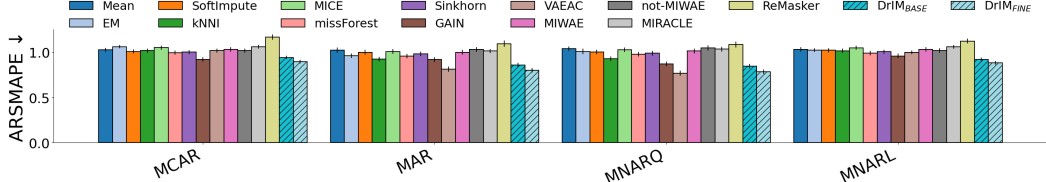

Figure 7: Imputation fidelity at **0.8** missingness rate. The missing mechanism corresponding to the figure is indicated below the figure. The means and the standard errors of the mean across 10 datasets and 10 repeated experiments are reported.

### A.9.2 MACHINE LEARNING UTILITY

Table 8: Machine learning utility at **0.2** missingness rate. The means and the standard errors of the mean across 10 datasets and 10 repeated experiments are reported. ↑ denotes higher is better. The best value is bolded, and the second best is underlined.

| model | MCAR $F_1$ ↑ | Model ↑ | Feature ↑ | MAR $F_1$ ↑ | Model ↑ | Feature ↑ | MNARL $F_1$ ↑ | Model ↑ | Feature ↑ | MNARQ $F_1$ ↑ | Model ↑ | Feature ↑ |
|---|---|---|---|---|---|---|---|---|---|---|---|---|
| Mean | $.646_{\pm.026}$ | $.543_{\pm.037}$ | $.936_{\pm.005}$ | $.642_{\pm.026}$ | $.710_{\pm.030}$ | $.863_{\pm.013}$ | $.629_{\pm.026}$ | $.623_{\pm.034}$ | $.904_{\pm.009}$ | $.668_{\pm.026}$ | $.651_{\pm.034}$ | $.899_{\pm.009}$ |
| EM | $.657_{\pm.025}$ | $.523_{\pm.032}$ | $.934_{\pm.007}$ | $.641_{\pm.026}$ | $.707_{\pm.037}$ | $.912_{\pm.010}$ | $.640_{\pm.026}$ | $.619_{\pm.038}$ | $.886_{\pm.012}$ | $.669_{\pm.025}$ | $.623_{\pm.032}$ | $.916_{\pm.008}$ |
| SoftImpute | $.650_{\pm.026}$ | $.530_{\pm.035}$ | $.939_{\pm.005}$ | $.643_{\pm.026}$ | $.694_{\pm.032}$ | $.905_{\pm.009}$ | $.631_{\pm.026}$ | $.576_{\pm.036}$ | $.903_{\pm.009}$ | $.669_{\pm.026}$ | $.670_{\pm.031}$ | $.932_{\pm.006}$ |
| kNNI | $.654_{\pm.025}$ | $.572_{\pm.032}$ | $.940_{\pm.008}$ | $.649_{\pm.025}$ | $.705_{\pm.032}$ | $.923_{\pm.010}$ | $.636_{\pm.026}$ | $.605_{\pm.039}$ | $.899_{\pm.012}$ | $.672_{\pm.026}$ | $.646_{\pm.032}$ | $.930_{\pm.007}$ |
| MICE | $.656_{\pm.026}$ | $.489_{\pm.035}$ | $.946_{\pm.005}$ | $.648_{\pm.026}$ | $.690_{\pm.034}$ | $.912_{\pm.009}$ | $.637_{\pm.026}$ | $.601_{\pm.036}$ | $.904_{\pm.010}$ | $.673_{\pm.026}$ | $.633_{\pm.033}$ | $.933_{\pm.007}$ |
| missForest | $.651_{\pm.026}$ | $.557_{\pm.033}$ | $.925_{\pm.007}$ | $.646_{\pm.026}$ | $.705_{\pm.035}$ | $.903_{\pm.009}$ | $.634_{\pm.026}$ | $.595_{\pm.039}$ | $.894_{\pm.011}$ | $.671_{\pm.026}$ | $.659_{\pm.033}$ | $.920_{\pm.007}$ |
| Sinkhorn | $.651_{\pm.026}$ | $.563_{\pm.035}$ | $.932_{\pm.008}$ | $.649_{\pm.025}$ | $.696_{\pm.033}$ | $.911_{\pm.009}$ | $.635_{\pm.026}$ | $.617_{\pm.037}$ | $.900_{\pm.011}$ | $.671_{\pm.026}$ | $.649_{\pm.033}$ | $.918_{\pm.008}$ |
| GAIN | $\underline{.669}_{\pm.025}$ | $.659_{\pm.022}$ | $.935_{\pm.005}$ | $.673_{\pm.025}$ | $.682_{\pm.025}$ | $.931_{\pm.006}$ | $\underline{.662}_{\pm.024}$ | $.630_{\pm.032}$ | $.914_{\pm.008}$ | $.685_{\pm.025}$ | $.682_{\pm.026}$ | $.931_{\pm.005}$ |
| VAEAC | $.611_{\pm.025}$ | $.634_{\pm.019}$ | $.857_{\pm.004}$ | $.610_{\pm.025}$ | $.651_{\pm.017}$ | $.853_{\pm.005}$ | $.609_{\pm.025}$ | $.626_{\pm.019}$ | $.851_{\pm.005}$ | $.614_{\pm.024}$ | $.679_{\pm.014}$ | $.847_{\pm.008}$ |
| MIWAE | $.632_{\pm.026}$ | $.581_{\pm.034}$ | $.931_{\pm.007}$ | $.641_{\pm.026}$ | $.669_{\pm.034}$ | $.902_{\pm.009}$ | $.629_{\pm.026}$ | $.578_{\pm.039}$ | $.896_{\pm.011}$ | $.667_{\pm.026}$ | $.645_{\pm.034}$ | $.929_{\pm.006}$ |
| not-MIWAE | $.647_{\pm.025}$ | $.593_{\pm.032}$ | $.939_{\pm.006}$ | $.643_{\pm.026}$ | $.663_{\pm.035}$ | $.904_{\pm.009}$ | $.628_{\pm.026}$ | $.597_{\pm.037}$ | $.898_{\pm.010}$ | $.669_{\pm.026}$ | $.669_{\pm.032}$ | $.932_{\pm.006}$ |
| MIRACLE | $.648_{\pm.025}$ | $.474_{\pm.038}$ | $.937_{\pm.006}$ | $.646_{\pm.026}$ | $.609_{\pm.039}$ | $.911_{\pm.009}$ | $.630_{\pm.026}$ | $.549_{\pm.041}$ | $.889_{\pm.012}$ | $.670_{\pm.026}$ | $.623_{\pm.033}$ | $.923_{\pm.008}$ |
| ReMasker | $.648_{\pm.026}$ | $.493_{\pm.037}$ | $.907_{\pm.010}$ | $.643_{\pm.026}$ | $.646_{\pm.037}$ | $.888_{\pm.014}$ | $.630_{\pm.027}$ | $.523_{\pm.041}$ | $.870_{\pm.013}$ | $.666_{\pm.026}$ | $.599_{\pm.036}$ | $.919_{\pm.008}$ |
| DrIM$_{BASE}$ | $.665_{\pm.025}$ | $\mathbf{.692}_{\pm.026}$ | $\mathbf{.949}_{\pm.004}$ | $\underline{.673}_{\pm.024}$ | $\underline{.732}_{\pm.029}$ | $\underline{.934}_{\pm.006}$ | $.660_{\pm.024}$ | $.658_{\pm.031}$ | $.925_{\pm.007}$ | $\underline{.682}_{\pm.025}$ | $\underline{.771}_{\pm.026}$ | $.945_{\pm.005}$ |
| DrIM$_{FINE}$ | $\mathbf{.678}_{\pm.025}$ | $.690_{\pm.029}$ | $\underline{.948}_{\pm.004}$ | $\mathbf{.684}_{\pm.025}$ | $\mathbf{.773}_{\pm.023}$ | $\mathbf{.948}_{\pm.004}$ | $\mathbf{.673}_{\pm.025}$ | $\mathbf{.711}_{\pm.026}$ | $\mathbf{.938}_{\pm.006}$ | $\mathbf{.689}_{\pm.025}$ | $\mathbf{.784}_{\pm.021}$ | $\mathbf{.948}_{\pm.004}$ |

Table 9: Machine learning utility at **0.3** missingness rate. The means and the standard errors of the mean across 10 datasets and 10 repeated experiments are reported. ↑ denotes higher is better. The best value is bolded, and the second best is underlined.

| model | MCAR $F_1$ ↑ | Model ↑ | Feature ↑ | MAR $F_1$ ↑ | Model ↑ | Feature ↑ | MNARL $F_1$ ↑ | Model ↑ | Feature ↑ | MNARQ $F_1$ ↑ | Model ↑ | Feature ↑ |
|---|---|---|---|---|---|---|---|---|---|---|---|---|
| Mean | $.581_{\pm.027}$ | $.289_{\pm.052}$ | $.839_{\pm.017}$ | $.591_{\pm.027}$ | $.548_{\pm.041}$ | $.744_{\pm.021}$ | $.560_{\pm.027}$ | $.428_{\pm.041}$ | $.797_{\pm.017}$ | $.620_{\pm.028}$ | $.417_{\pm.052}$ | $.771_{\pm.021}$ |
| kNNI | $.595_{\pm.026}$ | $.292_{\pm.047}$ | $.895_{\pm.014}$ | $.601_{\pm.027}$ | $.560_{\pm.043}$ | $.868_{\pm.014}$ | $.575_{\pm.027}$ | $.419_{\pm.049}$ | $.843_{\pm.016}$ | $.629_{\pm.027}$ | $.448_{\pm.048}$ | $.877_{\pm.013}$ |
| EM | $.599_{\pm.027}$ | $.244_{\pm.048}$ | $.873_{\pm.015}$ | $.598_{\pm.027}$ | $.531_{\pm.043}$ | $.833_{\pm.016}$ | $.580_{\pm.027}$ | $.407_{\pm.048}$ | $.804_{\pm.019}$ | $.626_{\pm.028}$ | $.412_{\pm.050}$ | $.835_{\pm.018}$ |
| SoftImpute | $.583_{\pm.027}$ | $.233_{\pm.044}$ | $.866_{\pm.013}$ | $.590_{\pm.027}$ | $.554_{\pm.039}$ | $.845_{\pm.014}$ | $.562_{\pm.027}$ | $.342_{\pm.045}$ | $.818_{\pm.015}$ | $.622_{\pm.028}$ | $.403_{\pm.050}$ | $.871_{\pm.013}$ |
| MICE | $.593_{\pm.027}$ | $.247_{\pm.046}$ | $.892_{\pm.013}$ | $.598_{\pm.027}$ | $.543_{\pm.043}$ | $.859_{\pm.014}$ | $.571_{\pm.027}$ | $.339_{\pm.050}$ | $.825_{\pm.017}$ | $.628_{\pm.028}$ | $.410_{\pm.051}$ | $.876_{\pm.014}$ |
| missForest | $.588_{\pm.027}$ | $.229_{\pm.046}$ | $.858_{\pm.015}$ | $.596_{\pm.027}$ | $.548_{\pm.044}$ | $.833_{\pm.015}$ | $.569_{\pm.027}$ | $.366_{\pm.048}$ | $.809_{\pm.016}$ | $.625_{\pm.028}$ | $.401_{\pm.048}$ | $.857_{\pm.014}$ |
| Sinkhorn | $.589_{\pm.027}$ | $.242_{\pm.043}$ | $.868_{\pm.015}$ | $.599_{\pm.027}$ | $.545_{\pm.044}$ | $.845_{\pm.015}$ | $.569_{\pm.027}$ | $.377_{\pm.049}$ | $.821_{\pm.017}$ | $.626_{\pm.028}$ | $.448_{\pm.045}$ | $.840_{\pm.019}$ |
| GAIN | $.629_{\pm.025}$ | $.473_{\pm.034}$ | $.856_{\pm.012}$ | $\underline{.640}_{\pm.025}$ | $.602_{\pm.029}$ | $.877_{\pm.011}$ | $.621_{\pm.025}$ | $.446_{\pm.040}$ | $.843_{\pm.016}$ | $\underline{.659}_{\pm.026}$ | $.567_{\pm.037}$ | $.875_{\pm.013}$ |
| VAEAC | $.604_{\pm.025}$ | $\underline{.558}_{\pm.022}$ | $.832_{\pm.007}$ | $.606_{\pm.025}$ | $.612_{\pm.018}$ | $.827_{\pm.007}$ | $.598_{\pm.025}$ | $\mathbf{.553}_{\pm.024}$ | $.816_{\pm.012}$ | $.610_{\pm.025}$ | $.593_{\pm.020}$ | $.823_{\pm.012}$ |
| MIWAE | $.577_{\pm.027}$ | $.245_{\pm.052}$ | $.864_{\pm.014}$ | $.588_{\pm.027}$ | $.519_{\pm.048}$ | $.834_{\pm.014}$ | $.556_{\pm.028}$ | $.369_{\pm.053}$ | $.821_{\pm.016}$ | $.618_{\pm.028}$ | $.405_{\pm.054}$ | $.863_{\pm.014}$ |
| not-MIWAE | $.580_{\pm.027}$ | $.234_{\pm.049}$ | $.869_{\pm.015}$ | $.587_{\pm.027}$ | $.540_{\pm.043}$ | $.839_{\pm.013}$ | $.558_{\pm.027}$ | $.394_{\pm.049}$ | $.818_{\pm.017}$ | $.622_{\pm.028}$ | $.438_{\pm.049}$ | $.874_{\pm.013}$ |
| MIRACLE | $.576_{\pm.027}$ | $.167_{\pm.048}$ | $.876_{\pm.013}$ | $.597_{\pm.027}$ | $.473_{\pm.046}$ | $.851_{\pm.014}$ | $.564_{\pm.027}$ | $.264_{\pm.051}$ | $.825_{\pm.016}$ | $.625_{\pm.028}$ | $.333_{\pm.053}$ | $.865_{\pm.016}$ |
| ReMasker | $.583_{\pm.029}$ | $.229_{\pm.046}$ | $.823_{\pm.017}$ | $.592_{\pm.027}$ | $.480_{\pm.046}$ | $.817_{\pm.019}$ | $.566_{\pm.028}$ | $.279_{\pm.050}$ | $.782_{\pm.021}$ | $.617_{\pm.028}$ | $.340_{\pm.050}$ | $.835_{\pm.016}$ |
| DrIM$_{BASE}$ | $\underline{.638}_{\pm.024}$ | $.517_{\pm.037}$ | $\underline{.920}_{\pm.006}$ | $.640_{\pm.024}$ | $\underline{.617}_{\pm.041}$ | $.894_{\pm.010}$ | $.625_{\pm.024}$ | $.496_{\pm.043}$ | $.886_{\pm.009}$ | $.658_{\pm.025}$ | $\mathbf{.694}_{\pm.029}$ | $\mathbf{.913}_{\pm.007}$ |
| DrIM$_{FINE}$ | $\mathbf{.656}_{\pm.025}$ | $\mathbf{.616}_{\pm.032}$ | $\mathbf{.921}_{\pm.006}$ | $\mathbf{.659}_{\pm.025}$ | $\mathbf{.658}_{\pm.032}$ | $\mathbf{.905}_{\pm.007}$ | $\mathbf{.653}_{\pm.025}$ | $.553_{\pm.040}$ | $\mathbf{.915}_{\pm.006}$ | $\mathbf{.673}_{\pm.025}$ | $\underline{.691}_{\pm.030}$ | $.910_{\pm.007}$ |

Table 10: Machine learning utility at **0.4** missingness rate. The means and the standard errors of the mean across 10 datasets and 10 repeated experiments are reported. ↑ denotes higher is better. The best value is bolded, and the second best is underlined.

| model | MCAR $F_1$ ↑ | Model ↑ | Feature ↑ | MAR $F_1$ ↑ | Model ↑ | Feature ↑ | MNARL $F_1$ ↑ | Model ↑ | Feature ↑ | MNARQ $F_1$ ↑ | Model ↑ | Feature ↑ |
|---|---|---|---|---|---|---|---|---|---|---|---|---|
| Mean | $.496_{\pm.030}$ | $.062_{\pm.046}$ | $.672_{\pm.037}$ | $.533_{\pm.028}$ | $.370_{\pm.047}$ | $.607_{\pm.029}$ | $.493_{\pm.029}$ | $.201_{\pm.050}$ | $.662_{\pm.031}$ | $.561_{\pm.030}$ | $.293_{\pm.057}$ | $.632_{\pm.030}$ |
| EM | $.532_{\pm.028}$ | $.093_{\pm.052}$ | $.819_{\pm.018}$ | $.540_{\pm.028}$ | $.369_{\pm.050}$ | $.780_{\pm.020}$ | $.522_{\pm.028}$ | $.179_{\pm.050}$ | $.733_{\pm.024}$ | $.578_{\pm.030}$ | $.302_{\pm.056}$ | $.767_{\pm.024}$ |
| SoftImpute | $.491_{\pm.030}$ | $-.080_{\pm.045}$ | $.740_{\pm.025}$ | $.534_{\pm.029}$ | $.285_{\pm.049}$ | $.743_{\pm.024}$ | $.491_{\pm.029}$ | $.118_{\pm.052}$ | $.715_{\pm.022}$ | $.563_{\pm.030}$ | $.240_{\pm.058}$ | $.785_{\pm.023}$ |
| kNNI | $.529_{\pm.028}$ | $.087_{\pm.053}$ | $.849_{\pm.014}$ | $.547_{\pm.028}$ | $.343_{\pm.051}$ | $.806_{\pm.018}$ | $.521_{\pm.027}$ | $.238_{\pm.055}$ | $.775_{\pm.021}$ | $.576_{\pm.030}$ | $.271_{\pm.055}$ | $.806_{\pm.021}$ |
| MICE | $.506_{\pm.030}$ | $-.032_{\pm.049}$ | $.851_{\pm.015}$ | $.539_{\pm.029}$ | $.319_{\pm.053}$ | $.817_{\pm.017}$ | $.501_{\pm.029}$ | $.084_{\pm.052}$ | $.774_{\pm.020}$ | $.569_{\pm.031}$ | $.265_{\pm.057}$ | $.816_{\pm.020}$ |
| missForest | $.509_{\pm.029}$ | $-.001_{\pm.044}$ | $.791_{\pm.021}$ | $.541_{\pm.028}$ | $.329_{\pm.050}$ | $.775_{\pm.019}$ | $.503_{\pm.029}$ | $.182_{\pm.049}$ | $.731_{\pm.022}$ | $.570_{\pm.030}$ | $.266_{\pm.056}$ | $.779_{\pm.022}$ |
| Sinkhorn | $.511_{\pm.029}$ | $.001_{\pm.047}$ | $.798_{\pm.021}$ | $.540_{\pm.028}$ | $.341_{\pm.048}$ | $.776_{\pm.021}$ | $.505_{\pm.028}$ | $.173_{\pm.049}$ | $.748_{\pm.022}$ | $.572_{\pm.030}$ | $.271_{\pm.056}$ | $.748_{\pm.027}$ |
| GAIN | $.587_{\pm.026}$ | $.320_{\pm.044}$ | $.725_{\pm.028}$ | $\underline{.612}_{\pm.026}$ | $.475_{\pm.036}$ | $.781_{\pm.019}$ | $.583_{\pm.026}$ | $.250_{\pm.043}$ | $.726_{\pm.022}$ | $\underline{.638}_{\pm.026}$ | $.461_{\pm.040}$ | $.794_{\pm.019}$ |
| VAEAC | $.595_{\pm.025}$ | $\underline{.488}_{\pm.026}$ | $.789_{\pm.013}$ | $.598_{\pm.025}$ | $\underline{.521}_{\pm.025}$ | $.790_{\pm.012}$ | $\underline{.592}_{\pm.025}$ | $\mathbf{.495}_{\pm.025}$ | $.788_{\pm.014}$ | $.603_{\pm.025}$ | $\underline{.585}_{\pm.023}$ | $.782_{\pm.016}$ |
| MIWAE | $.491_{\pm.030}$ | $-.041_{\pm.047}$ | $.791_{\pm.021}$ | $.530_{\pm.028}$ | $.317_{\pm.051}$ | $.781_{\pm.019}$ | $.487_{\pm.029}$ | $.083_{\pm.049}$ | $.739_{\pm.021}$ | $.560_{\pm.030}$ | $.236_{\pm.056}$ | $.790_{\pm.021}$ |
| not-MIWAE | $.490_{\pm.030}$ | $-.074_{\pm.050}$ | $.798_{\pm.020}$ | $.529_{\pm.028}$ | $.347_{\pm.049}$ | $.770_{\pm.020}$ | $.486_{\pm.029}$ | $.103_{\pm.054}$ | $.746_{\pm.019}$ | $.559_{\pm.030}$ | $.246_{\pm.058}$ | $.780_{\pm.022}$ |
| MIRACLE | $.499_{\pm.029}$ | $-.055_{\pm.049}$ | $.812_{\pm.015}$ | $.535_{\pm.028}$ | $.277_{\pm.048}$ | $.769_{\pm.021}$ | $.493_{\pm.028}$ | $.029_{\pm.051}$ | $.737_{\pm.024}$ | $.566_{\pm.030}$ | $.203_{\pm.054}$ | $.797_{\pm.023}$ |
| ReMasker | $.496_{\pm.031}$ | $.081_{\pm.050}$ | $.750_{\pm.026}$ | $.535_{\pm.029}$ | $.341_{\pm.048}$ | $.718_{\pm.028}$ | $.499_{\pm.029}$ | $.104_{\pm.054}$ | $.703_{\pm.027}$ | $.557_{\pm.031}$ | $.249_{\pm.050}$ | $.759_{\pm.025}$ |
| DrIM$_{\text{BASE}}$ | $\underline{.602}_{\pm.024}$ | $.301_{\pm.048}$ | $\underline{.853}_{\pm.012}$ | $.607_{\pm.024}$ | $.452_{\pm.044}$ | $\underline{.837}_{\pm.015}$ | $.591_{\pm.023}$ | $.326_{\pm.048}$ | $\underline{.832}_{\pm.014}$ | $.629_{\pm.025}$ | $.534_{\pm.037}$ | $\underline{.850}_{\pm.012}$ |
| DrIM$_{\text{FINE}}$ | $\mathbf{.632}_{\pm.025}$ | $\mathbf{.503}_{\pm.038}$ | $\mathbf{.889}_{\pm.011}$ | $\mathbf{.639}_{\pm.025}$ | $\mathbf{.556}_{\pm.035}$ | $\mathbf{.853}_{\pm.012}$ | $\mathbf{.625}_{\pm.024}$ | $\underline{.481}_{\pm.042}$ | $\mathbf{.864}_{\pm.010}$ | $\mathbf{.655}_{\pm.025}$ | $\mathbf{.618}_{\pm.033}$ | $\mathbf{.853}_{\pm.012}$ |

Table 11: Machine learning utility at **0.6** missingness rate. The means and the standard errors of the mean across 10 datasets and 10 repeated experiments are reported. ↑ denotes higher is better. The best value is bolded, and the second best is underlined.

| model | MCAR $F_1$ ↑ | Model ↑ | Feature ↑ | MAR $F_1$ ↑ | Model ↑ | Feature ↑ | MNARL $F_1$ ↑ | Model ↑ | Feature ↑ | MNARQ $F_1$ ↑ | Model ↑ | Feature ↑ |
|---|---|---|---|---|---|---|---|---|---|---|---|---|
| Mean | $.377_{\pm.031}$ | $-.186_{\pm.051}$ | $.422_{\pm.047}$ | $.433_{\pm.030}$ | $.012_{\pm.059}$ | $.411_{\pm.037}$ | $.380_{\pm.030}$ | $-.138_{\pm.048}$ | $.427_{\pm.043}$ | $.464_{\pm.034}$ | $.034_{\pm.063}$ | $.450_{\pm.041}$ |
| EM | $.429_{\pm.030}$ | $-.255_{\pm.047}$ | $.669_{\pm.025}$ | $.461_{\pm.029}$ | $.024_{\pm.054}$ | $.626_{\pm.029}$ | $.432_{\pm.029}$ | $-.175_{\pm.049}$ | $.569_{\pm.033}$ | $.494_{\pm.033}$ | $.024_{\pm.059}$ | $.567_{\pm.034}$ |
| SoftImpute | $.363_{\pm.030}$ | $-.277_{\pm.053}$ | $.445_{\pm.045}$ | $.432_{\pm.031}$ | $-.027_{\pm.060}$ | $.469_{\pm.043}$ | $.374_{\pm.030}$ | $-.226_{\pm.048}$ | $.472_{\pm.041}$ | $.462_{\pm.034}$ | $-.006_{\pm.064}$ | $.567_{\pm.037}$ |
| kNNI | $.441_{\pm.029}$ | $-.151_{\pm.058}$ | $\mathbf{.749}_{\pm.020}$ | $.458_{\pm.030}$ | $.033_{\pm.057}$ | $.677_{\pm.024}$ | $.439_{\pm.029}$ | $-.100_{\pm.054}$ | $.609_{\pm.027}$ | $.486_{\pm.033}$ | $.074_{\pm.061}$ | $.659_{\pm.031}$ |
| MICE | $.361_{\pm.031}$ | $-.295_{\pm.052}$ | $.732_{\pm.019}$ | $.441_{\pm.031}$ | $-.001_{\pm.055}$ | $.684_{\pm.026}$ | $.378_{\pm.031}$ | $-.247_{\pm.052}$ | $.624_{\pm.028}$ | $.468_{\pm.034}$ | $.029_{\pm.061}$ | $.673_{\pm.031}$ |
| missForest | $.379_{\pm.031}$ | $-.257_{\pm.048}$ | $.626_{\pm.029}$ | $.447_{\pm.031}$ | $-.038_{\pm.057}$ | $.610_{\pm.031}$ | $.389_{\pm.030}$ | $-.229_{\pm.050}$ | $.546_{\pm.030}$ | $.475_{\pm.034}$ | $.005_{\pm.064}$ | $.609_{\pm.034}$ |
| Sinkhorn | $.389_{\pm.030}$ | $-.230_{\pm.052}$ | $.542_{\pm.037}$ | $.448_{\pm.030}$ | $-.025_{\pm.058}$ | $.572_{\pm.033}$ | $.399_{\pm.030}$ | $-.214_{\pm.052}$ | $.539_{\pm.031}$ | $.479_{\pm.033}$ | $.034_{\pm.064}$ | $.560_{\pm.039}$ |
| GAIN | $.499_{\pm.027}$ | $-.073_{\pm.053}$ | $.482_{\pm.040}$ | $.525_{\pm.027}$ | $.172_{\pm.049}$ | $.585_{\pm.030}$ | $.494_{\pm.025}$ | $-.058_{\pm.049}$ | $.456_{\pm.035}$ | $.575_{\pm.027}$ | $.237_{\pm.054}$ | $.593_{\pm.035}$ |
| VAEAC | $\underline{.565}_{\pm.024}$ | $\underline{.391}_{\pm.028}$ | $.693_{\pm.021}$ | $\underline{.574}_{\pm.025}$ | $\underline{.421}_{\pm.033}$ | $\underline{.723}_{\pm.019}$ | $\underline{.549}_{\pm.025}$ | $\underline{.339}_{\pm.033}$ | $.643_{\pm.026}$ | $\underline{.587}_{\pm.025}$ | $\mathbf{.462}_{\pm.032}$ | $\underline{.693}_{\pm.026}$ |
| MIWAE | $.357_{\pm.031}$ | $-.313_{\pm.048}$ | $.669_{\pm.028}$ | $.433_{\pm.031}$ | $-.028_{\pm.056}$ | $.638_{\pm.028}$ | $.376_{\pm.031}$ | $-.299_{\pm.047}$ | $.583_{\pm.030}$ | $.460_{\pm.034}$ | $.032_{\pm.064}$ | $.656_{\pm.033}$ |
| not-MIWAE | $.355_{\pm.031}$ | $-.290_{\pm.049}$ | $.625_{\pm.032}$ | $.420_{\pm.031}$ | $-.056_{\pm.057}$ | $.583_{\pm.031}$ | $.367_{\pm.030}$ | $-.298_{\pm.049}$ | $.556_{\pm.032}$ | $.456_{\pm.034}$ | $-.020_{\pm.063}$ | $.617_{\pm.033}$ |
| MIRACLE | $.394_{\pm.029}$ | $-.253_{\pm.049}$ | $.634_{\pm.022}$ | $.448_{\pm.030}$ | $-.089_{\pm.058}$ | $.567_{\pm.033}$ | $.393_{\pm.028}$ | $-.217_{\pm.051}$ | $.521_{\pm.031}$ | $.480_{\pm.033}$ | $.015_{\pm.059}$ | $.612_{\pm.032}$ |
| ReMasker | $.378_{\pm.032}$ | $-.221_{\pm.052}$ | $.585_{\pm.032}$ | $.441_{\pm.031}$ | $-.018_{\pm.057}$ | $.521_{\pm.031}$ | $.386_{\pm.030}$ | $-.253_{\pm.051}$ | $.509_{\pm.041}$ | $.469_{\pm.034}$ | $.048_{\pm.061}$ | $.586_{\pm.036}$ |
| DrIM$_{\text{BASE}}$ | $.539_{\pm.024}$ | $.104_{\pm.051}$ | $.708_{\pm.024}$ | $.545_{\pm.024}$ | $\underline{.231}_{\pm.052}$ | $.684_{\pm.024}$ | $.522_{\pm.022}$ | $.102_{\pm.052}$ | $\underline{.652}_{\pm.029}$ | $.583_{\pm.026}$ | $.266_{\pm.050}$ | $.683_{\pm.028}$ |
| DrIM$_{\text{FINE}}$ | $\mathbf{.588}_{\pm.025}$ | $\mathbf{.349}_{\pm.044}$ | $\underline{.748}_{\pm.019}$ | $\mathbf{.589}_{\pm.024}$ | $\mathbf{.373}_{\pm.040}$ | $\mathbf{.734}_{\pm.026}$ | $\mathbf{.573}_{\pm.024}$ | $\underline{.313}_{\pm.045}$ | $\mathbf{.730}_{\pm.021}$ | $\mathbf{.614}_{\pm.026}$ | $\underline{.434}_{\pm.042}$ | $\mathbf{.748}_{\pm.021}$ |

Table 12: Machine learning utility at **0.8** missingness rate. The means and the standard errors of the mean across 10 datasets and 10 repeated experiments are reported. ↑ denotes higher is better. The best value is bolded, and the second best is underlined.

| model | MCAR $F_1$ ↑ | Model ↑ | Feature ↑ | MAR $F_1$ ↑ | Model ↑ | Feature ↑ | MNARL $F_1$ ↑ | Model ↑ | Feature ↑ | MNARQ $F_1$ ↑ | Model ↑ | Feature ↑ |
|---|---|---|---|---|---|---|---|---|---|---|---|---|
| Mean | $.322_{\pm.030}$ | $-.180_{\pm.049}$ | $.247_{\pm.052}$ | $.353_{\pm.031}$ | $-.017_{\pm.056}$ | $.278_{\pm.039}$ | $.311_{\pm.030}$ | $-.064_{\pm.051}$ | $.294_{\pm.043}$ | $.413_{\pm.034}$ | $.043_{\pm.061}$ | $.346_{\pm.041}$ |
| EM | $.366_{\pm.029}$ | $-.186_{\pm.053}$ | $.381_{\pm.042}$ | $.386_{\pm.029}$ | $-.101_{\pm.060}$ | $.418_{\pm.039}$ | $.362_{\pm.028}$ | $-.200_{\pm.050}$ | $.345_{\pm.039}$ | $.441_{\pm.033}$ | $-.039_{\pm.061}$ | $.402_{\pm.038}$ |
| SoftImpute | $.315_{\pm.030}$ | $-.311_{\pm.048}$ | $.263_{\pm.048}$ | $.362_{\pm.031}$ | $-.077_{\pm.057}$ | $.322_{\pm.040}$ | $.311_{\pm.030}$ | $-.165_{\pm.050}$ | $.312_{\pm.044}$ | $.415_{\pm.035}$ | $.050_{\pm.060}$ | $.376_{\pm.041}$ |
| kNNI | $.396_{\pm.031}$ | $-.199_{\pm.055}$ | $\mathbf{.585}_{\pm.030}$ | $.386_{\pm.032}$ | $-.120_{\pm.053}$ | $.549_{\pm.031}$ | $.386_{\pm.031}$ | $-.061_{\pm.054}$ | $\underline{.440}_{\pm.036}$ | $.431_{\pm.035}$ | $-.026_{\pm.060}$ | $.507_{\pm.039}$ |
| MICE | $.292_{\pm.031}$ | $-.331_{\pm.049}$ | $.476_{\pm.030}$ | $.369_{\pm.033}$ | $-.109_{\pm.061}$ | $.502_{\pm.035}$ | $.295_{\pm.031}$ | $-.260_{\pm.049}$ | $.381_{\pm.040}$ | $.417_{\pm.036}$ | $-.040_{\pm.060}$ | $.494_{\pm.039}$ |
| missForest | $.315_{\pm.031}$ | $-.306_{\pm.047}$ | $.331_{\pm.038}$ | $.375_{\pm.032}$ | $-.138_{\pm.055}$ | $.426_{\pm.039}$ | $.314_{\pm.031}$ | $-.215_{\pm.051}$ | $.314_{\pm.042}$ | $.422_{\pm.035}$ | $-.030_{\pm.059}$ | $.422_{\pm.041}$ |
| Sinkhorn | $.330_{\pm.030}$ | $-.211_{\pm.054}$ | $.272_{\pm.040}$ | $.373_{\pm.031}$ | $-.010_{\pm.055}$ | $.389_{\pm.039}$ | $.324_{\pm.029}$ | $-.099_{\pm.055}$ | $.299_{\pm.039}$ | $.425_{\pm.034}$ | $-.063_{\pm.058}$ | $.401_{\pm.042}$ |
| GAIN | $.411_{\pm.026}$ | $-.161_{\pm.054}$ | $.298_{\pm.042}$ | $.438_{\pm.027}$ | $-.001_{\pm.054}$ | $.364_{\pm.040}$ | $.391_{\pm.026}$ | $-.136_{\pm.052}$ | $.244_{\pm.040}$ | $.514_{\pm.029}$ | $.155_{\pm.059}$ | $.451_{\pm.041}$ |
| VAEAC | $\mathbf{.505}_{\pm.024}$ | $\underline{.185}_{\pm.039}$ | $\underline{.534}_{\pm.030}$ | $\mathbf{.536}_{\pm.025}$ | $\mathbf{.316}_{\pm.036}$ | $\mathbf{.611}_{\pm.026}$ | $.492_{\pm.024}$ | $\underline{.217}_{\pm.041}$ | $\mathbf{.488}_{\pm.033}$ | $\mathbf{.558}_{\pm.026}$ | $\mathbf{.322}_{\pm.037}$ | $\underline{.560}_{\pm.032}$ |
| MIWAE | $.287_{\pm.031}$ | $-.322_{\pm.049}$ | $.397_{\pm.039}$ | $.364_{\pm.032}$ | $-.116_{\pm.059}$ | $.454_{\pm.036}$ | $.299_{\pm.031}$ | $-.283_{\pm.046}$ | $.348_{\pm.038}$ | $.409_{\pm.035}$ | $-.032_{\pm.064}$ | $.487_{\pm.038}$ |
| not-MIWAE | $.288_{\pm.030}$ | $-.364_{\pm.042}$ | $.357_{\pm.035}$ | $.347_{\pm.032}$ | $-.146_{\pm.056}$ | $.406_{\pm.039}$ | $.288_{\pm.030}$ | $-.332_{\pm.044}$ | $.335_{\pm.037}$ | $.405_{\pm.035}$ | $-.024_{\pm.057}$ | $.473_{\pm.033}$ |
| MIRACLE | $.320_{\pm.028}$ | $-.273_{\pm.049}$ | $.368_{\pm.032}$ | $.369_{\pm.030}$ | $-.183_{\pm.058}$ | $.367_{\pm.040}$ | $.316_{\pm.028}$ | $-.216_{\pm.050}$ | $.290_{\pm.036}$ | $.419_{\pm.033}$ | $-.097_{\pm.054}$ | $.420_{\pm.039}$ |
| ReMasker | $.304_{\pm.028}$ | $-.283_{\pm.053}$ | $.299_{\pm.049}$ | $.374_{\pm.031}$ | $-.140_{\pm.055}$ | $.364_{\pm.042}$ | $.306_{\pm.029}$ | $-.285_{\pm.051}$ | $.286_{\pm.045}$ | $.418_{\pm.035}$ | $-.034_{\pm.058}$ | $.418_{\pm.037}$ |
| DrIM$_{\text{BASE}}$ | $.471_{\pm.023}$ | $.027_{\pm.050}$ | $.272_{\pm.043}$ | $.477_{\pm.023}$ | $.072_{\pm.054}$ | $.446_{\pm.041}$ | $.449_{\pm.022}$ | $.049_{\pm.051}$ | $.286_{\pm.043}$ | $.515_{\pm.026}$ | $.151_{\pm.060}$ | $.545_{\pm.033}$ |
| DrIM$_{\text{FINE}}$ | $\underline{.503}_{\pm.025}$ | $\mathbf{.176}_{\pm.055}$ | $.309_{\pm.039}$ | $\underline{.534}_{\pm.024}$ | $\underline{.257}_{\pm.047}$ | $\underline{.604}_{\pm.035}$ | $\mathbf{.492}_{\pm.024}$ | $\mathbf{.235}_{\pm.054}$ | $.287_{\pm.042}$ | $\underline{.572}_{\pm.025}$ | $\underline{.281}_{\pm.052}$ | $\mathbf{.616}_{\pm.031}$ |

### A.9.3 MACHINE LEARNING UTILITY FOR EACH DATASET

Table 13: Machine learning utility for each dataset under **MCAR** at 0.3 missingness. The means and the standard errors of the mean across 10 repeated experiments are reported. ↑ denotes higher is better.

| model | abalone | | | anuran | | |
|---|---|---|---|---|---|---|
| | $F_1 \uparrow$ | Model ↑ | Feature ↑ | $F_1 \uparrow$ | Model ↑ | Feature ↑ |
| Mean | $.125_{\pm.002}$ | $-.587_{\pm.059}$ | $.864_{\pm.015}$ | $.858_{\pm.002}$ | $.172_{\pm.113}$ | $.836_{\pm.012}$ |
| kNNI | $.135_{\pm.003}$ | $-.475_{\pm.069}$ | $.983_{\pm.005}$ | $.832_{\pm.003}$ | $.105_{\pm.106}$ | $.496_{\pm.008}$ |
| EM | $.128_{\pm.003}$ | $-.505_{\pm.074}$ | $.962_{\pm.012}$ | $.841_{\pm.003}$ | $-.133_{\pm.091}$ | $.457_{\pm.011}$ |
| SoftImpute | $.118_{\pm.004}$ | $-.523_{\pm.054}$ | $.967_{\pm.010}$ | $.855_{\pm.003}$ | $.125_{\pm.126}$ | $.626_{\pm.026}$ |
| MICE | $.120_{\pm.003}$ | $-.542_{\pm.048}$ | $.974_{\pm.007}$ | $.858_{\pm.004}$ | $.105_{\pm.111}$ | $.545_{\pm.014}$ |
| missForest | $.121_{\pm.003}$ | $-.507_{\pm.045}$ | $.976_{\pm.006}$ | $.850_{\pm.003}$ | $.080_{\pm.109}$ | $.485_{\pm.012}$ |
| Sinkhorn | $.125_{\pm.003}$ | $-.547_{\pm.069}$ | $.967_{\pm.010}$ | $.843_{\pm.003}$ | $.015_{\pm.094}$ | $.473_{\pm.009}$ |
| GAIN | $.196_{\pm.003}$ | $.019_{\pm.150}$ | $.850_{\pm.014}$ | $.902_{\pm.011}$ | $.452_{\pm.059}$ | $.772_{\pm.032}$ |
| VAEAC | $.132_{\pm.002}$ | $.322_{\pm.087}$ | $.821_{\pm.013}$ | $.892_{\pm.000}$ | $.535_{\pm.070}$ | $.872_{\pm.004}$ |
| MIWAE | $.113_{\pm.003}$ | $-.578_{\pm.047}$ | $.967_{\pm.016}$ | $.855_{\pm.003}$ | $.225_{\pm.099}$ | $.517_{\pm.019}$ |
| not-MIWAE | $.114_{\pm.003}$ | $-.607_{\pm.049}$ | $.957_{\pm.011}$ | $.854_{\pm.003}$ | $.201_{\pm.121}$ | $.521_{\pm.014}$ |
| MIRACLE | $.126_{\pm.003}$ | $-.446_{\pm.051}$ | $.967_{\pm.009}$ | $.821_{\pm.005}$ | $-.188_{\pm.093}$ | $.538_{\pm.022}$ |
| ReMasker | $.119_{\pm.004}$ | $-.543_{\pm.051}$ | $.886_{\pm.022}$ | $.837_{\pm.003}$ | $.045_{\pm.090}$ | $.479_{\pm.010}$ |
| DrIM$_{\text{BASE}}$ | $.200_{\pm.002}$ | $.571_{\pm.142}$ | $.893_{\pm.021}$ | $.926_{\pm.001}$ | $.262_{\pm.092}$ | $.947_{\pm.006}$ |
| DrIM$_{\text{FINE}}$ | $.203_{\pm.003}$ | $.534_{\pm.109}$ | $.926_{\pm.012}$ | $.970_{\pm.007}$ | $.384_{\pm.112}$ | $.938_{\pm.005}$ |

| model | banknote | | | breast | | |
|---|---|---|---|---|---|---|
| | $F_1 \uparrow$ | Model ↑ | Feature ↑ | $F_1 \uparrow$ | Model ↑ | Feature ↑ |
| Mean | $.863_{\pm.006}$ | $.731_{\pm.056}$ | $1.000_{\pm.000}$ | $.896_{\pm.006}$ | $.198_{\pm.119}$ | $.872_{\pm.011}$ |
| kNNI | $.869_{\pm.006}$ | $.641_{\pm.095}$ | $1.000_{\pm.000}$ | $.883_{\pm.010}$ | $-.007_{\pm.107}$ | $.878_{\pm.014}$ |
| EM | $.912_{\pm.005}$ | $.525_{\pm.075}$ | $1.000_{\pm.000}$ | $.885_{\pm.008}$ | $.004_{\pm.074}$ | $.876_{\pm.015}$ |
| SoftImpute | $.886_{\pm.006}$ | $.438_{\pm.055}$ | $1.000_{\pm.000}$ | $.881_{\pm.008}$ | $.147_{\pm.082}$ | $.865_{\pm.015}$ |
| MICE | $.906_{\pm.006}$ | $.493_{\pm.056}$ | $1.000_{\pm.000}$ | $.888_{\pm.007}$ | $.023_{\pm.089}$ | $.873_{\pm.016}$ |
| missForest | $.895_{\pm.005}$ | $.498_{\pm.047}$ | $1.000_{\pm.000}$ | $.880_{\pm.007}$ | $-.072_{\pm.117}$ | $.892_{\pm.009}$ |
| Sinkhorn | $.882_{\pm.004}$ | $.483_{\pm.042}$ | $1.000_{\pm.000}$ | $.886_{\pm.008}$ | $.035_{\pm.111}$ | $.885_{\pm.014}$ |
| GAIN | $.909_{\pm.012}$ | $.577_{\pm.072}$ | $.980_{\pm.020}$ | $.919_{\pm.008}$ | $.113_{\pm.100}$ | $.913_{\pm.009}$ |
| VAEAC | $.846_{\pm.003}$ | $.623_{\pm.032}$ | $.900_{\pm.021}$ | $.847_{\pm.004}$ | $.655_{\pm.027}$ | $.850_{\pm.004}$ |
| MIWAE | $.871_{\pm.008}$ | $.745_{\pm.060}$ | $.980_{\pm.020}$ | $.888_{\pm.008}$ | $-.036_{\pm.089}$ | $.901_{\pm.008}$ |
| not-MIWAE | $.867_{\pm.007}$ | $.596_{\pm.073}$ | $1.000_{\pm.000}$ | $.883_{\pm.009}$ | $.017_{\pm.139}$ | $.901_{\pm.010}$ |
| MIRACLE | $.902_{\pm.007}$ | $.525_{\pm.069}$ | $1.000_{\pm.000}$ | $.783_{\pm.022}$ | $-.096_{\pm.093}$ | $.832_{\pm.020}$ |
| ReMasker | $.911_{\pm.006}$ | $.497_{\pm.044}$ | $1.000_{\pm.000}$ | $.905_{\pm.008}$ | $.405_{\pm.125}$ | $.849_{\pm.019}$ |
| DrIM$_{\text{BASE}}$ | $.879_{\pm.004}$ | $.723_{\pm.083}$ | $1.000_{\pm.000}$ | $.887_{\pm.008}$ | $.171_{\pm.166}$ | $.824_{\pm.012}$ |
| DrIM$_{\text{FINE}}$ | $.867_{\pm.007}$ | $.674_{\pm.075}$ | $.980_{\pm.020}$ | $.932_{\pm.004}$ | $.556_{\pm.138}$ | $.891_{\pm.011}$ |

| | concrete | | | kings | | |
|---|---|---|---|---|---|---|
| model | $F_1 \uparrow$ | Model $\uparrow$ | Feature $\uparrow$ | $F_1 \uparrow$ | Model $\uparrow$ | Feature $\uparrow$ |
| Mean | $.326_{\pm.009}$ | $.175_{\pm.126}$ | $.945_{\pm.007}$ | $.440_{\pm.005}$ | $.507_{\pm.053}$ | $.978_{\pm.002}$ |
| kNNI | $.349_{\pm.007}$ | $.346_{\pm.156}$ | $.940_{\pm.013}$ | $.460_{\pm.004}$ | $.410_{\pm.064}$ | $.972_{\pm.001}$ |
| EM | $.372_{\pm.006}$ | $.419_{\pm.210}$ | $.952_{\pm.014}$ | $.459_{\pm.003}$ | $.270_{\pm.047}$ | $.945_{\pm.002}$ |
| SoftImpute | $.328_{\pm.008}$ | $.209_{\pm.187}$ | $.945_{\pm.010}$ | $.445_{\pm.004}$ | $.310_{\pm.010}$ | $.941_{\pm.004}$ |
| MICE | $.345_{\pm.008}$ | $.341_{\pm.147}$ | $.960_{\pm.009}$ | $.455_{\pm.004}$ | $.290_{\pm.043}$ | $.962_{\pm.001}$ |
| missForest | $.340_{\pm.006}$ | $.304_{\pm.130}$ | $.952_{\pm.012}$ | $.449_{\pm.005}$ | $.470_{\pm.058}$ | $.946_{\pm.002}$ |
| Sinkhorn | $.341_{\pm.006}$ | $.513_{\pm.115}$ | $.933_{\pm.021}$ | $.450_{\pm.004}$ | $.490_{\pm.059}$ | $.976_{\pm.002}$ |
| GAIN | $.430_{\pm.011}$ | $.491_{\pm.074}$ | $.929_{\pm.011}$ | $.491_{\pm.004}$ | $.810_{\pm.035}$ | $.952_{\pm.004}$ |
| VAEAC | $.363_{\pm.007}$ | $.553_{\pm.062}$ | $.771_{\pm.015}$ | $.513_{\pm.002}$ | $.750_{\pm.017}$ | $.862_{\pm.002}$ |
| MIWAE | $.294_{\pm.010}$ | $-.095_{\pm.210}$ | $.926_{\pm.018}$ | $.437_{\pm.005}$ | $.450_{\pm.056}$ | $.960_{\pm.004}$ |
| not-MIWAE | $.313_{\pm.009}$ | $.137_{\pm.186}$ | $.955_{\pm.014}$ | $.441_{\pm.003}$ | $.470_{\pm.047}$ | $.970_{\pm.003}$ |
| MIRACLE | $.283_{\pm.011}$ | $-.214_{\pm.177}$ | $.924_{\pm.014}$ | $.463_{\pm.004}$ | $.280_{\pm.055}$ | $.946_{\pm.001}$ |
| ReMasker | $.269_{\pm.014}$ | $.111_{\pm.111}$ | $.921_{\pm.015}$ | $.437_{\pm.012}$ | $.390_{\pm.108}$ | $.929_{\pm.006}$ |
| DrIM$_{\text{BASE}}$ | $.407_{\pm.006}$ | $.426_{\pm.108}$ | $.931_{\pm.011}$ | $.498_{\pm.008}$ | $.910_{\pm.010}$ | $.964_{\pm.004}$ |
| DrIM$_{\text{FINE}}$ | $.397_{\pm.010}$ | $.504_{\pm.098}$ | $.936_{\pm.022}$ | $.540_{\pm.004}$ | $.950_{\pm.017}$ | $.962_{\pm.003}$ |

| | letter | | | loan | | |
|---|---|---|---|---|---|---|
| model | $F_1 \uparrow$ | Model $\uparrow$ | Feature $\uparrow$ | $F_1 \uparrow$ | Model $\uparrow$ | Feature $\uparrow$ |
| Mean | $.584_{\pm.002}$ | $.700_{\pm.000}$ | $.500_{\pm.015}$ | $.916_{\pm.001}$ | $.707_{\pm.033}$ | $.895_{\pm.007}$ |
| kNNI | $.674_{\pm.002}$ | $.800_{\pm.000}$ | $.948_{\pm.002}$ | $.918_{\pm.001}$ | $.783_{\pm.047}$ | $.902_{\pm.008}$ |
| EM | $.656_{\pm.002}$ | $.800_{\pm.000}$ | $.955_{\pm.003}$ | $.919_{\pm.001}$ | $.697_{\pm.037}$ | $.878_{\pm.017}$ |
| SoftImpute | $.594_{\pm.002}$ | $.690_{\pm.038}$ | $.668_{\pm.015}$ | $.915_{\pm.001}$ | $.724_{\pm.046}$ | $.874_{\pm.008}$ |
| MICE | $.621_{\pm.002}$ | $.860_{\pm.027}$ | $.920_{\pm.005}$ | $.918_{\pm.001}$ | $.733_{\pm.047}$ | $.865_{\pm.014}$ |
| missForest | $.614_{\pm.001}$ | $.660_{\pm.031}$ | $.742_{\pm.004}$ | $.917_{\pm.001}$ | $.787_{\pm.054}$ | $.858_{\pm.016}$ |
| Sinkhorn | $.625_{\pm.001}$ | $.620_{\pm.033}$ | $.789_{\pm.011}$ | $.917_{\pm.001}$ | $.646_{\pm.043}$ | $.880_{\pm.007}$ |
| GAIN | $.599_{\pm.007}$ | $.640_{\pm.033}$ | $.575_{\pm.022}$ | $.915_{\pm.002}$ | $.597_{\pm.039}$ | $.846_{\pm.010}$ |
| VAEAC | $.712_{\pm.002}$ | $.780_{\pm.013}$ | $.884_{\pm.002}$ | $.827_{\pm.002}$ | $.652_{\pm.040}$ | $.849_{\pm.011}$ |
| MIWAE | $.573_{\pm.003}$ | $.812_{\pm.031}$ | $.705_{\pm.017}$ | $.914_{\pm.001}$ | $.720_{\pm.033}$ | $.872_{\pm.007}$ |
| not-MIWAE | $.599_{\pm.002}$ | $.620_{\pm.013}$ | $.689_{\pm.013}$ | $.916_{\pm.000}$ | $.692_{\pm.032}$ | $.876_{\pm.010}$ |
| MIRACLE | $.648_{\pm.006}$ | $.790_{\pm.023}$ | $.940_{\pm.004}$ | $.919_{\pm.001}$ | $.730_{\pm.054}$ | $.882_{\pm.014}$ |
| ReMasker | $.645_{\pm.002}$ | $.740_{\pm.031}$ | $.931_{\pm.004}$ | $.918_{\pm.001}$ | $.727_{\pm.040}$ | $.887_{\pm.016}$ |
| DrIM$_{\text{BASE}}$ | $.661_{\pm.003}$ | $.790_{\pm.023}$ | $.952_{\pm.003}$ | $.910_{\pm.001}$ | $.647_{\pm.030}$ | $.878_{\pm.006}$ |
| DrIM$_{\text{FINE}}$ | $.716_{\pm.004}$ | $1.000_{\pm.000}$ | $.967_{\pm.002}$ | $.912_{\pm.001}$ | $.690_{\pm.038}$ | $.874_{\pm.006}$ |

| | redwine | | | whitewine | | |
|---|---|---|---|---|---|---|
| model | $F_1 \uparrow$ | Model $\uparrow$ | Feature $\uparrow$ | $F_1 \uparrow$ | Model $\uparrow$ | Feature $\uparrow$ |
| Mean | $.426_{\pm.007}$ | $.206_{\pm.090}$ | $.905_{\pm.014}$ | $.373_{\pm.005}$ | $.086_{\pm.122}$ | $.593_{\pm.048}$ |
| kNNI | $.440_{\pm.006}$ | $.171_{\pm.137}$ | $.909_{\pm.021}$ | $.386_{\pm.004}$ | $.150_{\pm.079}$ | $.917_{\pm.008}$ |
| EM | $.435_{\pm.006}$ | $.102_{\pm.144}$ | $.892_{\pm.017}$ | $.384_{\pm.004}$ | $.260_{\pm.056}$ | $.809_{\pm.023}$ |
| SoftImpute | $.430_{\pm.007}$ | $.047_{\pm.124}$ | $.869_{\pm.018}$ | $.377_{\pm.005}$ | $.160_{\pm.083}$ | $.906_{\pm.026}$ |
| MICE | $.428_{\pm.006}$ | $-.022_{\pm.133}$ | $.895_{\pm.020}$ | $.388_{\pm.005}$ | $.190_{\pm.066}$ | $.921_{\pm.021}$ |
| missForest | $.434_{\pm.005}$ | $.001_{\pm.131}$ | $.859_{\pm.023}$ | $.383_{\pm.004}$ | $.070_{\pm.088}$ | $.871_{\pm.009}$ |
| Sinkhorn | $.435_{\pm.005}$ | $.044_{\pm.123}$ | $.884_{\pm.018}$ | $.386_{\pm.005}$ | $.120_{\pm.059}$ | $.894_{\pm.013}$ |
| GAIN | $.509_{\pm.007}$ | $.404_{\pm.110}$ | $.897_{\pm.014}$ | $.425_{\pm.007}$ | $.630_{\pm.072}$ | $.846_{\pm.028}$ |
| VAEAC | $.483_{\pm.005}$ | $.263_{\pm.046}$ | $.785_{\pm.019}$ | $.423_{\pm.004}$ | $.450_{\pm.017}$ | $.722_{\pm.025}$ |
| MIWAE | $.444_{\pm.007}$ | $-.038_{\pm.140}$ | $.883_{\pm.017}$ | $.380_{\pm.004}$ | $.240_{\pm.090}$ | $.925_{\pm.007}$ |
| not-MIWAE | $.432_{\pm.007}$ | $.061_{\pm.151}$ | $.909_{\pm.011}$ | $.384_{\pm.004}$ | $.150_{\pm.092}$ | $.914_{\pm.022}$ |
| MIRACLE | $.430_{\pm.007}$ | $.001_{\pm.085}$ | $.877_{\pm.020}$ | $.379_{\pm.006}$ | $.290_{\pm.050}$ | $.855_{\pm.018}$ |
| ReMasker | $.415_{\pm.013}$ | $-.030_{\pm.099}$ | $.785_{\pm.028}$ | $.370_{\pm.008}$ | $-.050_{\pm.120}$ | $.563_{\pm.048}$ |
| DrIM$_{\text{BASE}}$ | $.532_{\pm.006}$ | $.353_{\pm.110}$ | $.911_{\pm.014}$ | $.476_{\pm.005}$ | $.321_{\pm.057}$ | $.902_{\pm.017}$ |
| DrIM$_{\text{FINE}}$ | $.536_{\pm.008}$ | $.448_{\pm.062}$ | $.892_{\pm.016}$ | $.487_{\pm.006}$ | $.417_{\pm.047}$ | $.840_{\pm.020}$ |

Table 14: Machine learning utility for each dataset under **MAR** at 0.3 missingness. The means and the standard errors of the mean across 10 repeated experiments are reported. ↑ denotes higher is better.

| | abalone | | | anuran | | |
|---|---|---|---|---|---|---|
| model | $F_1$ ↑ | Model ↑ | Feature ↑ | $F_1$ ↑ | Model ↑ | Feature ↑ |
| Mean | $.163_{\pm.014}$ | $.147_{\pm.163}$ | $.836_{\pm.021}$ | $.830_{\pm.018}$ | $-.046_{\pm.150}$ | $.552_{\pm.059}$ |
| kNNI | $.167_{\pm.013}$ | $.142_{\pm.177}$ | $.938_{\pm.016}$ | $.816_{\pm.019}$ | $-.061_{\pm.149}$ | $.683_{\pm.040}$ |
| EM | $.164_{\pm.013}$ | $.182_{\pm.140}$ | $.902_{\pm.037}$ | $.825_{\pm.018}$ | $-.152_{\pm.159}$ | $.658_{\pm.044}$ |
| SoftImpute | $.161_{\pm.012}$ | $.211_{\pm.133}$ | $.926_{\pm.010}$ | $.827_{\pm.018}$ | $.005_{\pm.134}$ | $.671_{\pm.050}$ |
| MICE | $.161_{\pm.013}$ | $.283_{\pm.149}$ | $.962_{\pm.012}$ | $.831_{\pm.017}$ | $-.051_{\pm.145}$ | $.699_{\pm.046}$ |
| missForest | $.161_{\pm.013}$ | $.313_{\pm.159}$ | $.869_{\pm.041}$ | $.831_{\pm.017}$ | $-.015_{\pm.157}$ | $.650_{\pm.043}$ |
| Sinkhorn | $.164_{\pm.013}$ | $.044_{\pm.202}$ | $.960_{\pm.010}$ | $.825_{\pm.018}$ | $.055_{\pm.131}$ | $.668_{\pm.045}$ |
| GAIN | $.206_{\pm.006}$ | $.415_{\pm.171}$ | $.905_{\pm.015}$ | $.863_{\pm.023}$ | $.451_{\pm.066}$ | $.808_{\pm.051}$ |
| VAEAC | $.131_{\pm.005}$ | $.497_{\pm.043}$ | $.824_{\pm.015}$ | $.891_{\pm.001}$ | $.565_{\pm.076}$ | $.871_{\pm.003}$ |
| MIWAE | $.160_{\pm.013}$ | $-.013_{\pm.194}$ | $.931_{\pm.010}$ | $.825_{\pm.018}$ | $-.096_{\pm.162}$ | $.631_{\pm.045}$ |
| not-MIWAE | $.164_{\pm.013}$ | $.055_{\pm.157}$ | $.914_{\pm.019}$ | $.825_{\pm.018}$ | $-.065_{\pm.147}$ | $.681_{\pm.043}$ |
| MIRACLE | $.163_{\pm.013}$ | $.146_{\pm.146}$ | $.900_{\pm.035}$ | $.822_{\pm.018}$ | $-.113_{\pm.144}$ | $.701_{\pm.046}$ |
| ReMasker | $.169_{\pm.012}$ | $.283_{\pm.173}$ | $.888_{\pm.043}$ | $.815_{\pm.019}$ | $-.183_{\pm.170}$ | $.668_{\pm.041}$ |
| DrIM$_{BASE}$ | $.202_{\pm.007}$ | $.688_{\pm.058}$ | $.917_{\pm.017}$ | $.897_{\pm.012}$ | $-.188_{\pm.197}$ | $.904_{\pm.012}$ |
| DrIM$_{FINE}$ | $.197_{\pm.009}$ | $.606_{\pm.118}$ | $.879_{\pm.025}$ | $.971_{\pm.005}$ | $.210_{\pm.125}$ | $.929_{\pm.012}$ |

| | banknote | | | breast | | |
|---|---|---|---|---|---|---|
| model | $F_1$ ↑ | Model ↑ | Feature ↑ | $F_1$ ↑ | Model ↑ | Feature ↑ |
| Mean | $.861_{\pm.020}$ | $.786_{\pm.065}$ | $.800_{\pm.073}$ | $.921_{\pm.013}$ | $.586_{\pm.098}$ | $.810_{\pm.038}$ |
| kNNI | $.876_{\pm.015}$ | $.633_{\pm.074}$ | $.940_{\pm.031}$ | $.911_{\pm.019}$ | $.462_{\pm.124}$ | $.898_{\pm.021}$ |
| EM | $.884_{\pm.015}$ | $.726_{\pm.058}$ | $.940_{\pm.031}$ | $.909_{\pm.020}$ | $.432_{\pm.141}$ | $.861_{\pm.024}$ |
| SoftImpute | $.868_{\pm.020}$ | $.673_{\pm.053}$ | $.960_{\pm.027}$ | $.912_{\pm.018}$ | $.536_{\pm.122}$ | $.903_{\pm.021}$ |
| MICE | $.881_{\pm.020}$ | $.673_{\pm.078}$ | $.940_{\pm.031}$ | $.910_{\pm.021}$ | $.475_{\pm.182}$ | $.912_{\pm.018}$ |
| missForest | $.889_{\pm.017}$ | $.623_{\pm.093}$ | $.960_{\pm.027}$ | $.910_{\pm.019}$ | $.375_{\pm.181}$ | $.910_{\pm.020}$ |
| Sinkhorn | $.873_{\pm.020}$ | $.658_{\pm.066}$ | $.960_{\pm.027}$ | $.917_{\pm.015}$ | $.497_{\pm.125}$ | $.905_{\pm.020}$ |
| GAIN | $.934_{\pm.005}$ | $.729_{\pm.044}$ | $1.000_{\pm.000}$ | $.923_{\pm.013}$ | $.674_{\pm.097}$ | $.914_{\pm.018}$ |
| VAEAC | $.849_{\pm.004}$ | $.678_{\pm.036}$ | $.860_{\pm.027}$ | $.848_{\pm.003}$ | $.622_{\pm.042}$ | $.853_{\pm.005}$ |
| MIWAE | $.865_{\pm.020}$ | $.840_{\pm.069}$ | $.900_{\pm.033}$ | $.908_{\pm.022}$ | $.436_{\pm.163}$ | $.904_{\pm.023}$ |
| not-MIWAE | $.863_{\pm.020}$ | $.570_{\pm.097}$ | $.880_{\pm.033}$ | $.909_{\pm.019}$ | $.550_{\pm.150}$ | $.895_{\pm.024}$ |
| MIRACLE | $.889_{\pm.015}$ | $.709_{\pm.055}$ | $.960_{\pm.027}$ | $.883_{\pm.021}$ | $.250_{\pm.169}$ | $.770_{\pm.027}$ |
| ReMasker | $.887_{\pm.018}$ | $.740_{\pm.045}$ | $.940_{\pm.031}$ | $.905_{\pm.017}$ | $.316_{\pm.183}$ | $.816_{\pm.022}$ |
| DrIM$_{BASE}$ | $.906_{\pm.008}$ | $.776_{\pm.072}$ | $.960_{\pm.027}$ | $.877_{\pm.023}$ | $.485_{\pm.157}$ | $.777_{\pm.057}$ |
| DrIM$_{FINE}$ | $.879_{\pm.014}$ | $.711_{\pm.081}$ | $.940_{\pm.031}$ | $.931_{\pm.008}$ | $.672_{\pm.085}$ | $.893_{\pm.018}$ |

| | concrete | | | kings | | |
|---|---|---|---|---|---|---|
| model | $F_1 \uparrow$ | Model $\uparrow$ | Feature $\uparrow$ | $F_1 \uparrow$ | Model $\uparrow$ | Feature $\uparrow$ |
| Mean | $.350_{\pm.025}$ | $.511_{\pm.100}$ | $.881_{\pm.028}$ | $.421_{\pm.009}$ | $.678_{\pm.083}$ | $.900_{\pm.017}$ |
| kNNI | $.372_{\pm.024}$ | $.769_{\pm.098}$ | $.929_{\pm.013}$ | $.432_{\pm.009}$ | $.656_{\pm.084}$ | $.957_{\pm.005}$ |
| EM | $.391_{\pm.024}$ | $.733_{\pm.043}$ | $.845_{\pm.037}$ | $.423_{\pm.010}$ | $.689_{\pm.065}$ | $.913_{\pm.012}$ |
| SoftImpute | $.349_{\pm.023}$ | $.620_{\pm.101}$ | $.879_{\pm.026}$ | $.419_{\pm.010}$ | $.600_{\pm.091}$ | $.938_{\pm.006}$ |
| MICE | $.370_{\pm.027}$ | $.549_{\pm.132}$ | $.886_{\pm.029}$ | $.423_{\pm.010}$ | $.562_{\pm.089}$ | $.942_{\pm.007}$ |
| missForest | $.353_{\pm.022}$ | $.601_{\pm.149}$ | $.826_{\pm.033}$ | $.428_{\pm.010}$ | $.700_{\pm.087}$ | $.948_{\pm.005}$ |
| Sinkhorn | $.370_{\pm.023}$ | $.595_{\pm.164}$ | $.860_{\pm.020}$ | $.423_{\pm.009}$ | $.725_{\pm.070}$ | $.966_{\pm.004}$ |
| GAIN | $.421_{\pm.013}$ | $.624_{\pm.080}$ | $.869_{\pm.022}$ | $.500_{\pm.011}$ | $.760_{\pm.048}$ | $.959_{\pm.008}$ |
| VAEAC | $.357_{\pm.007}$ | $.649_{\pm.051}$ | $.721_{\pm.027}$ | $.514_{\pm.001}$ | $.770_{\pm.015}$ | $.860_{\pm.006}$ |
| MIWAE | $.335_{\pm.023}$ | $.389_{\pm.111}$ | $.845_{\pm.038}$ | $.416_{\pm.010}$ | $.644_{\pm.078}$ | $.949_{\pm.006}$ |
| not-MIWAE | $.338_{\pm.023}$ | $.602_{\pm.089}$ | $.879_{\pm.026}$ | $.423_{\pm.008}$ | $.690_{\pm.078}$ | $.949_{\pm.006}$ |
| MIRACLE | $.339_{\pm.025}$ | $.247_{\pm.147}$ | $.893_{\pm.019}$ | $.436_{\pm.009}$ | $.578_{\pm.074}$ | $.950_{\pm.010}$ |
| ReMasker | $.332_{\pm.025}$ | $.263_{\pm.084}$ | $.893_{\pm.019}$ | $.423_{\pm.014}$ | $.533_{\pm.078}$ | $.935_{\pm.006}$ |
| DrIM$_{\text{BASE}}$ | $.415_{\pm.011}$ | $.666_{\pm.080}$ | $.900_{\pm.019}$ | $.512_{\pm.008}$ | $.890_{\pm.035}$ | $.948_{\pm.009}$ |
| DrIM$_{\text{FINE}}$ | $.409_{\pm.009}$ | $.690_{\pm.069}$ | $.879_{\pm.025}$ | $.548_{\pm.009}$ | $.960_{\pm.016}$ | $.966_{\pm.003}$ |

| | letter | | | loan | | |
|---|---|---|---|---|---|---|
| model | $F_1 \uparrow$ | Model $\uparrow$ | Feature $\uparrow$ | $F_1 \uparrow$ | Model $\uparrow$ | Feature $\uparrow$ |
| Mean | $.621_{\pm.028}$ | $.840_{\pm.050}$ | $.715_{\pm.032}$ | $.915_{\pm.003}$ | $.670_{\pm.063}$ | $.898_{\pm.020}$ |
| kNNI | $.664_{\pm.028}$ | $.832_{\pm.029}$ | $.936_{\pm.012}$ | $.919_{\pm.002}$ | $.804_{\pm.047}$ | $.935_{\pm.019}$ |
| EM | $.666_{\pm.025}$ | $.840_{\pm.027}$ | $.952_{\pm.006}$ | $.893_{\pm.022}$ | $.732_{\pm.048}$ | $.910_{\pm.021}$ |
| SoftImpute | $.627_{\pm.029}$ | $.880_{\pm.025}$ | $.781_{\pm.023}$ | $.916_{\pm.003}$ | $.673_{\pm.042}$ | $.910_{\pm.021}$ |
| MICE | $.646_{\pm.028}$ | $.890_{\pm.023}$ | $.894_{\pm.014}$ | $.917_{\pm.002}$ | $.759_{\pm.052}$ | $.908_{\pm.027}$ |
| missForest | $.637_{\pm.027}$ | $.857_{\pm.040}$ | $.803_{\pm.027}$ | $.918_{\pm.002}$ | $.761_{\pm.045}$ | $.902_{\pm.026}$ |
| Sinkhorn | $.646_{\pm.027}$ | $.840_{\pm.045}$ | $.843_{\pm.020}$ | $.915_{\pm.003}$ | $.731_{\pm.039}$ | $.912_{\pm.021}$ |
| GAIN | $.645_{\pm.027}$ | $.810_{\pm.038}$ | $.794_{\pm.017}$ | $.915_{\pm.003}$ | $.572_{\pm.059}$ | $.888_{\pm.020}$ |
| VAEAC | $.724_{\pm.002}$ | $.800_{\pm.000}$ | $.890_{\pm.002}$ | $.827_{\pm.001}$ | $.640_{\pm.054}$ | $.821_{\pm.020}$ |
| MIWAE | $.616_{\pm.029}$ | $.880_{\pm.025}$ | $.809_{\pm.010}$ | $.916_{\pm.003}$ | $.802_{\pm.052}$ | $.891_{\pm.023}$ |
| not-MIWAE | $.629_{\pm.028}$ | $.897_{\pm.021}$ | $.805_{\pm.020}$ | $.916_{\pm.003}$ | $.744_{\pm.048}$ | $.908_{\pm.021}$ |
| MIRACLE | $.671_{\pm.027}$ | $.840_{\pm.027}$ | $.951_{\pm.009}$ | $.920_{\pm.002}$ | $.834_{\pm.040}$ | $.931_{\pm.025}$ |
| ReMasker | $.659_{\pm.027}$ | $.840_{\pm.027}$ | $.947_{\pm.008}$ | $.919_{\pm.002}$ | $.803_{\pm.048}$ | $.910_{\pm.026}$ |
| DrIM$_{\text{BASE}}$ | $.659_{\pm.025}$ | $.780_{\pm.047}$ | $.947_{\pm.006}$ | $.915_{\pm.003}$ | $.730_{\pm.040}$ | $.902_{\pm.021}$ |
| DrIM$_{\text{FINE}}$ | $.691_{\pm.020}$ | $.940_{\pm.022}$ | $.954_{\pm.007}$ | $.915_{\pm.003}$ | $.757_{\pm.041}$ | $.880_{\pm.019}$ |

| | redwine | | | whitewine | | |
|---|---|---|---|---|---|---|
| model | $F_1 \uparrow$ | Model $\uparrow$ | Feature $\uparrow$ | $F_1 \uparrow$ | Model $\uparrow$ | Feature $\uparrow$ |
| Mean | $.428_{\pm.027}$ | $.673_{\pm.092}$ | $.543_{\pm.085}$ | $.386_{\pm.021}$ | $.647_{\pm.096}$ | $.525_{\pm.071}$ |
| kNNI | $.437_{\pm.027}$ | $.697_{\pm.115}$ | $.716_{\pm.064}$ | $.395_{\pm.021}$ | $.675_{\pm.044}$ | $.757_{\pm.056}$ |
| EM | $.436_{\pm.024}$ | $.502_{\pm.122}$ | $.699_{\pm.053}$ | $.398_{\pm.022}$ | $.660_{\pm.096}$ | $.667_{\pm.078}$ |
| SoftImpute | $.421_{\pm.026}$ | $.571_{\pm.116}$ | $.689_{\pm.067}$ | $.387_{\pm.023}$ | $.777_{\pm.078}$ | $.806_{\pm.038}$ |
| MICE | $.430_{\pm.026}$ | $.633_{\pm.103}$ | $.686_{\pm.058}$ | $.390_{\pm.024}$ | $.660_{\pm.081}$ | $.769_{\pm.054}$ |
| missForest | $.426_{\pm.026}$ | $.597_{\pm.108}$ | $.717_{\pm.061}$ | $.387_{\pm.023}$ | $.687_{\pm.093}$ | $.753_{\pm.049}$ |
| Sinkhorn | $.431_{\pm.025}$ | $.679_{\pm.109}$ | $.686_{\pm.056}$ | $.389_{\pm.022}$ | $.660_{\pm.086}$ | $.714_{\pm.043}$ |
| GAIN | $.526_{\pm.009}$ | $.382_{\pm.078}$ | $.812_{\pm.028}$ | $.467_{\pm.013}$ | $.600_{\pm.074}$ | $.822_{\pm.062}$ |
| VAEAC | $.485_{\pm.004}$ | $.403_{\pm.053}$ | $.808_{\pm.018}$ | $.431_{\pm.002}$ | $.500_{\pm.037}$ | $.757_{\pm.025}$ |
| MIWAE | $.431_{\pm.026}$ | $.602_{\pm.094}$ | $.676_{\pm.057}$ | $.389_{\pm.023}$ | $.720_{\pm.074}$ | $.814_{\pm.035}$ |
| not-MIWAE | $.420_{\pm.027}$ | $.622_{\pm.118}$ | $.704_{\pm.057}$ | $.387_{\pm.022}$ | $.740_{\pm.048}$ | $.773_{\pm.044}$ |
| MIRACLE | $.435_{\pm.024}$ | $.487_{\pm.125}$ | $.731_{\pm.057}$ | $.393_{\pm.022}$ | $.760_{\pm.070}$ | $.731_{\pm.045}$ |
| ReMasker | $.415_{\pm.029}$ | $.603_{\pm.092}$ | $.567_{\pm.074}$ | $.381_{\pm.022}$ | $.605_{\pm.104}$ | $.615_{\pm.086}$ |
| DrIM$_{\text{BASE}}$ | $.539_{\pm.010}$ | $.634_{\pm.078}$ | $.853_{\pm.036}$ | $.471_{\pm.009}$ | $.707_{\pm.062}$ | $.835_{\pm.030}$ |
| DrIM$_{\text{FINE}}$ | $.555_{\pm.009}$ | $.451_{\pm.095}$ | $.878_{\pm.015}$ | $.498_{\pm.004}$ | $.580_{\pm.066}$ | $.858_{\pm.023}$ |

Table 15: Machine learning utility for each dataset under **MNARL** at 0.3 missingness. The means and the standard errors of the mean across 10 repeated experiments are reported. ↑ denotes higher is better.

| | abalone | | | anuran | | |
|---|---|---|---|---|---|---|
| model | $F_1$ ↑ | Model ↑ | Feature ↑ | $F_1$ ↑ | Model ↑ | Feature ↑ |
| Mean | $.131_{\pm.005}$ | $-.206_{\pm.080}$ | $.895_{\pm.018}$ | $.817_{\pm.010}$ | $.035_{\pm.100}$ | $.683_{\pm.039}$ |
| kNNI | $.140_{\pm.004}$ | $-.220_{\pm.133}$ | $.924_{\pm.017}$ | $.788_{\pm.007}$ | $-.233_{\pm.126}$ | $.513_{\pm.029}$ |
| EM | $.133_{\pm.005}$ | $-.238_{\pm.125}$ | $.926_{\pm.013}$ | $.801_{\pm.006}$ | $-.313_{\pm.113}$ | $.519_{\pm.038}$ |
| SoftImpute | $.125_{\pm.005}$ | $-.297_{\pm.137}$ | $.914_{\pm.016}$ | $.811_{\pm.007}$ | $-.090_{\pm.123}$ | $.576_{\pm.047}$ |
| MICE | $.126_{\pm.005}$ | $-.457_{\pm.070}$ | $.955_{\pm.013}$ | $.813_{\pm.007}$ | $-.237_{\pm.131}$ | $.577_{\pm.044}$ |
| missForest | $.129_{\pm.005}$ | $-.282_{\pm.104}$ | $.893_{\pm.023}$ | $.810_{\pm.007}$ | $-.166_{\pm.139}$ | $.518_{\pm.032}$ |
| Sinkhorn | $.133_{\pm.004}$ | $-.374_{\pm.093}$ | $.940_{\pm.010}$ | $.803_{\pm.008}$ | $-.172_{\pm.125}$ | $.505_{\pm.032}$ |
| GAIN | $.187_{\pm.005}$ | $-.008_{\pm.180}$ | $.898_{\pm.027}$ | $.875_{\pm.020}$ | $.088_{\pm.117}$ | $.751_{\pm.043}$ |
| VAEAC | $.109_{\pm.023}$ | $.369_{\pm.081}$ | $.740_{\pm.095}$ | $.890_{\pm.001}$ | $.538_{\pm.081}$ | $.870_{\pm.004}$ |
| MIWAE | $.121_{\pm.005}$ | $-.417_{\pm.074}$ | $.921_{\pm.017}$ | $.811_{\pm.007}$ | $-.133_{\pm.142}$ | $.522_{\pm.032}$ |
| not-MIWAE | $.126_{\pm.005}$ | $-.387_{\pm.100}$ | $.933_{\pm.019}$ | $.809_{\pm.008}$ | $-.141_{\pm.156}$ | $.547_{\pm.038}$ |
| MIRACLE | $.129_{\pm.005}$ | $-.276_{\pm.076}$ | $.914_{\pm.012}$ | $.787_{\pm.009}$ | $-.304_{\pm.104}$ | $.589_{\pm.035}$ |
| ReMasker | $.122_{\pm.006}$ | $-.417_{\pm.090}$ | $.910_{\pm.025}$ | $.789_{\pm.007}$ | $-.283_{\pm.135}$ | $.494_{\pm.036}$ |
| DrIM$_{\text{BASE}}$ | $.196_{\pm.004}$ | $.639_{\pm.074}$ | $.940_{\pm.011}$ | $.891_{\pm.007}$ | $-.272_{\pm.168}$ | $.907_{\pm.007}$ |
| DrIM$_{\text{FINE}}$ | $.191_{\pm.007}$ | $.394_{\pm.138}$ | $.890_{\pm.029}$ | $.960_{\pm.005}$ | $-.044_{\pm.117}$ | $.933_{\pm.005}$ |

| | banknote | | | breast | | |
|---|---|---|---|---|---|---|
| model | $F_1$ ↑ | Model ↑ | Feature ↑ | $F_1$ ↑ | Model ↑ | Feature ↑ |
| Mean | $.839_{\pm.015}$ | $.677_{\pm.084}$ | $.860_{\pm.060}$ | $.903_{\pm.009}$ | $.529_{\pm.116}$ | $.783_{\pm.027}$ |
| kNNI | $.846_{\pm.014}$ | $.701_{\pm.089}$ | $.940_{\pm.031}$ | $.890_{\pm.014}$ | $.175_{\pm.115}$ | $.866_{\pm.021}$ |
| EM | $.886_{\pm.013}$ | $.690_{\pm.040}$ | $.940_{\pm.031}$ | $.889_{\pm.017}$ | $.047_{\pm.127}$ | $.861_{\pm.023}$ |
| SoftImpute | $.859_{\pm.014}$ | $.503_{\pm.060}$ | $.920_{\pm.033}$ | $.890_{\pm.015}$ | $.185_{\pm.123}$ | $.849_{\pm.022}$ |
| MICE | $.879_{\pm.016}$ | $.627_{\pm.060}$ | $.940_{\pm.031}$ | $.889_{\pm.018}$ | $.161_{\pm.139}$ | $.865_{\pm.024}$ |
| missForest | $.879_{\pm.014}$ | $.583_{\pm.093}$ | $.940_{\pm.031}$ | $.889_{\pm.016}$ | $.119_{\pm.156}$ | $.861_{\pm.026}$ |
| Sinkhorn | $.857_{\pm.014}$ | $.658_{\pm.073}$ | $.940_{\pm.031}$ | $.898_{\pm.012}$ | $.201_{\pm.138}$ | $.871_{\pm.022}$ |
| GAIN | $.901_{\pm.011}$ | $.587_{\pm.059}$ | $.980_{\pm.020}$ | $.922_{\pm.006}$ | $.525_{\pm.158}$ | $.865_{\pm.031}$ |
| VAEAC | $.845_{\pm.003}$ | $.612_{\pm.047}$ | $.900_{\pm.000}$ | $.843_{\pm.003}$ | $.644_{\pm.047}$ | $.841_{\pm.004}$ |
| MIWAE | $.848_{\pm.016}$ | $.713_{\pm.077}$ | $.940_{\pm.031}$ | $.889_{\pm.018}$ | $.088_{\pm.165}$ | $.872_{\pm.020}$ |
| not-MIWAE | $.840_{\pm.014}$ | $.685_{\pm.046}$ | $1.000_{\pm.000}$ | $.891_{\pm.017}$ | $.162_{\pm.149}$ | $.879_{\pm.020}$ |
| MIRACLE | $.878_{\pm.014}$ | $.577_{\pm.050}$ | $.920_{\pm.061}$ | $.818_{\pm.017}$ | $-.140_{\pm.162}$ | $.755_{\pm.033}$ |
| ReMasker | $.874_{\pm.014}$ | $.580_{\pm.094}$ | $.960_{\pm.027}$ | $.901_{\pm.007}$ | $.274_{\pm.113}$ | $.759_{\pm.035}$ |
| DrIM$_{\text{BASE}}$ | $.877_{\pm.010}$ | $.767_{\pm.060}$ | $.920_{\pm.033}$ | $.858_{\pm.018}$ | $.293_{\pm.131}$ | $.737_{\pm.034}$ |
| DrIM$_{\text{FINE}}$ | $.877_{\pm.008}$ | $.704_{\pm.091}$ | $.960_{\pm.027}$ | $.928_{\pm.007}$ | $.727_{\pm.082}$ | $.884_{\pm.013}$ |

| model | concrete | | | kings | | |
|---|---|---|---|---|---|---|
| | $F_1 \uparrow$ | Model $\uparrow$ | Feature $\uparrow$ | $F_1 \uparrow$ | Model $\uparrow$ | Feature $\uparrow$ |
| Mean | $.307_{\pm.013}$ | $.184_{\pm.110}$ | $.917_{\pm.020}$ | $.413_{\pm.010}$ | $.730_{\pm.065}$ | $.959_{\pm.005}$ |
| kNNI | $.328_{\pm.009}$ | $.534_{\pm.093}$ | $.869_{\pm.028}$ | $.432_{\pm.010}$ | $.640_{\pm.095}$ | $.956_{\pm.003}$ |
| EM | $.359_{\pm.013}$ | $.654_{\pm.080}$ | $.707_{\pm.070}$ | $.432_{\pm.009}$ | $.540_{\pm.083}$ | $.926_{\pm.010}$ |
| SoftImpute | $.307_{\pm.013}$ | $.184_{\pm.110}$ | $.917_{\pm.020}$ | $.416_{\pm.009}$ | $.500_{\pm.099}$ | $.926_{\pm.004}$ |
| MICE | $.322_{\pm.012}$ | $.433_{\pm.104}$ | $.740_{\pm.075}$ | $.430_{\pm.009}$ | $.390_{\pm.075}$ | $.942_{\pm.004}$ |
| missForest | $.317_{\pm.010}$ | $.227_{\pm.090}$ | $.788_{\pm.055}$ | $.423_{\pm.009}$ | $.680_{\pm.074}$ | $.936_{\pm.004}$ |
| Sinkhorn | $.324_{\pm.010}$ | $.393_{\pm.139}$ | $.762_{\pm.061}$ | $.422_{\pm.008}$ | $.730_{\pm.065}$ | $.962_{\pm.003}$ |
| GAIN | $.414_{\pm.009}$ | $.371_{\pm.153}$ | $.900_{\pm.017}$ | $.488_{\pm.008}$ | $.770_{\pm.037}$ | $.963_{\pm.007}$ |
| VAEAC | $.347_{\pm.010}$ | $.472_{\pm.056}$ | $.743_{\pm.036}$ | $.510_{\pm.002}$ | $.760_{\pm.016}$ | $.854_{\pm.002}$ |
| MIWAE | $.276_{\pm.012}$ | $.095_{\pm.167}$ | $.843_{\pm.041}$ | $.415_{\pm.008}$ | $.730_{\pm.079}$ | $.949_{\pm.002}$ |
| not-MIWAE | $.290_{\pm.010}$ | $.441_{\pm.145}$ | $.831_{\pm.041}$ | $.410_{\pm.010}$ | $.682_{\pm.072}$ | $.947_{\pm.003}$ |
| MIRACLE | $.289_{\pm.019}$ | $-.064_{\pm.121}$ | $.860_{\pm.032}$ | $.437_{\pm.007}$ | $.460_{\pm.095}$ | $.941_{\pm.005}$ |
| ReMasker | $.273_{\pm.018}$ | $.141_{\pm.141}$ | $.905_{\pm.019}$ | $.428_{\pm.012}$ | $.430_{\pm.092}$ | $.905_{\pm.011}$ |
| DrIM$_\text{BASE}$ | $.401_{\pm.007}$ | $.386_{\pm.144}$ | $.924_{\pm.016}$ | $.490_{\pm.010}$ | $.920_{\pm.029}$ | $.955_{\pm.004}$ |
| DrIM$_\text{FINE}$ | $.408_{\pm.008}$ | $.489_{\pm.091}$ | $.929_{\pm.014}$ | $.530_{\pm.007}$ | $.940_{\pm.016}$ | $.960_{\pm.002}$ |

| model | letter | | | loan | | |
|---|---|---|---|---|---|---|
| | $F_1 \uparrow$ | Model $\uparrow$ | Feature $\uparrow$ | $F_1 \uparrow$ | Model $\uparrow$ | Feature $\uparrow$ |
| Mean | $.561_{\pm.005}$ | $.740_{\pm.048}$ | $.665_{\pm.030}$ | $.912_{\pm.002}$ | $.630_{\pm.065}$ | $.881_{\pm.023}$ |
| kNNI | $.651_{\pm.007}$ | $.800_{\pm.000}$ | $.950_{\pm.005}$ | $.916_{\pm.002}$ | $.740_{\pm.027}$ | $.902_{\pm.020}$ |
| EM | $.632_{\pm.004}$ | $.810_{\pm.028}$ | $.953_{\pm.005}$ | $.916_{\pm.001}$ | $.697_{\pm.038}$ | $.855_{\pm.020}$ |
| SoftImpute | $.569_{\pm.005}$ | $.777_{\pm.051}$ | $.749_{\pm.026}$ | $.913_{\pm.002}$ | $.702_{\pm.040}$ | $.868_{\pm.014}$ |
| MICE | $.595_{\pm.006}$ | $.840_{\pm.027}$ | $.906_{\pm.012}$ | $.917_{\pm.002}$ | $.714_{\pm.040}$ | $.859_{\pm.021}$ |
| missForest | $.586_{\pm.005}$ | $.740_{\pm.052}$ | $.796_{\pm.022}$ | $.914_{\pm.002}$ | $.750_{\pm.052}$ | $.876_{\pm.018}$ |
| Sinkhorn | $.599_{\pm.006}$ | $.730_{\pm.050}$ | $.856_{\pm.016}$ | $.912_{\pm.003}$ | $.688_{\pm.052}$ | $.884_{\pm.019}$ |
| GAIN | $.580_{\pm.010}$ | $.620_{\pm.042}$ | $.690_{\pm.037}$ | $.911_{\pm.002}$ | $.587_{\pm.015}$ | $.890_{\pm.021}$ |
| VAEAC | $.712_{\pm.002}$ | $.800_{\pm.000}$ | $.887_{\pm.002}$ | $.825_{\pm.001}$ | $.620_{\pm.042}$ | $.838_{\pm.014}$ |
| MIWAE | $.549_{\pm.005}$ | $.860_{\pm.022}$ | $.772_{\pm.025}$ | $.914_{\pm.002}$ | $.713_{\pm.052}$ | $.858_{\pm.022}$ |
| not-MIWAE | $.573_{\pm.005}$ | $.740_{\pm.050}$ | $.760_{\pm.028}$ | $.912_{\pm.002}$ | $.710_{\pm.035}$ | $.869_{\pm.019}$ |
| MIRACLE | $.620_{\pm.009}$ | $.820_{\pm.013}$ | $.932_{\pm.007}$ | $.918_{\pm.002}$ | $.783_{\pm.040}$ | $.878_{\pm.022}$ |
| ReMasker | $.624_{\pm.005}$ | $.780_{\pm.020}$ | $.942_{\pm.007}$ | $.918_{\pm.002}$ | $.760_{\pm.041}$ | $.882_{\pm.024}$ |
| DrIM$_\text{BASE}$ | $.631_{\pm.004}$ | $.760_{\pm.027}$ | $.940_{\pm.005}$ | $.911_{\pm.003}$ | $.626_{\pm.045}$ | $.862_{\pm.015}$ |
| DrIM$_\text{FINE}$ | $.691_{\pm.008}$ | $.950_{\pm.027}$ | $.957_{\pm.006}$ | $.910_{\pm.002}$ | $.720_{\pm.042}$ | $.863_{\pm.014}$ |

| model | redwine | | | whitewine | | |
|---|---|---|---|---|---|---|
| | $F_1 \uparrow$ | Model $\uparrow$ | Feature $\uparrow$ | $F_1 \uparrow$ | Model $\uparrow$ | Feature $\uparrow$ |
| Mean | $.375_{\pm.009}$ | $.422_{\pm.114}$ | $.682_{\pm.072}$ | $.344_{\pm.007}$ | $.540_{\pm.117}$ | $.648_{\pm.069}$ |
| kNNI | $.400_{\pm.010}$ | $.477_{\pm.161}$ | $.705_{\pm.049}$ | $.357_{\pm.008}$ | $.577_{\pm.119}$ | $.804_{\pm.047}$ |
| EM | $.395_{\pm.009}$ | $.529_{\pm.121}$ | $.666_{\pm.052}$ | $.360_{\pm.008}$ | $.656_{\pm.097}$ | $.688_{\pm.070}$ |
| SoftImpute | $.386_{\pm.010}$ | $.426_{\pm.113}$ | $.671_{\pm.050}$ | $.342_{\pm.007}$ | $.530_{\pm.125}$ | $.792_{\pm.044}$ |
| MICE | $.390_{\pm.012}$ | $.467_{\pm.148}$ | $.677_{\pm.052}$ | $.349_{\pm.009}$ | $.450_{\pm.138}$ | $.788_{\pm.062}$ |
| missForest | $.390_{\pm.011}$ | $.402_{\pm.142}$ | $.712_{\pm.050}$ | $.350_{\pm.007}$ | $.606_{\pm.109}$ | $.773_{\pm.055}$ |
| Sinkhorn | $.389_{\pm.011}$ | $.454_{\pm.135}$ | $.720_{\pm.052}$ | $.352_{\pm.006}$ | $.465_{\pm.143}$ | $.774_{\pm.059}$ |
| GAIN | $.506_{\pm.007}$ | $.382_{\pm.110}$ | $.796_{\pm.031}$ | $.427_{\pm.010}$ | $.537_{\pm.075}$ | $.695_{\pm.095}$ |
| VAEAC | $.472_{\pm.005}$ | $.266_{\pm.115}$ | $.785_{\pm.016}$ | $.423_{\pm.004}$ | $.450_{\pm.031}$ | $.700_{\pm.021}$ |
| MIWAE | $.392_{\pm.010}$ | $.506_{\pm.121}$ | $.715_{\pm.060}$ | $.347_{\pm.008}$ | $.537_{\pm.130}$ | $.817_{\pm.046}$ |
| not-MIWAE | $.383_{\pm.011}$ | $.479_{\pm.116}$ | $.662_{\pm.055}$ | $.346_{\pm.007}$ | $.575_{\pm.115}$ | $.755_{\pm.050}$ |
| MIRACLE | $.399_{\pm.009}$ | $.309_{\pm.145}$ | $.699_{\pm.056}$ | $.360_{\pm.005}$ | $.470_{\pm.114}$ | $.765_{\pm.058}$ |
| ReMasker | $.388_{\pm.020}$ | $.185_{\pm.098}$ | $.534_{\pm.056}$ | $.339_{\pm.010}$ | $.342_{\pm.169}$ | $.532_{\pm.076}$ |
| DrIM$_\text{BASE}$ | $.525_{\pm.008}$ | $.334_{\pm.098}$ | $.839_{\pm.021}$ | $.470_{\pm.006}$ | $.510_{\pm.072}$ | $.837_{\pm.032}$ |
| DrIM$_\text{FINE}$ | $.544_{\pm.005}$ | $.105_{\pm.072}$ | $.902_{\pm.013}$ | $.490_{\pm.005}$ | $.541_{\pm.080}$ | $.872_{\pm.027}$ |

Table 16: Machine learning utility for each dataset under **MNARQ** at 0.3 missingness.. The means and the standard errors of the mean across 10 repeated experiments are reported. ↑ denotes higher is better.

| | abalone | | | anuran | | |
|---|---|---|---|---|---|---|
| model | $F_1$ ↑ | Model ↑ | Feature ↑ | $F_1$ ↑ | Model ↑ | Feature ↑ |
| Mean | $.143_{\pm.015}$ | $-.292_{\pm.211}$ | $.774_{\pm.035}$ | $.921_{\pm.021}$ | $.278_{\pm.118}$ | $.549_{\pm.075}$ |
| kNNI | $.146_{\pm.013}$ | $-.284_{\pm.198}$ | $.926_{\pm.021}$ | $.914_{\pm.023}$ | $.333_{\pm.128}$ | $.700_{\pm.052}$ |
| EM | $.142_{\pm.013}$ | $-.317_{\pm.173}$ | $.855_{\pm.033}$ | $.911_{\pm.021}$ | $.223_{\pm.120}$ | $.648_{\pm.052}$ |
| SoftImpute | $.141_{\pm.014}$ | $-.262_{\pm.199}$ | $.924_{\pm.023}$ | $.918_{\pm.022}$ | $.300_{\pm.118}$ | $.717_{\pm.063}$ |
| MICE | $.143_{\pm.015}$ | $-.213_{\pm.194}$ | $.950_{\pm.017}$ | $.920_{\pm.021}$ | $.344_{\pm.120}$ | $.738_{\pm.052}$ |
| missForest | $.145_{\pm.014}$ | $-.262_{\pm.205}$ | $.919_{\pm.024}$ | $.919_{\pm.022}$ | $.325_{\pm.125}$ | $.687_{\pm.050}$ |
| Sinkhorn | $.143_{\pm.013}$ | $-.154_{\pm.172}$ | $.929_{\pm.020}$ | $.917_{\pm.022}$ | $.316_{\pm.121}$ | $.702_{\pm.053}$ |
| GAIN | $.182_{\pm.012}$ | $.021_{\pm.198}$ | $.907_{\pm.019}$ | $.951_{\pm.014}$ | $.490_{\pm.100}$ | $.831_{\pm.028}$ |
| VAEAC | $.136_{\pm.004}$ | $.538_{\pm.049}$ | $.829_{\pm.011}$ | $.893_{\pm.001}$ | $.613_{\pm.082}$ | $.871_{\pm.003}$ |
| MIWAE | $.138_{\pm.016}$ | $-.307_{\pm.202}$ | $.948_{\pm.010}$ | $.921_{\pm.021}$ | $.364_{\pm.128}$ | $.700_{\pm.058}$ |
| not-MIWAE | $.141_{\pm.014}$ | $-.217_{\pm.197}$ | $.950_{\pm.015}$ | $.919_{\pm.022}$ | $.356_{\pm.122}$ | $.726_{\pm.053}$ |
| MIRACLE | $.146_{\pm.014}$ | $-.224_{\pm.185}$ | $.924_{\pm.025}$ | $.917_{\pm.022}$ | $.288_{\pm.126}$ | $.726_{\pm.052}$ |
| ReMasker | $.148_{\pm.013}$ | $-.197_{\pm.154}$ | $.798_{\pm.037}$ | $.913_{\pm.024}$ | $.267_{\pm.112}$ | $.697_{\pm.050}$ |
| DrIM$_{BASE}$ | $.209_{\pm.005}$ | $.827_{\pm.051}$ | $.876_{\pm.022}$ | $.953_{\pm.012}$ | $.631_{\pm.069}$ | $.936_{\pm.013}$ |
| DrIM$_{FINE}$ | $.202_{\pm.006}$ | $.524_{\pm.126}$ | $.817_{\pm.028}$ | $.975_{\pm.007}$ | $.571_{\pm.162}$ | $.928_{\pm.010}$ |

| | banknote | | | breast | | |
|---|---|---|---|---|---|---|
| model | $F_1$ ↑ | Model ↑ | Feature ↑ | $F_1$ ↑ | Model ↑ | Feature ↑ |
| Mean | $.904_{\pm.012}$ | $.832_{\pm.053}$ | $.800_{\pm.052}$ | $.915_{\pm.011}$ | $.469_{\pm.164}$ | $.841_{\pm.032}$ |
| kNNI | $.916_{\pm.011}$ | $.680_{\pm.075}$ | $1.000_{\pm.000}$ | $.915_{\pm.011}$ | $.513_{\pm.176}$ | $.923_{\pm.016}$ |
| EM | $.922_{\pm.007}$ | $.753_{\pm.071}$ | $.960_{\pm.027}$ | $.915_{\pm.012}$ | $.333_{\pm.205}$ | $.884_{\pm.019}$ |
| SoftImpute | $.911_{\pm.011}$ | $.604_{\pm.087}$ | $1.000_{\pm.000}$ | $.915_{\pm.010}$ | $.406_{\pm.202}$ | $.915_{\pm.018}$ |
| MICE | $.922_{\pm.010}$ | $.711_{\pm.070}$ | $1.000_{\pm.000}$ | $.918_{\pm.013}$ | $.231_{\pm.216}$ | $.906_{\pm.025}$ |
| missForest | $.920_{\pm.010}$ | $.600_{\pm.066}$ | $1.000_{\pm.000}$ | $.915_{\pm.012}$ | $.380_{\pm.175}$ | $.927_{\pm.013}$ |
| Sinkhorn | $.912_{\pm.011}$ | $.670_{\pm.073}$ | $1.000_{\pm.000}$ | $.920_{\pm.011}$ | $.409_{\pm.195}$ | $.919_{\pm.022}$ |
| GAIN | $.943_{\pm.004}$ | $.720_{\pm.043}$ | $1.000_{\pm.000}$ | $.929_{\pm.009}$ | $.424_{\pm.161}$ | $.925_{\pm.020}$ |
| VAEAC | $.854_{\pm.003}$ | $.625_{\pm.035}$ | $.900_{\pm.000}$ | $.850_{\pm.003}$ | $.624_{\pm.044}$ | $.850_{\pm.005}$ |
| MIWAE | $.904_{\pm.013}$ | $.774_{\pm.050}$ | $1.000_{\pm.000}$ | $.919_{\pm.011}$ | $.412_{\pm.195}$ | $.900_{\pm.027}$ |
| not-MIWAE | $.902_{\pm.012}$ | $.819_{\pm.045}$ | $.980_{\pm.020}$ | $.915_{\pm.012}$ | $.474_{\pm.171}$ | $.912_{\pm.021}$ |
| MIRACLE | $.919_{\pm.013}$ | $.739_{\pm.056}$ | $1.000_{\pm.000}$ | $.894_{\pm.017}$ | $.088_{\pm.174}$ | $.788_{\pm.038}$ |
| ReMasker | $.918_{\pm.010}$ | $.663_{\pm.076}$ | $1.000_{\pm.000}$ | $.911_{\pm.009}$ | $.194_{\pm.181}$ | $.846_{\pm.028}$ |
| DrIM$_{BASE}$ | $.919_{\pm.009}$ | $.770_{\pm.048}$ | $1.000_{\pm.000}$ | $.909_{\pm.013}$ | $.366_{\pm.155}$ | $.878_{\pm.023}$ |
| DrIM$_{FINE}$ | $.917_{\pm.009}$ | $.753_{\pm.056}$ | $1.000_{\pm.000}$ | $.936_{\pm.004}$ | $.764_{\pm.062}$ | $.917_{\pm.012}$ |

| | concrete | | | kings | | |
|---|---|---|---|---|---|---|
| model | $F_1 \uparrow$ | Model $\uparrow$ | Feature $\uparrow$ | $F_1 \uparrow$ | Model $\uparrow$ | Feature $\uparrow$ |
| Mean | $.351_{\pm.021}$ | $.223_{\pm.175}$ | $.945_{\pm.009}$ | $.521_{\pm.024}$ | $.610_{\pm.119}$ | $.952_{\pm.006}$ |
| kNNI | $.370_{\pm.020}$ | $.634_{\pm.091}$ | $.819_{\pm.032}$ | $.535_{\pm.025}$ | $.580_{\pm.125}$ | $.976_{\pm.001}$ |
| EM | $.381_{\pm.019}$ | $.562_{\pm.112}$ | $.660_{\pm.067}$ | $.506_{\pm.027}$ | $.529_{\pm.122}$ | $.965_{\pm.006}$ |
| SoftImpute | $.351_{\pm.020}$ | $.386_{\pm.123}$ | $.807_{\pm.057}$ | $.525_{\pm.025}$ | $.580_{\pm.131}$ | $.962_{\pm.003}$ |
| MICE | $.367_{\pm.022}$ | $.418_{\pm.152}$ | $.717_{\pm.067}$ | $.532_{\pm.024}$ | $.535_{\pm.124}$ | $.961_{\pm.006}$ |
| missForest | $.360_{\pm.021}$ | $.344_{\pm.134}$ | $.705_{\pm.069}$ | $.526_{\pm.024}$ | $.660_{\pm.109}$ | $.963_{\pm.005}$ |
| Sinkhorn | $.360_{\pm.021}$ | $.639_{\pm.094}$ | $.638_{\pm.074}$ | $.529_{\pm.024}$ | $.630_{\pm.122}$ | $.982_{\pm.001}$ |
| GAIN | $.435_{\pm.012}$ | $.623_{\pm.051}$ | $.731_{\pm.072}$ | $.558_{\pm.018}$ | $.830_{\pm.038}$ | $.973_{\pm.005}$ |
| VAEAC | $.370_{\pm.011}$ | $.552_{\pm.065}$ | $.557_{\pm.069}$ | $.519_{\pm.002}$ | $.760_{\pm.016}$ | $.872_{\pm.004}$ |
| MIWAE | $.330_{\pm.021}$ | $.170_{\pm.154}$ | $.871_{\pm.029}$ | $.520_{\pm.025}$ | $.600_{\pm.134}$ | $.967_{\pm.002}$ |
| not-MIWAE | $.350_{\pm.020}$ | $.334_{\pm.155}$ | $.838_{\pm.040}$ | $.527_{\pm.025}$ | $.630_{\pm.122}$ | $.974_{\pm.003}$ |
| MIRACLE | $.336_{\pm.020}$ | $-.174_{\pm.132}$ | $.829_{\pm.044}$ | $.536_{\pm.025}$ | $.550_{\pm.123}$ | $.968_{\pm.006}$ |
| ReMasker | $.309_{\pm.021}$ | $.146_{\pm.143}$ | $.886_{\pm.020}$ | $.518_{\pm.022}$ | $.620_{\pm.119}$ | $.949_{\pm.007}$ |
| DrIM$_{BASE}$ | $.416_{\pm.011}$ | $.550_{\pm.118}$ | $.869_{\pm.027}$ | $.563_{\pm.012}$ | $.890_{\pm.043}$ | $.976_{\pm.003}$ |
| DrIM$_{FINE}$ | $.416_{\pm.012}$ | $.680_{\pm.087}$ | $.907_{\pm.015}$ | $.586_{\pm.007}$ | $.947_{\pm.016}$ | $.976_{\pm.003}$ |

| | letter | | | loan | | |
|---|---|---|---|---|---|---|
| model | $F_1 \uparrow$ | Model $\uparrow$ | Feature $\uparrow$ | $F_1 \uparrow$ | Model $\uparrow$ | Feature $\uparrow$ |
| Mean | $.646_{\pm.025}$ | $.800_{\pm.042}$ | $.651_{\pm.048}$ | $.921_{\pm.002}$ | $.829_{\pm.048}$ | $.926_{\pm.008}$ |
| kNNI | $.680_{\pm.025}$ | $.720_{\pm.053}$ | $.911_{\pm.014}$ | $.923_{\pm.001}$ | $.867_{\pm.047}$ | $.947_{\pm.010}$ |
| EM | $.687_{\pm.022}$ | $.800_{\pm.042}$ | $.951_{\pm.004}$ | $.893_{\pm.024}$ | $.847_{\pm.051}$ | $.927_{\pm.007}$ |
| SoftImpute | $.652_{\pm.025}$ | $.780_{\pm.049}$ | $.764_{\pm.028}$ | $.922_{\pm.001}$ | $.847_{\pm.049}$ | $.933_{\pm.007}$ |
| MICE | $.668_{\pm.024}$ | $.850_{\pm.037}$ | $.913_{\pm.007}$ | $.922_{\pm.002}$ | $.852_{\pm.037}$ | $.944_{\pm.009}$ |
| missForest | $.663_{\pm.024}$ | $.780_{\pm.049}$ | $.806_{\pm.026}$ | $.923_{\pm.001}$ | $.770_{\pm.045}$ | $.919_{\pm.013}$ |
| Sinkhorn | $.670_{\pm.024}$ | $.720_{\pm.053}$ | $.819_{\pm.021}$ | $.922_{\pm.002}$ | $.783_{\pm.047}$ | $.938_{\pm.010}$ |
| GAIN | $.680_{\pm.022}$ | $.740_{\pm.047}$ | $.764_{\pm.030}$ | $.919_{\pm.002}$ | $.660_{\pm.064}$ | $.922_{\pm.017}$ |
| VAEAC | $.727_{\pm.002}$ | $.800_{\pm.000}$ | $.891_{\pm.002}$ | $.828_{\pm.002}$ | $.627_{\pm.046}$ | $.850_{\pm.015}$ |
| MIWAE | $.641_{\pm.025}$ | $.890_{\pm.028}$ | $.802_{\pm.019}$ | $.920_{\pm.001}$ | $.835_{\pm.046}$ | $.890_{\pm.010}$ |
| not-MIWAE | $.657_{\pm.025}$ | $.780_{\pm.051}$ | $.781_{\pm.026}$ | $.922_{\pm.002}$ | $.807_{\pm.052}$ | $.937_{\pm.009}$ |
| MIRACLE | $.691_{\pm.024}$ | $.820_{\pm.036}$ | $.941_{\pm.008}$ | $.923_{\pm.001}$ | $.820_{\pm.043}$ | $.939_{\pm.010}$ |
| ReMasker | $.677_{\pm.025}$ | $.760_{\pm.050}$ | $.931_{\pm.010}$ | $.923_{\pm.001}$ | $.787_{\pm.037}$ | $.930_{\pm.013}$ |
| DrIM$_{BASE}$ | $.687_{\pm.020}$ | $.760_{\pm.050}$ | $.937_{\pm.006}$ | $.919_{\pm.002}$ | $.776_{\pm.048}$ | $.898_{\pm.010}$ |
| DrIM$_{FINE}$ | $.719_{\pm.016}$ | $.892_{\pm.031}$ | $.942_{\pm.008}$ | $.920_{\pm.002}$ | $.787_{\pm.047}$ | $.907_{\pm.009}$ |

| | redwine | | | whitewine | | |
|---|---|---|---|---|---|---|
| model | $F_1 \uparrow$ | Model $\uparrow$ | Feature $\uparrow$ | $F_1 \uparrow$ | Model $\uparrow$ | Feature $\uparrow$ |
| Mean | $.466_{\pm.018}$ | $.130_{\pm.127}$ | $.787_{\pm.036}$ | $.416_{\pm.018}$ | $.290_{\pm.097}$ | $.485_{\pm.080}$ |
| kNNI | $.469_{\pm.020}$ | $.145_{\pm.114}$ | $.843_{\pm.024}$ | $.423_{\pm.017}$ | $.295_{\pm.090}$ | $.730_{\pm.056}$ |
| EM | $.469_{\pm.021}$ | $.083_{\pm.126}$ | $.859_{\pm.025}$ | $.420_{\pm.019}$ | $.320_{\pm.083}$ | $.655_{\pm.086}$ |
| SoftImpute | $.467_{\pm.020}$ | $.060_{\pm.149}$ | $.874_{\pm.023}$ | $.416_{\pm.018}$ | $.330_{\pm.082}$ | $.811_{\pm.040}$ |
| MICE | $.470_{\pm.020}$ | $.104_{\pm.137}$ | $.858_{\pm.023}$ | $.417_{\pm.019}$ | $.270_{\pm.068}$ | $.777_{\pm.049}$ |
| missForest | $.465_{\pm.021}$ | $.129_{\pm.135}$ | $.879_{\pm.022}$ | $.417_{\pm.018}$ | $.280_{\pm.083}$ | $.770_{\pm.038}$ |
| Sinkhorn | $.468_{\pm.020}$ | $.170_{\pm.110}$ | $.838_{\pm.020}$ | $.419_{\pm.019}$ | $.300_{\pm.095}$ | $.639_{\pm.103}$ |
| GAIN | $.529_{\pm.009}$ | $.458_{\pm.088}$ | $.895_{\pm.012}$ | $.460_{\pm.013}$ | $.710_{\pm.048}$ | $.800_{\pm.040}$ |
| VAEAC | $.490_{\pm.005}$ | $.285_{\pm.062}$ | $.804_{\pm.009}$ | $.435_{\pm.004}$ | $.510_{\pm.043}$ | $.803_{\pm.013}$ |
| MIWAE | $.463_{\pm.021}$ | $.005_{\pm.139}$ | $.813_{\pm.037}$ | $.420_{\pm.018}$ | $.305_{\pm.081}$ | $.741_{\pm.067}$ |
| not-MIWAE | $.463_{\pm.019}$ | $.062_{\pm.125}$ | $.875_{\pm.017}$ | $.419_{\pm.018}$ | $.330_{\pm.090}$ | $.770_{\pm.067}$ |
| MIRACLE | $.467_{\pm.023}$ | $.086_{\pm.151}$ | $.864_{\pm.030}$ | $.420_{\pm.019}$ | $.340_{\pm.093}$ | $.671_{\pm.087}$ |
| ReMasker | $.449_{\pm.023}$ | $.039_{\pm.138}$ | $.745_{\pm.033}$ | $.405_{\pm.019}$ | $.117_{\pm.148}$ | $.572_{\pm.054}$ |
| DrIM$_{BASE}$ | $.542_{\pm.007}$ | $.503_{\pm.064}$ | $.886_{\pm.021}$ | $.462_{\pm.010}$ | $.870_{\pm.047}$ | $.872_{\pm.024}$ |
| DrIM$_{FINE}$ | $.564_{\pm.005}$ | $.411_{\pm.082}$ | $.877_{\pm.012}$ | $.491_{\pm.007}$ | $.577_{\pm.066}$ | $.825_{\pm.034}$ |

### A.9.4 EFFECT OF CONSTRATIVE LEARNING

Table 17: Effect of constrastive learning under **MCAR**. The means and the standard errors of the mean across 10 datasets and 10 repeated experiments are reported. $\uparrow$ denotes higher is better. Values in parentheses indicate the performance difference compared to $DrIM_{BASE}$, and the red highlights the positive improvement.

| Rate | model | $F_1 \uparrow$ | Model $\uparrow$ | Feature $\uparrow$ |
|---|---|---|---|---|
| 0.2 | $DrIM_{BASE}$ | $.665_{\pm.025}$ | $.692_{\pm.026}$ | $.949_{\pm.004}$ |
|  | $DrIM_{FINE}$ | $.678_{\pm.025}$ (+1.95%) | $.690_{\pm.029}$ (-0.29%) | $.948_{\pm.004}$ (-0.11%) |
| 0.4 | $DrIM_{BASE}$ | $.602_{\pm.024}$ | $.301_{\pm.048}$ | $.853_{\pm.012}$ |
|  | $DrIM_{FINE}$ | $.632_{\pm.025}$ (+4.98%) | $.503_{\pm.038}$ (+67.11%) | $.889_{\pm.011}$ (+4.29%) |
| 0.6 | $DrIM_{BASE}$ | $.539_{\pm.024}$ | $.104_{\pm.051}$ | $.708_{\pm.024}$ |
|  | $DrIM_{FINE}$ | $.588_{\pm.025}$ (+8.85%) | $.349_{\pm.044}$ (+233.65%) | $.748_{\pm.019}$ (+5.64%) |
| 0.8 | $DrIM_{BASE}$ | $.471_{\pm.023}$ | $.027_{\pm.050}$ | $.272_{\pm.043}$ |
|  | $DrIM_{FINE}$ | $.503_{\pm.025}$ (+6.81%) | $.176_{\pm.055}$ (+651.85%) | $.309_{\pm.039}$ (+13.59%) |

Table 18: Effect of constrastive learning under **MNARL**. The means and the standard errors of the mean across 10 datasets and 10 repeated experiments are reported. $\uparrow$ denotes higher is better. Values in parentheses indicate the performance difference compared to $DrIM_{BASE}$, and the red highlights the positive improvement.

| Rate | model | $F_1 \uparrow$ | Model $\uparrow$ | Feature $\uparrow$ |
|---|---|---|---|---|
| 0.2 | $DrIM_{BASE}$ | $.660_{\pm.024}$ | $.658_{\pm.031}$ | $.925_{\pm.007}$ |
|  | $DrIM_{FINE}$ | $.673_{\pm.025}$ (+1.97%) | $.711_{\pm.026}$ (+8.05%) | $.938_{\pm.006}$ (+1.42%) |
| 0.4 | $DrIM_{BASE}$ | $.591_{\pm.023}$ | $.326_{\pm.048}$ | $.832_{\pm.014}$ |
|  | $DrIM_{FINE}$ | $.625_{\pm.024}$ (+5.75%) | $.481_{\pm.042}$ (+47.55%) | $.864_{\pm.010}$ (+3.85%) |
| 0.6 | $DrIM_{BASE}$ | $.522_{\pm.022}$ | $.102_{\pm.052}$ | $.652_{\pm.029}$ |
|  | $DrIM_{FINE}$ | $.573_{\pm.024}$ (+9.78%) | $.313_{\pm.045}$ (+205.88%) | $.730_{\pm.021}$ (+12.00%) |
| 0.8 | $DrIM_{BASE}$ | $.449_{\pm.022}$ | $.049_{\pm.051}$ | $.286_{\pm.043}$ |
|  | $DrIM_{FINE}$ | $.492_{\pm.024}$ (+9.57%) | $.235_{\pm.054}$ (+379.59%) | $.287_{\pm.042}$ (+0.35%) |

Table 19: Effect of constrastive learning under **MNARQ**. The means and the standard errors of the mean across 10 datasets and 10 repeated experiments are reported. $\uparrow$ denotes higher is better. Values in parentheses indicate the performance difference compared to $DrIM_{BASE}$, and the red highlights the positive improvement.

| Rate | model | $F_1 \uparrow$ | Model $\uparrow$ | Feature $\uparrow$ |
|---|---|---|---|---|
| 0.2 | $DrIM_{BASE}$ | $.682_{\pm.025}$ | $.771_{\pm.026}$ | $.945_{\pm.005}$ |
|  | $DrIM_{FINE}$ | $.689_{\pm.025}$ (+1.03%) | $.784_{\pm.021}$ (+1.68%) | $.948_{\pm.004}$ (+0.32%) |
| 0.4 | $DrIM_{BASE}$ | $.629_{\pm.025}$ | $.534_{\pm.037}$ | $.850_{\pm.012}$ |
|  | $DrIM_{FINE}$ | $.655_{\pm.025}$ (+4.14%) | $.618_{\pm.033}$ (+15.36%) | $.853_{\pm.012}$ (+0.35%) |
| 0.6 | $DrIM_{BASE}$ | $.583_{\pm.026}$ | $.266_{\pm.050}$ | $.683_{\pm.028}$ |
|  | $DrIM_{FINE}$ | $.614_{\pm.026}$ (+5.32%) | $.434_{\pm.042}$ (+163.27%) | $.748_{\pm.021}$ (+9.96%) |
| 0.8 | $DrIM_{BASE}$ | $.515_{\pm.026}$ | $.151_{\pm.060}$ | $.545_{\pm.033}$ |
|  | $DrIM_{FINE}$ | $.572_{\pm.025}$ (+11.07%) | $.281_{\pm.052}$ (+86.75%) | $.616_{\pm.031}$ (+13.19%) |

A.9.5 SENSITIVITY ANALYSIS FOR MISSINGNESS RATES

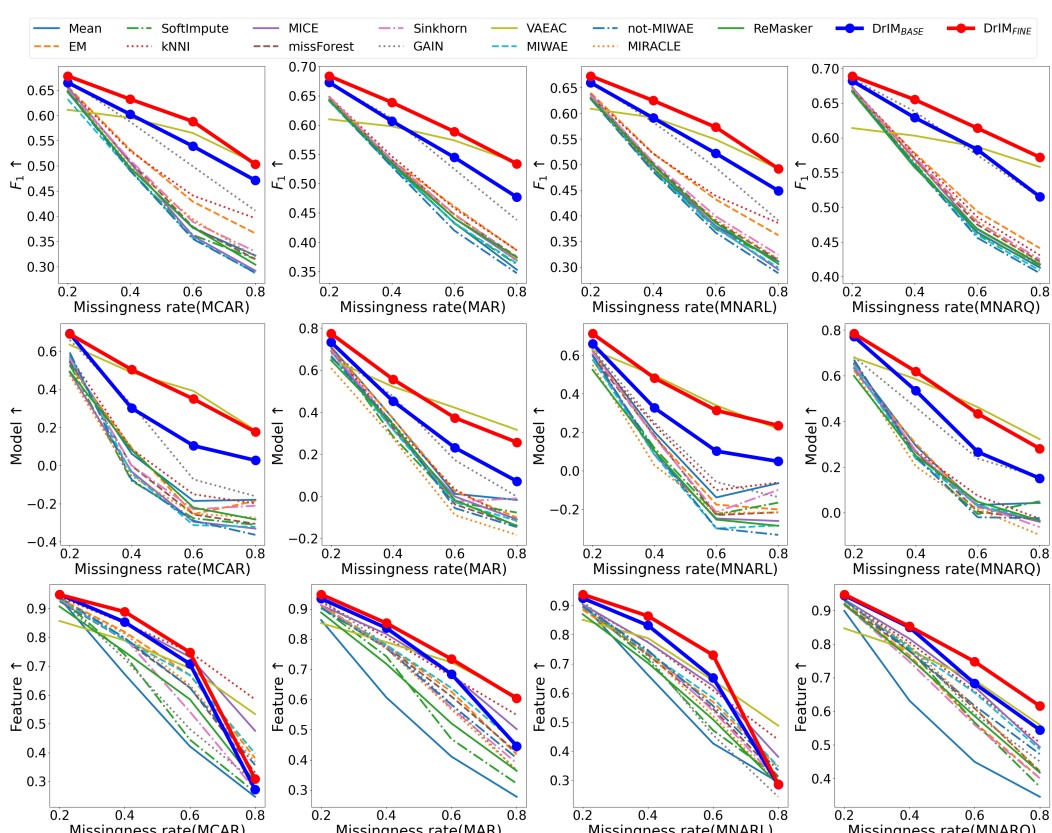

Figure 8: **Sensitivity analysis for missingness rates.** The results MCAR, MAR, MNARL, and MNARQ are shown, with the first row representing classification performance ($F_1$ score), and the last two rows displaying model selection and feature selection performance. The means of the average across 10 datasets and 10 repeated experiments are reported.

