# OpenReview forum: "Context-Driven Missing Data Imputation via Large Language Model"
_ICLR.cc/2025/Conference — Submitted to ICLR 2025_

### Official Review · Reviewer_mF3c · 2024-10-16

**Soundness:** 3
**Presentation:** 3
**Contribution:** 3
**Rating:** 5
**Confidence:** 3

**Summary:**

This paper introduces a context-driven missing imputation method via large language models, termed DrIM. It represents missing values as mask tokens and then maps both continuous and categorical columns into a continuous representation space in a nearest-neighbor-based manner. To find suitable neighbors, this paper incorporates a contrastive framework to refine the representations on the heterogeneous tabular datasets. Experiments show the effectiveness.

**Strengths:**

Tabular data is one of the most used formats in current machine learning studies. It is interesting to construct contextual imputations for missing values with the modern language models. The experiments and attached code were well prepared.

**Weaknesses:**

- The selection of language models: Are there particular factors to consider when selecting BERT-based models? Does it truly represent the current landscape of large language models?

- I recognize that including more mathematical content and pages can enhance reader comprehension. However, I found this paper somewhat lengthy, featuring several meaningless formulas that represent basic knowledge instead of things like theoretical proofs and bounds, such as cosine similarities in Section 3.2.

**Questions:**

I wonder whether it could be applied to multiple-table data or not, like relational databases. I noticed recent efforts that apply machine learning models to many tasks on relational datasets. If not, considering there existed many Missing Data Imputation methods, the use of such manner might be quite limited.

---

> ### Author Response · Authors · 2024-11-24
>
> We sincerely appreciate the time and effort you put into reviewing our work.
>
> **Weakness 1**
>
> We greatly appreciate your insightful clarification. We acknowledge that our initial terminology may have been misleading, as it did not fully align with the current understanding of LLMs. To address this, we have revised the title of the paper as follows: ``Context-Driven Nearest Neighbor Imputation using Language Representation".
>
> We provide a theoretical foundation demonstrating that BERT representations possess two critical properties to support our proposed method in section 3.1 on Pages 3-4 of the revised manuscript: (1) the $L_2$-distance between representation vectors effectively reflects the similarity between observations, and (2) the Euclidean distance between representation vectors acts as an upper bound on the log-likelihood ratio for the observed dataset.
>
> Additionally, we have conducted extensive additional experiments to address the concern regarding the applicability of models beyond BERT using a variety of pre-trained language models, including both autoencoding (e.g., BERT, RoBERTa) and autoregressive (e.g., GPT-2, GPT--NEO, LLaMA) models, across different parameter scales . All experimental results, including additional language models and their configurations, have been incorporated into Appendix 8 on Page 24 of the revised manuscript.
>
> Table 6 on Page 24 indicates that BERT_BASE delivers competitive performance in $F_1$ and feature selection metrics. In contrast, larger models like LLaMA achieve superior results in model selection metrics. However, as shown in Table 7 on Page 24, the incremental performance gains diminish with increasing model size,
> suggesting that the added computational cost of using larger models may not always be justified.
>
> **Weakness 2**
>
> We revised the manuscript as follows:
> - Representation of BERT: In this paper, we selected BERT to extract representation vectors. To support this choice, we have included theoretical results—specifically, Definition 1, Definition 2, and Theorem 1—in Section 3.1 on Pages 3–4 of the revised manuscript. These results provide a partial theoretical foundation for defining neighbors based on the Euclidean distance between CLS tokens in a pre-trained BERT model.
> - Phase2 Contextual Imputation: Instead of presenting lengthy and non-informative formulas, we have clearly defined the nearest neighbor set, which is reflected in Section 3.3 of the revised version on Pages 5-6, Lines 251-381.
> - Fine-tuning Contrastive Learning: We theoretically verify that fine-tuning with contrastive learning enables the representations of BERT to be trained in a manner that captures and maximizes the mutual information between input sequences through Theorem 2 and 3 in section 3.4 on Pages 6-7 of the revised version.
>
> **Question 1**
>
> We sincerely appreciate your thought-provoking question. Regarding your question about applying the method to multiple-table data or relational databases, we understand your concern. Based on our current approach, we focused on single-table data, but we recognize the importance of extending the method to relational databases. This is an interesting avenue for future work, and we plan to explore how the model could be adapted to handle relational data structures.

---

### Official Review · Reviewer_EiBx · 2024-10-31

**Soundness:** 2
**Presentation:** 3
**Contribution:** 2
**Rating:** 3
**Confidence:** 4

**Summary:**

In this manuscript, the authors propose the use of language models to encode tabular data and further employ representations in the semantic space to identify similar data for the purpose of data imputation. Additionally, the authors introduce contrastive learning to enhance the expressiveness of the language models. Extensive experiments are conducted, demonstrating the robust performance of the proposed method.

**Strengths:**

1. This manuscript addresses an important issue: the presence of numerous missing values in tabular data, which is often heterogeneous, thereby complicating the challenge of data representation.
2. This manuscript conducts extensive experiments, and statistical analyses of the results indicate that the proposed method exhibits superior performance."
3. The manuscript is well-written, and the proposed idea is clearly explained, making it easy to follow.

**Weaknesses:**

1. The experimental results presented in the manuscript suggest that the proposed use of textual space to represent tabular data is promising. However, this is concerning because a substantial body of existing work [1,2,3] indicates that language models exhibit insufficient representational capacity for tabular data, particularly for numerical tabular data, especially in settings without fine-tuning. The manuscript employs an approach that contradicts the conclusions of existing studies, which raises my concerns.
2. Despite the extensive experimental results, the authors lack further analysis and discussion of these findings. In fact, the authors have ample text space available for a more in-depth analysis.
3. The authors suggest that the interpretability of the introduced contrastive learning deserves further investigation, which is puzzling. From my perspective, the incorporation of contrastive learning into the BERT model represents merely a straightforward data injection, with the performance improvements being quite evident. If the authors believe that further interpretive analysis is necessary, they should consider integrating techniques such as SFT (supervised fine-tuning) to explore the effectiveness of various data injection methods.
4. The construction of positive example pairs in contrastive learning can be problematic. In  essence, the authors assume missing at random (MAR). However, the MAR assumption might not be satisfied in many domains, i.e., the missing of values in tabular can be not random. For example, in medical domain, the missing value of some medical measurement is the result of medical experts’ judgement. The value is missing because the medical expert think it is unnecessary for some patient to do the test.
5. The authors need to pay attention to the formatting of different citations. For example, citations in lines 36 and 37 of the manuscript should be presented without parentheses.

[1] GLM: General Language Model Pretraining with Autoregressive Blank Infilling, ACL, 2022
[2] Table meets llm: Can large language models understand structured table data? a benchmark and empirical study, WSDM, 2024
[3] NumLLM: Numeric-Sensitive Large Language Model for Chinese Finance, Arxiv, 2024

**Questions:**

Please address the comments in weaknesses.

---

> ### Author Response · Authors · 2024-11-24
>
> **Weakness 1 and 3**
>
> We sincerely thank you for taking the time to compare our paper with other works. We acknowledge that we should have conducted a more comprehensive review of the literature and regret not having done so. As you pointed out, some of the referenced works appear to present conclusions contradicting our findings, and we fully understand your concerns. Taking this opportunity, we have theoretically clarified the following aspects regarding BERT representations and contrastive learning:
>
> We theoretically show that the representation of BERT has the property that the $L_2$-distance between representation vectors reflects the similarity between observations in Section 3.1 of the revised manuscript on Pages 3-4. Based on our empirical results and a review of related works such as [1,2,3], we believe that applying LLMs to tabular data is feasible and that representation learning can further enhance $k$-nearest neighbor imputation.
>
> We also demonstrate that fine-tuning using contrastive learning allows the representations of BERT to be trained in such a way that they maximize the mutual information between input sequences through Theorem 2 and 3 in Section 3.4 on Pages 6-7 of the revised manuscript. In other words, our fine-tuning further enhances the representational capacity of the language model, capturing statistical dependencies between observed and missing entries in the unseen tabular data scenario. Therefore, through fine-tuning with contrastive learning, we can perform missing data imputation using the fully observed data points with high mutual information with the observations containing missing entries. We empirically demonstrate state-of-the-art imputation performance, as shown in Table 1 on Page 8 and discussed in Section 4.2 on page 8 of the revised version.
>
> In Section 3.1 on Page 3-4, we provide a theoretical result that shows the properties of the representation are determined by the pre-training method of the given language model. Therefore, we believe that the approach to obtaining meaningful representations or exploiting these representations from (large) language models will vary depending on the data type and task, and studying this will be part of our future work.
>
> Additionally, since missing data imputation is fundamentally an unsupervised learning task, we did not fully consider incorporating supervised contrastive learning into our fine-tuning method. As the reviewer pointed out, integrating supervised contrastive learning into the fine-tuning process for our proposed method will be part of our future work.
>
> -Reference:
> [1] Hollmann, Noah et al. TabPFN: A Transformer That Solves Small Tabular Classification Problems in a Second. International Conference on Learning Representations (2022).
> [2] Margeloiu, Andrei, et al. TabMDA: Tabular Manifold Data Augmentation for Any Classifier Using Transformers with In-Context Subsetting. ICML 2024 Workshop on In-Context Learning, 2024
> [3] Hegselmann, Stefan, Alejandro Buendia, Hunter Lang, Monica Agrawal, Xiaoyi Jiang, and David Sontag. TabLLM: Few-shot Classification of Tabular Data with Large Language Models. Proceedings of The 26th International Conference on Artificial Intelligence and Statistics, vol. 206, Proceedings of Machine Learning Research, 2023, pp. 5549–5581.
>
> **Weakness 2**
>
> We have revised the experimental results in Section 4.2, Pages 8-9, and Line 417-485, to provide more detailed analyses.
>
> **Weakness 4**
>
> While we acknowledge that the MAR assumption is a limitation and may not always hold in the tabular domain, it serves as a practical framework for our work due to the inherent challenges in obtaining external validation data to verify the assumption of ignorability [1]. Additionally, the $k$-nearest neighbor-based imputation method inherently relies on the assumption that missing data are MAR.
>
> Under the assumption of ignorability, with a slight abuse of notation, the posterior distribution of the missing data (LHS) is given by $P(\{\mathbf{x}_j:\mathbf{m}_j = 0\}|\{\mathbf{x}_j:\mathbf{m}_j = 1\}, \mathbf{m})$ = $P(\{\mathbf{x}_j:\mathbf{m}_j = 0\}|\{\mathbf{x}_j:\mathbf{m}_j = 1\})$.
> Maximizing the RHS term is the objective of the masked language modeling of BERT, which justifies the use of BERT in generating contextualized representations (in BERT, missing entries are represented by MASK tokens).
>
> We have empirically demonstrated that our proposed method not only performs well under the MAR assumption but also achieves notable performance under the MNAR assumption, as shown Table 1 on Page 8 and discussed inSection 4.2 on page 8 of the revised version.
>
> -Reference: [1] Stef van Buuren. Flexible imputation of missing data. 2012
>
> **Weakness 5**
>
> Thank you for carefully reviewing our manuscript. We have corrected the citation formatting in the revision and will ensure that such issues do not occur again.

---

### Official Review · Reviewer_RyGn · 2024-11-03

**Soundness:** 2
**Presentation:** 2
**Contribution:** 2
**Rating:** 3
**Confidence:** 5

**Summary:**

This paper introduces DrIM, a nearest-neighbor-based imputation method for heterogeneous tabular datasets. It utilizes Large Language Models (LLMs) to transform tabular data into text format, enabling better similarity measurements. By replacing missing entries with mask tokens, it captures contextual information. DrIM shows promising results in imputing missing data across various real-world datasets, enhancing machine learning utility (MLu) effectively.

**Strengths:**

- **Research Topic.** The studied problem is interesting and practically important for tabular datasets.

- **Experiments.** Experiments are carried out on 10 datasets with 14 baseline methods, which is quite convincing.

**Weaknesses:**

- **Lack of Novelty.** The LLM part of the paper is merely used for similarity computation. And the contrastive learning part of the paper is also very simple.

- **Confusing Title.** LLM is commonly associated with models such as ChatGPT among current readers, a term that in conjunction with BERT at least appears to deviate from readers' expectations.

- **Article Structure.** The relatively simple method introductions and experimental charts in this article resulting in a limited depth of experimental analysis with just three short paragraphs. I believe this is not reasonable and changes need to be made.

- **Experiments Analysis.** For feature selection task, the performance of the proposed methods drops quickly when missingness rate reaches 0.8, which needs more discussion and explanation. This is just an example, apart from it, more discussion is needed in the experimental section as well.

**Questions:**

- **Experiments.** The author only conducts experiments using BERT. Comparing the results obtained using BERT with those from models like ChatGPT would be valuable and further enhance the contribution of the paper.

---

> ### Author Response · Authors · 2024-11-24
>
> We sincerely appreciate your thoughtful comments.
>
> **Weakness 1**
>
> We acknowledge the reviewer’s concern that using LLMs in our work may appear straightforward and that the contrastive learning framework might seem simple. However, we would like to emphasize the following key aspects that distinguish our approach and underscore its contributions:
>
> Our approach focuses on nearest-neighbor-based imputation using contextual representations from a pre-trained language model, as highlighted on Page 2, Lines 58–61 of the revised manuscript.
>
> To support our proposed method, in section 3.1 on Pages 3-4, we theoretically demonstrate that BERT representations have two key properties: 1) the $L_2$-distance between representation vectors captures the similarity between observations, and 2) the Euclidean distance between representation vectors serves as an upper bound for the log-likelihood ratio of the observed dataset.
>
> Additionally, we theoretically confirm that fine-tuning BERT with contrastive learning enhances its representations to maximize the mutual information between input sequences in Section 3.4 on Pages 6-7. This fine-tuning strategy enables us to perform missing data imputation by utilizing fully observed data points with high mutual information relative to the missing entries.
>
> Furthermore, we empirically demonstrate state-of-the-art performance using BERT, which has significantly fewer parameters and does not require computationally expensive fine-tuning, as shown in Table 1 on Page 8 and discussed in Section 4.2 on Pages 8–9 of the revised version. This efficiency makes our approach particularly appealing for real-world scenarios with constrained computational resources. Additionally, Tables 8–11 in Appendix 9.2 of the revised manuscript show that our method maintains robust performance across various missingness mechanisms, including MCAR and MNARQ, further enhancing its applicability to diverse real-world datasets.
>
> **Weakness 2**
>
> We apologize for the confusion caused by using the term Large Language Model and sincerely appreciate your helpful clarification. We recognize that our initial terminology may have been misleading, as it did not fully consider how LLM is widely perceived today, particularly in association with models like ChatGPT. To address this, we have revised the title of the paper as follows: ``Context-Driven Nearest Neighbor Imputation using Language Representation".
>
> **Question 1**
>
> Thank you for considering ways to enhance our paper's contribution. We have performed extensive additional experiments with a range of pre-trained language models, including autoencoding models (BERT, RoBERTa) and autoregressive models (GPT-2, OPT, LLaMA), spanning various parameter scales. The complete experimental results and their configurations, are provided in Appendix 8 on Page 24 of the revised manuscript.
>
> Table 6 on Page 24 demonstrates that BERT_BASE achieves strong performance in $F_1$ and feature selection metrics, while larger models like LLaMA show better results in model selection metrics. However, as shown in Table 7 on Page 24, the gains in performance diminish with increasing model size, suggesting that the higher computational cost of larger models may not always be justified.
>
> **Weakness 3 and 4**
>
> We revised the manuscript as follows:
> - Related Works: We have revised the Section 2 on Pages 2-3, to clarify our method position among imputation approaches. Our approach is a single imputation method based on the kNNI employing representations for each tubular record using language models.
> - Representation of BERT: We selected BERT to extract representation vectors in this paper. To support this choice, we have included theoretical results, including Definition 1, 2, and Theorem 1, in Section 3.1 on Pages 3-4 of the revised manuscript, providing theoretical justification for defining neighbors based on the Euclidean distance between CLS tokens in a pre-trained BERT model.
> - Phase2 Contextual Imputation: Instead of presenting lengthy and non-informative formulas, we have clearly defined the nearest neighbor set, which is reflected in Section 3.3 of the revised version on Page on Page 5-6 and Lines 251-279.
> - Fine-tuning Contrastive Learning: We theoretically verify that fine-tuning with contrastive learning enables the representations of BERT to be trained in a manner that captures and maximizes the mutual information between input sequences via Theorem 2 and 3 in Section 3.4 on Pages 6-7 of the revised version.
> - Experiment: We have added experiments on imputation fidelity in Section 4.2 of the revised manuscript on Pages 8 and Line 378 and revised the experimental results in Section 4.2 on Pages 8-9 to provide more detailed analyses.

---

> ### Comment · Reviewer_RyGn · 2024-12-03
> **Official Comment by Reviewer RyGn**
>
> Thank you very much for your response. I have read it carefully and will maintain my score.

---

### Official Review · Reviewer_ULA7 · 2024-11-05

**Soundness:** 2
**Presentation:** 3
**Contribution:** 2
**Rating:** 5
**Confidence:** 4

**Summary:**

This paper addresses missing data challenges for tabular data to enhance post-imputation performance, i.e., machine learning utility. Specifically, it proposes a nearest-neighbor-based imputation method for heterogeneous tabular data. The method leverages LLMs to learn the representations of tabular data and incorporate a contrastive learning framework to refine the representations. The experimental results on 10 dataset demonstrate the effectiveness of the proposed method for imputation with high machine learning utility.

**Strengths:**

1. The paper is well-written and easy to follow.
2. The idea of using LLMs for missing value imputation is interesting. Learning representations with LLMs could benefit to capture contextual information for missing values and is helpful for resolving heterogeneous problem.
3. The results on 10 datasets show that the proposed methods achieve better perfomrnace than the SOTA baselines.

**Weaknesses:**

1. Though the idea of leveraging LLMs for imputation is interesting, some studies have proposed similar ideas to this work. It would be more appropriate to incorporate them and discuss both the differences and advantages of the proposed method in relation to them.

    a. CLAIM Your Data: Enhancing Imputation Accuracy with Contextual Large Language Models

    b. LLM-Forest for Health Tabular Data Imputation

2. The proposed method employed BERT as the language model for learning sample representations. However, BERT cannot be considered a large language model (LLM) given that it has a relatively small number of parameters when compared to typical LLMs.
3. For the sensitivity analysis, it can be observed that the two proposed methods experience a greater decline compared to the baselines in terms of the feature selection metric. More detailed analysis ought to be provided.
4. Again, Remasker exhibits greater stability than DrIM_base when it comes to the F_1 metric. Moreover, it appears to outperform DrIM_Fine in the aspect of model selection. However, the current analysis is rather limited in scope, making it insufficient to conduct a comprehensive and in-depth analysis. Table 1 presents the experimental findings at a missingness level of 0.3. Nevertheless, the results are not as satisfactory at other levels of missingness.
5. In the related work section, it is necessary to clarify the position of this paper within the literature. By doing so, the contributions can be highlighted more effectively.
6. It would be better to include other pretrained language models (or even LLMs) for comparison.

**Questions:**

1. It would be better to include more recent related studies and discuss the advantages of this paper.
2. Why do you choose the missingness level of 0.3 to report in Table 1? Are the results consistent as the missingness level varies?

---

> ### Author Response · Authors · 2024-11-24
>
> We sincerely thank the reviewers for their valuable feedback and constructive suggestions.
>
> **Weakness 1, 5, and Question 1.**
>
> As emphasized on Page 2, Lines 58-61 of our revised manuscript, our approach focuses on nearest-neighbor-based imputation using contextual embeddings extracted from a pre-trained language model. CLAIM and LLM-Forest rely on resource-intensive fine-tuning for optimal results and generate the samples for imputation via language model.
>
> In contrast, our method leverages the language model's ability to generate contextualized representations, pre-trained on a large collection of text datasets, and further enhances its capability through fine-tuning with contrastive learning. We provide theoretical justification for defining neighbors based on the representation of (pre-trained) BERT in section 3.1 on Pages 3-4 of the revised manuscript. Moreover, we empirically demonstrate that even with BERT, which has significantly fewer parameters and without computationally expensive fine-tuning, our proposed method achieves state-of-the-art performance, as shown in Table 1 and discussed in Section 4.2 on page 8 of the revised version.
>
> Furthermore, our approach consistently outperforms post-imputation performance across various missingness mechanisms at 0.2, 0.3, and 0.4 missingness rates . Even at higher 0.6 and 0.8 missingness rates, our method achieves competitive results.
>
> In response to the reviewer’s feedback, we have revised the related works on Pages 2-3 of the revised version to clarify our method position.
>
> **Weakness 2 and 6**
>
> We acknowledge that our work may have previously overlooked a comprehensive exploration of the landscape for large language models. To address the concern regarding the applicability of models beyond BERT, we have conducted extensive additional experiments using a variety of pre-trained language models, including both autoencoding (BERT, RoBERTa) and autoregressive (GPT-2, GPT-NEO, LLaMA) models across different parameter scales. All experimental results and additional language models' configurations have been incorporated into Appendix 8 on Page 24 of the revised manuscript.
>
> While Table 6 on Page 24 indicates that BERT_BASE delivers competitive performance in $F_1$ and feature selection metrics, larger models like LLaMA achieve superior results in model selection metrics. However, as shown in Table 7 on Page 24, the incremental performance gains diminish with increasing model size, suggesting that the added computational cost of using larger models may not always be justified.
>
> As the reviewer points out, including LLMs to extract representation vectors would indeed be beneficial. However, we provide a theoretical justification for defining neighbors based on the Euclidean distance between CLS tokens in a pre-trained BERT model in Section 3.1 on Pages 3-4 of the revised manuscript.
>
> Furthermore, in section 3.4 on Pages 6-7 of the revised manuscript, we theoretically verify that fine-tuning BERT with contrastive learning enables its representations to maximize the mutual information between input sequences. This fine-tuning approach allows us to perform missing data imputation by leveraging fully observed data points that share high mutual information with missing entries.
>
> **Weakness 3**
>
> We apologize for any confusion caused by the earlier presentation of results. In response, we have carefully reviewed and corrected the visualizations in Figure 3 on Page 9, and Figure 8 of Appendix 9.5 on Page 37 of the revised manuscript.
>
> As a result, contrary to the reviewer’s concerns, Figure 3 demonstrates that while the performance of each model declines as the missingness rate increases, both DrIM_BASE and DrIM_FINE maintain stable performance compared to the baseline models. Notably, DrIM_FINE consistently outperforms the baseline models in terms of $F_1$ score and feature selection performance. Additionally, we have incorporated a detailed analysis of the sensitivity study into Section 4.2 on Page 9, Lines 466-485 of the revised version, as per your suggestion.
>
> **Weakness 4 and Question 2**
>
> We regret not including all of our experimental results in the initial submission, which may have impacted the reviewer’s understanding of the paper. To address this, we have added experimental results for various missingness rates (0.2, 0.4, 0.6, 0.8) in Tables 8–12 in Appendix 9.2 on Pages 26–27 of the revised manuscript.
>
> Tables 8, 9, and 10 demonstrate that our approach achieves state-of-the-art performance in post-imputation tasks at missingness rates of 0.2, 0.3, and 0.4. Additionally, at a 0.6 missingness rate, as shown in Table 11, our method outperforms baseline models in two out of three post-imputation task performances across each missingness mechanism. Furthermore, at a 0.8 missingness rate , Table 12 shows that our approach exceeds the performance of 12 other baseline models, except for VAEAC.

---

> > ### Comment · Reviewer_ULA7 · 2024-11-25
> > **Thanks for the response**
> >
> > Thanks for the response. I have read the response and would like to keep my evaluation score unchanged. The paper should be further improved. It is preferable to incorporate more elaborate experimental analysis and offer interesting observations rather than including unnecessary definitions and theorems.

---

### Author Response · Authors · 2024-11-24

First, I would like to express my sincere gratitude to the reviewers for their thoughtful advice and detailed evaluation of the manuscript. Thanks to their feedback, we tried to improve the paper and reflected deeply on the theoretical background of the proposed method.

Through this revision process, we derived three theorems on the interpretability of metrics between CLS tokens in language models and included additional results from applying large language models (LLMs).

Given the extensive revisions made compared to the initial draft, I feel apologetic for placing a significant burden on the reviewers to examine the changes. Despite your busy schedules, I kindly ask for your review of the revised manuscript and would greatly appreciate your valuable insights to help us further refine the paper.

---

### Meta-Review · Area_Chair_MoFi · 2024-12-16

**Metareview:**

This paper presents a missing data imputation method combining language representations and nearest neighbor techniques. While the methodology is well-explained and demonstrates strong performance on various datasets, I have concerns regarding its novelty and positioning within the existing literature.

The authors propose a straightforward yet effective approach for imputing missing data in tabular datasets with diverse feature types. The experimental results are impressive, showcasing the method's efficacy. However, the paper's contribution and its place within the broader field require further clarification. As other reviewers have noted, the related work section needs improvement, and the authors should articulate the unique aspects of their approach more clearly.

Additionally, certain experimental results appear counterintuitive and inconsistent. More detailed discussion and further experiments are needed to address these discrepancies and strengthen the analysis. These weaknesses ultimately led to my decision to reject this paper.

**Additional Comments On Reviewer Discussion:**

I appreciate you engaging with the reviewers and providing a comprehensive rebuttal and revised manuscript. The inclusion of additional results using larger language models, beyond the original BERT models, significantly strengthens your response. Furthermore, the added analyses of the experiments enhance the paper's clarity and rigor.

However, concerns remain regarding the novelty and contribution of this work within the field. While the revisions are valuable, they don't fully address the fundamental issue of insufficiently distinguishing this approach from existing literature. Additionally, some inconsistencies in the experimental results persist and require further investigation. To further improve this paper, I recommend focusing on clarifying its unique contributions and addressing the remaining inconsistencies in the results.

---

### Decision · Program_Chairs · 2025-01-22

Reject